# Generation of antigen-specific mature T cells from *RAG1⁻/⁻ RAG2⁻/⁻ B2M⁻/⁻* stem cells by engineering their microenvironment

Patrick C Chang[1,2], Xuegang Yuan[2,9], Alexandre Zampieri[2,9], Chloe Towns[2], Sang Pil Yoo[1,2], Claire Engstrom[2], Steven Tsai[3], Christopher R. Robles[4], Yuhua Zhu[2], Shawn Lopez[2], Amelie Montel-Hagen[2], Christopher S. Seet[3,5,6,7] & Gay M. Crooks ®[2,5,6,7,8] ✉

Pluripotent stem cells (PSCs) are a promising source of allogeneic T cells for off-the-shelf immunotherapies. However, the process of differentiating genetically engineered PSCs to generate mature T cells requires that the same molecular elements that are crucial for the selection of these cells be removed to prevent alloreactivity. Here we show that antigen-restricted mature T cells can be generated in vitro from PSCs edited via CRISPR to lack endogenous T cell receptors (TCRs) and class I major histocompatibility complexes. Specifically, we used T cell precursors from *RAG1⁻/⁻ RAG2⁻/⁻ B2M⁻/⁻* human PSCs expressing a single TCR, and a murine stromal cell line providing the cognate human major histocompatibility complex molecule and other critical signals for T cell maturation. Possibly owing to the absence of TCR mispairing, the generated T cells showed substantially better tumour control in mice than T cells with an intact endogenous TCR. Introducing the T cell selection components into the stromal microenvironment of the PSCs overcomes inherent biological challenges associated with the development of T cell immunotherapies from allogeneic PSCs.

Cell-based immunotherapies in which autologous T cells have been engineered to express chimaeric antigen receptors and antigen-specific T cell receptors (TCRs) have produced impressive clinical responses in patients with otherwise treatment-refractory diseases[1–4]. Despite producing curative results, the widespread application of adoptive cell therapies is limited by the complex and costly manufacturing processes needed for the delivery of patient-specific treatments[2]. Currently, autologously engineered T cell products rely on adequate patient lymphocyte counts and on a clinical status that allows the incorporation

of cell therapy[5–7]. In addition, clinical responses are uneven, at least in part because of the variability of the T cell composition of each patient product[8].

To overcome these constraints, development of an 'off-the-shelf' engineered T cell product has garnered increasing interest. However, the requirement for an allogeneic source with this approach presents two main barriers: donor T cell-mediated alloreactivity (causing graft-versus-host disease)[9] and host-mediated rejection of human leucocyte antigen (HLA)-mismatched allogeneic donor cells[10].

[1]Molecular Biology Interdepartmental Program, University of California, Los Angeles (UCLA), Los Angeles, CA, USA. [2]Department of Pathology and Laboratory Medicine, David Geffen School of Medicine, UCLA, Los Angeles, CA, USA. [3]Division of Hematology-Oncology, Department of Medicine, David Geffen School of Medicine, UCLA, Los Angeles, CA, USA. [4]Department of Computational Medicine, UCLA, Los Angeles, CA, USA. [5]Molecular Biology Institute, UCLA, Los Angeles, CA, USA. [6]Broad Stem Cell Research Center, UCLA, Los Angeles, CA, USA. [7]Jonsson Comprehensive Cancer Center, UCLA, Los Angeles, CA, USA. [8]Department of Pediatrics, David Geffen School of Medicine, UCLA, Los Angeles, CA, USA. [9]These authors contributed equally: Xuegang Yuan, Alexandre Zampieri. ✉e-mail: GCrooks@mednet.ucla.edu

Although the manipulation of these allogeneic mechanisms via gene editing of healthy donor peripheral blood (PB) T cells has been performed[7], current techniques do not guarantee complete ablation of endogenous TCRs, and the process of gene editing carries the risk of genomic translocations and deleterious off-target events. Thus, there is a need to identify self-renewing sources of engineered T cells that allow for the validation and expansion of suitable clones after multiple cycles of gene editing.

Owing to their self-renewing capacity, human pluripotent stem cells (PSCs) are especially amenable to gene editing and clonal selection. A unique hurdle for the use of PSCs, however, is the complex in vitro process of haematopoietic and T cell development that must follow genetic manipulation of PSCs. Recent development of a three-dimensional culture method called the artificial thymic organoid (ATO) allows highly efficient and reproducible T cell differentiation with positive selection of mature, conventional CD8αβ+ (CD3+TCRαβ+CD8αβ+, hereafter SP8) T cells from multiple haematopoietic and PSC sources[11,12]. Induction of T cell development in the ATO system is accomplished using the murine MS5 stromal cell line engineered to express human delta-like ligand 4 (hDLL4) as a transgene. PSCs engineered using lentivirus to express a transgenic tumour-specific TCR[12] or chimaeric antigen receptor[13] can produce antigen-specific T cells after differentiation in ATOs; however, these previous studies were performed in cells with intact endogenous TCR and major histocompatibility complex (MHC) expression, thus allowing positive selection through a broad repertoire of endogenous TCRs and a diverse range of peptide MHCs (pMHCs) presented by PSC-derived cells (self-selection).

In this article, we report a method that provides all the necessary signals required for the positive selection of antigen-specific, mature, conventional CD8αβ+ T cells from PSCs that lack all endogenous TCR rearrangements and MHC class I expression. To prevent the generation of any endogenous TCRs, recombination activation gene 1 (RAG1) and RAG2, which together facilitate V(D)J recombination of germline TCR loci[14], was ablated via CRISPR–Cas9 gene editing in PSCs. To generate the class I MHC-null phenotype, β2-microglobulin (B2M) was targeted for knockout (KO)[15,16]. Full T cell differentiation, including positive selection, was achieved from RAG1−/− RAG2−/− B2M−/− PSCs by combining transgenic expression of a single TCR in PSCs, with exogenous expression of the cognate human MHC in the MS5-hDLL4 stromal line. The resulting SP8 T cells showed potent antigen-specific cytotoxicity in vitro and tumour control in vivo that was superior to PSC-derived T cells with intact endogenous TCR rearrangement. Overall, these results show that the presentation of a single human MHC complex in a murine stromal line can rescue developing human T cells that express a single, exogenous TCR.

## Results

### KO of RAG1 and RAG2 prevents TCR rearrangement and T cell maturation

As RAG1 and RAG2 are located within an ~30 kb region of chromosome 11, complete deletion of their coding sequences was achieved with two single guide RNAs (sgRNAs; Fig. 1a and Supplementary Fig. 1a,b). Following identification of optimal sgRNAs (Supplementary Fig. 1c–g), single-cell RAG1 and RAG2 double KO (DKO) clones were generated from both the H1 and ESI017 parent embryonic stem cell lines, and then differentiated towards the T cell lineage using our previously developed protocol for the ATO system[12] (Fig. 1a and Supplementary Fig. 2a). Unedited (wild type, hereafter WT) PSCs proceeded through development normally, with surface expression of TCR and CD3 and spontaneous positive selection predominantly towards the CD8αβ+ T cell lineage (Fig. 1b,c and Supplementary Fig. 2b,c). Differentiation of both DKO PSC lines proceeded normally into the T lineage-committed phase (CD45+CD5+CD7+; Supplementary Fig. 2d,e), but did not express surface TCR at any stage of differentiation, confirming a functional loss of endogenous TCR rearrangement

(Fig. 1b). Although there was an early, transient population of CD8α+ cells, they did not persist in cultures and lacked surface TCR expression (CD45+CD5+CD7+CD8α+TCRαβ−CD3−), indicating that they did not undergo positive selection. Differentiation of DKO PSCs proceeded no further than the double-positive (DP; CD45+CD5+CD7+CD8α+CD4+) T precursor stage, indicating a failure to undergo positive selection in the absence of endogenous TCRs (Fig. 1c). Output of DKO cultures peaked at 3 weeks, with a dramatic decrease in DP T precursors over the remainder of differentiation, consistent with 'death by neglect' (Fig. 1c,d and Supplementary Fig. 2d,e). Together, the differentiation profile of DKO PSCs showed that RAG1 and RAG2 deletion halts conventional T cell development at the DP T precursor stage and prevents positive selection, similar to previous studies characterizing T cell development from RAG-deficient PSCs[17,18].

### Positive selection of RAG1 and RAG2 knockout PSCs is MHC restricted

We next investigated whether lentiviral expression of a single exogenous TCR could overcome the halt in differentiation seen in DKO PSCs and restore positive selection. The WT and DKO clones of H1, an HLA-A*02:01 positive (A*0201pos) PSC line, and ESI017, an HLA-A*02:01 negative (A*0201neg) PSC line, were transduced with a lentiviral vector encoding the α- and β-chains of the HLA-A*02:01-restricted 1G4 TCR specific for the tumour-associated NYESO157–165 peptide (hereafter, WT + TCR and DKO + TCR PSCs, respectively)[19,20].

Lentiviral transduction of the 1G4 TCR in both H1 and ESI017 DKO PSC lines produced high levels of expression of the transgene in all resulting T cells after differentiation in ATOs (Fig. 2a). Expression of the transgenic TCR did not impair differentiation from either of the WT + TCR PSC lines, with robust production of mature, conventional, single-positive CD8αβ+ T cells (SP8), regardless of MHC expression (Fig. 2b,c and Supplementary Fig. 3).

In contrast, when 1G4 TCR was expressed in DKO lines, only the A*0201pos line (H1) generated SP8 T cells (Fig. 2b and Supplementary Fig. 3). The A*0201neg (ESI017) line did not proceed beyond the DP T precursor stage (Fig. 2b and Supplementary Fig. 3), and cultures were dominated by CD4−CD8− (double negative) cells (Fig. 2b,d). Thus, in the RAG-deficient setting, positive selection only occurred in cultures initiated from PSCs that expressed the cognate MHC to which the transgenic TCR was restricted, showing that 'self-selection' occurs during T cell differentiation.

Although positively selected SP8s generated from H1 DKO + TCR PSCs showed a conventional CD8αβ+ phenotype and naive T cell markers (CD8αβ+CD45RO−CD45RA+CD62L+; Fig. 2c and Supplementary Fig. 3), the overall SP8 percentage and yield per ATO aggregate was significantly lower compared with that of the WT line, indicating that rescue of positive selection was incomplete with the exogenous 1G4 TCR in a RAG-deficient setting (Fig. 2d,e).

### Generating RAG1, RAG2 and B2M triple-KO PSCs

To remove class I MHC expression, a CRISPR–Cas9 sgRNA was designed to target the start of the B2M coding sequence to generate insertion–deletion (indel) mutations (Fig. 3a and Supplementary Fig. 4a–e) in ESI017 DKO + TCR PSCs, thus producing RAG1−/− RAG2−/− B2M−/− (that is, triple KO (TKO)) PSCs transduced with the 1G4 TCR (TKO + TCR, hereafter). Edited cells were isolated by flow cytometry based on a class I MHC-null phenotype (HLA-A,-B,-C−B2M−; Supplementary Fig. 4f). TKO + TCR PSCs were expanded and growth, morphology and karyotype remained unchanged from the parent or DKO + TCR PSC lines.

### Engineering ATOs to induce positive selection from TKO PSCs

As class I MHC molecules were ablated in TKO + TCR PSCs, rescue of positive selection via the transgenic TCR could not be achieved through self-selection on MHC from PSC-derived cells (Fig. 3b). Therefore, to provide a positive selection signal for developing DP T

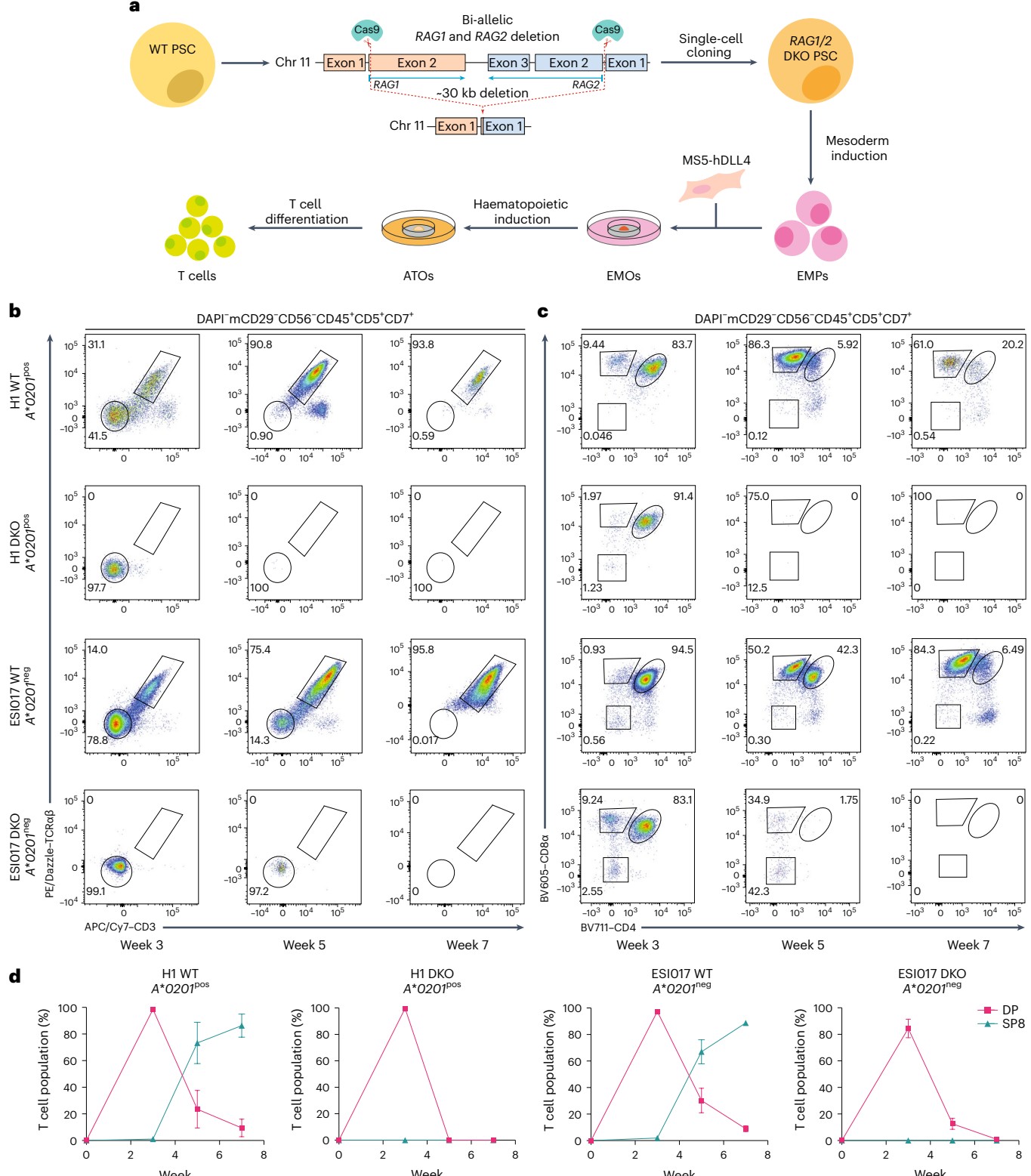

**Fig. 1 | Inhibition of T cell development at the DP stage of T cell differentiation by DKO of *RAG1* and *RAG2*. a**, Schematic showing the generation of clonal *RAG1* and *RAG2* DKO PSCs derived from the H1 or ESI017 parent embryonic stem cell lines and their differentiation within the ATO system. **b**, Representative flow cytometry analysis of ATOs generated from unedited (WT) and DKO PSCs from the *HLA-A*02:01* positive (*A*0201*pos) H1 and HLA-A*02:01 negative (*A*0201*neg) ESI017 parent lines. Surface expression of TCRαβ and CD3 from T lineage-committed cells is shown at the indicated time points. **c**, Temporal dynamics of CD8α and CD4 expression in T lineage-committed cells from WT and DKO PSCs differentiated in the ATO system are shown at the indicated

time points. In **b** and **c**, the gating strategy is shown above the plots, and the numbers in the plots indicate the percentage of cells within each gate. Gates for individual populations are drawn on flow cytometry plots. Fluorophores are as follows: DAPI (4′,6-diamidino-2-phenylindole); PE (phycoerythrin); APC (allophycocyanin); BV (Brilliant Violet). **d**, The percentage of T cell populations from the DAPI⁻mCD29⁻CD56⁻CD45⁺CD5⁺CD7⁺ gate in **c** at the indicated time points (data shown as mean ± s.e.m.; H1, $n = 3$ independent experiments; ESI017, $n = 4$ independent experiments). Populations are defined as DP (CD8α⁺CD4⁺) and mature SP8 (CD8α⁺CD4⁻TCRαβ⁺CD3⁺). Chr 11, chromosome 11.

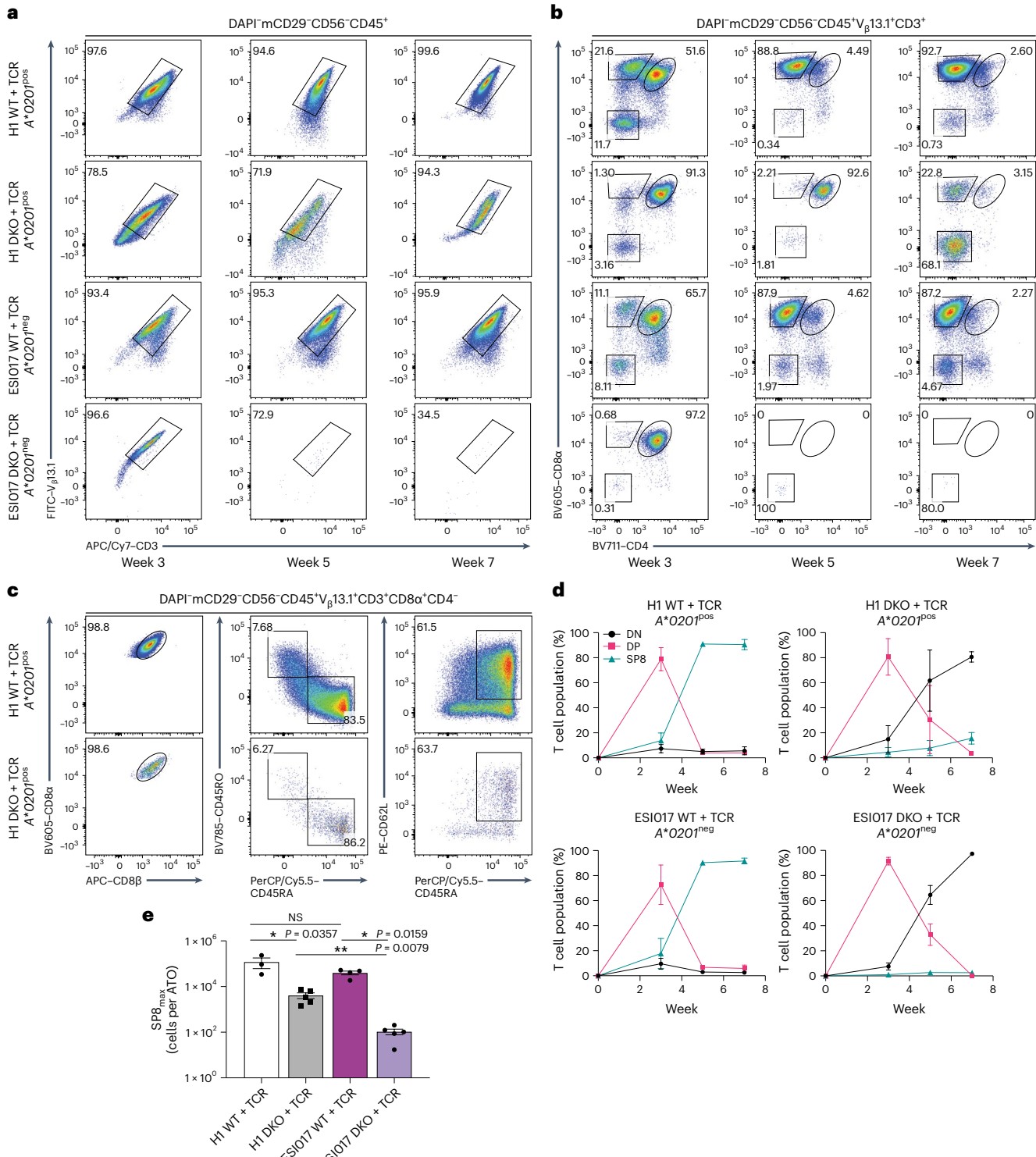

**Fig. 2 | Class I MHC-restricted rescue of T cell development by 1G4 TCR in *RAG1* and *RAG2* DKO PSCs. a,b**, Representative flow cytometry analysis of WT and *RAG1* and *RAG2* DKO PSCs from both parent lines, H1 (*A*0201*pos) and ESI017 (*A*0201*neg), stably transduced to express the *HLA-A*02:01*-restricted 1G4 TCR (WT + TCR and DKO + TCR). **a**, Surface expression of the transgenic V$_\beta$13.1 and CD3 (gated as shown) during ATO differentiation of PSC-derived cells at the indicated time points. **b**, Differentiation kinetics of V$_\beta$13.1+CD3+ T cells based on CD8α and CD4 expression at the indicated time points of ATO differentiation. The gating strategy is indicated above the plots, and the numbers in the plots indicate the percentage of cells within each gate. **c**, At 7 weeks of T cell differentiation, ATO-derived H1 WT + TCR and DKO + TCR SP8 T cells (gated as shown) were analysed for maturation markers of conventional T cells as shown;

the numbers indicate the percentage of cells within each gate. **d**, The percentage of T cell populations from the DAPI−mCD29−CD56−CD45+V$_\beta$13.1+CD3+ gate in **b** at the indicated time points (data shown as mean ± s.e.m.; H1, *n* = 3 independent experiments; ESI017, *n* = 4 independent experiments). Populations are defined as double negative (DN; CD8α−CD4−), DP (CD8α+CD4+) and SP8 (CD8αβ+CD4−). **e**, Maximum SP8 T cell output per ATO reached over the 7 week course of T cell differentiation (SP8$_{max}$). The mean ± s.e.m. (*$P$ < 0.05, **$P$ < 0.01, two-tailed Mann–Whitney *U*-test) is shown for each group (H1 WT, *n* = 3 independent experiments; ESI017 WT, *n* = 4 independent experiments; H1 DKO + TCR and ESI017 DKO + TCR, *n* = 5 independent experiments). NS, not significant; FITC, fluorescein isothiocyanate; PerCP, peridinin-chlorophyll-protein.

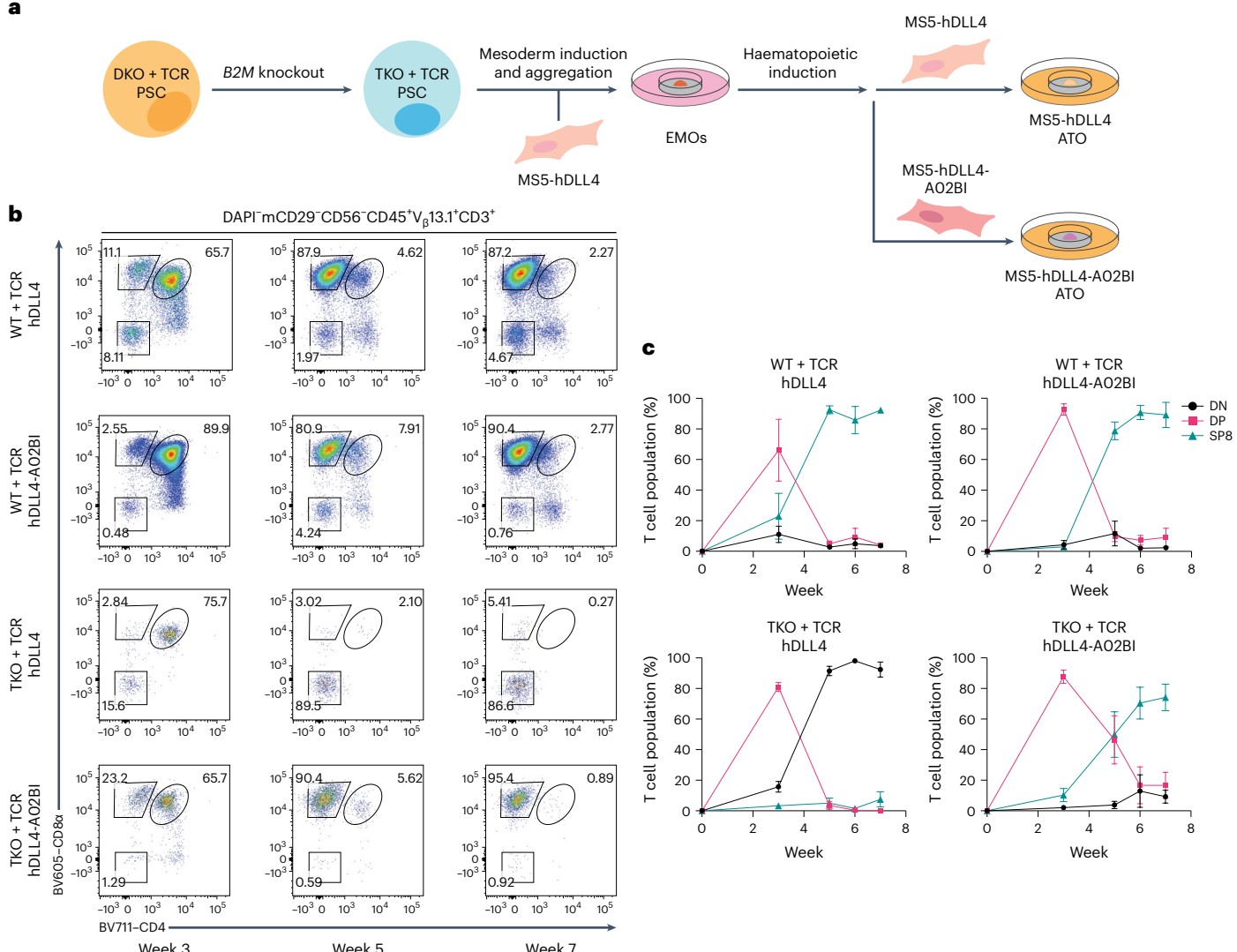

**Fig. 3 | Engineering the ATO stroma to express *HLA-A\*02:01* rescues differentiation of 1G4 TCR-transduced *RAG1*, *RAG2* and *B2M* TKO PSCs.**
**a**, Schematic showing ESI017 TKO + TCR PSC generation. ESI017 DKO + TCR PSC clones were gene edited to KO *B2M* to generate the polyclonal ESI017 TKO + TCR line. Following haematopoietic induction, EMOs were collected and then aggregated with either hDLL4 or hDLL4-A02BI stroma for T cell differentiation in ATOs. **b**, Differentiation kinetics of WT + TCR and TKO + TCR T cells with hDLL4 or hDLL4-A02BI at the indicated time points of ATO differentiation. The gating strategy is indicated above the plots, and the numbers within the plots indicate the percentage of cells within each gate. **c**, The percentage of T cell populations from the DAPI⁻mCD29⁻CD56⁻CD45⁺V$_\beta$13.1⁺CD3⁺ gate in **b** at the indicated time points (data are shown as mean ± s.e.m.; WT + TCR with hDLL4 stroma, $n = 4$ independent experiments; WT + TCR with hDLL4-A02BI stroma, $n = 3$ independent experiments; TKO + TCR with hDLL4 stroma, $n = 4$ independent experiments; TKO + TCR with hDLL4-A02BI stroma, $n = 5$ independent experiments). The populations are defined as DN (CD8α⁻CD4⁻), DP (CD8α⁺CD4⁺) and SP8 (CD8αβ⁺CD4⁻).

precursors, the MS5-hDLL4 line (hDLL4 hereafter) was further transduced to express *HLA-A\*02:01* (the cognate MHC for the 1G4 TCR), with or without human *B2M* (h*B2M*), generating the hDLL4-A\*0201-hB2M (hDLL4-A02B) and hDLL4-A\*0201 (hDLL4-A02) lines, respectively (Supplementary Fig. 5). A subset of the hDLL4-A02B stroma was transduced to also express cell adhesion molecule intercellular adhesion molecule 1 (*ICAM1*), which has been shown to interact with lymphocyte function-associated antigen 1 (*LFA1*) on T cells to increase the strength of signalling through the TCR and survival of thymocytes[21–23]. The resulting hDLL4-A\*0201-hB2M-ICAM1 stroma is hereafter referred to as 'hDLL4-A02BI'.

Full differentiation of WT + TCR PSCs was induced in the ATO using either standard hDLL4 or modified stromal lines expressing the additional transgenes mentioned above (Fig. 3a,b and Supplementary Fig. 6a–c). In contrast, development of T cells from TKO + TCR PSCs

was halted at the DP stage of development using either hDLL4 (Fig. 3b and Supplementary Fig. 6d–f) or hDLL4-A02 stroma (Supplementary Fig. 7a). In these conditions, DP T precursors from TKO + TCR PSCs decreased rapidly (Fig. 3b,c and Supplementary Fig. 7b), similar to previous experiments with DKO + TCR PSCs (Fig. 2d).

In contrast, stromal conditions that expressed both *HLA-A\*02:01* and h*B2M* induced positive selection of SP8 cells from TKO + TCR PSCs (Fig. 3b,c and Supplementary Figs. 6e and 7a). Interestingly, although inclusion of ICAM1 in the hDLL4-A02BI stroma did not change the percentage of SP8 T cells in ATO cultures, it significantly increased the output of SP8 T cells per ATO (Supplementary Fig. 7a–c). TKO + TCR SP8 T cells reached peak enrichment in organoid cultures after 6 weeks of T cell differentiation (Fig. 4a,b), and showed a mature, conventional, naive phenotype (Fig. 4c and Supplementary Fig. 6e,f). In addition, the use of hDLL4-A02BI stroma in ATOs induced

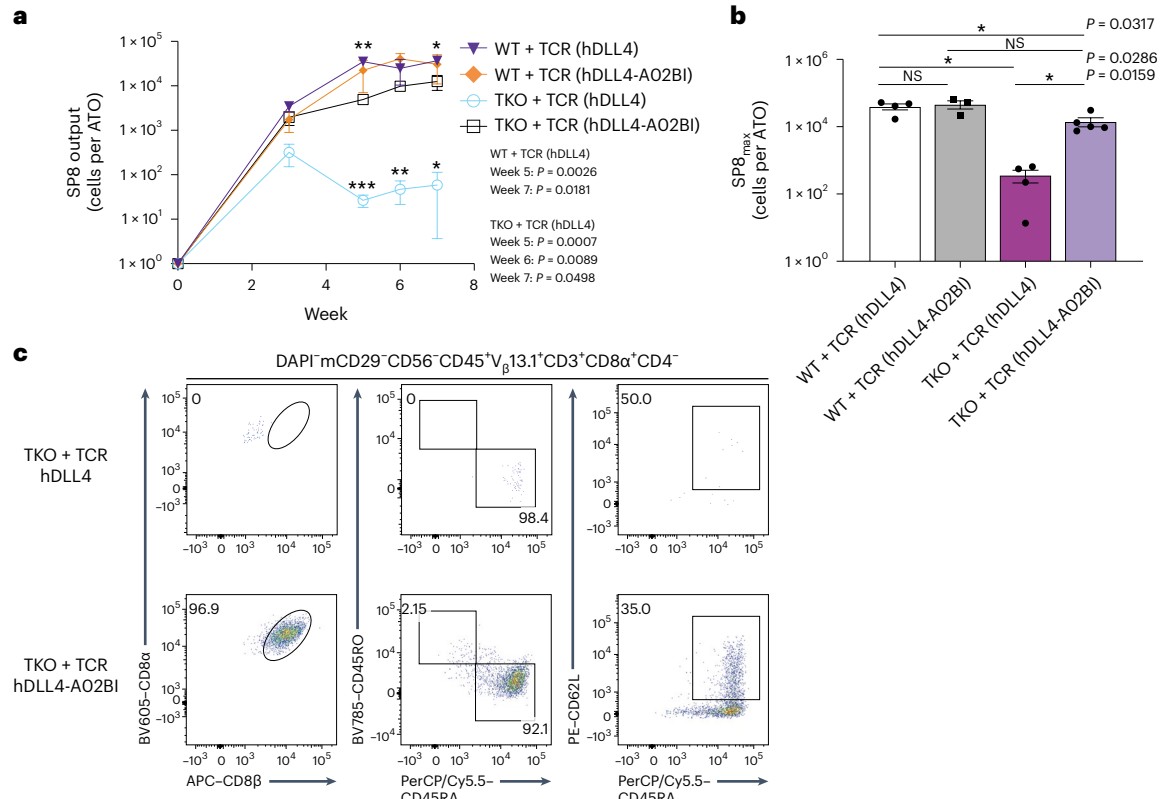

**Fig. 4 | Output and phenotypic characterization of SP8 T cells from *RAG1*, *RAG2* and *B2M* TKO PSCs. a**, Output of SP8 T cells per WT + TCR or TKO + TCR PSC ATO aggregated with hDLL4 or hDLL4-A02BI stroma over the 7 week course of T cell differentiation (mean ± s.e.m.; *P* values are shown compared with TKO + TCR with hDLL4-A02BI stroma; *P < 0.05, **P < 0.01, ***P < 0.001, two-tailed unpaired *t*-test; test is not significant if no *P* value is indicated; WT + TCR with hDLL4 stroma, *n* = 4 independent experiments; WT + TCR with hDLL4-A02BI stroma, *n* = 3 independent experiments; TKO + TCR with hDLL4 stroma, *n* = 4 independent experiments; TKO + TCR with hDLL4-A02BI stroma, *n* = 5 independent experiments). **b**, SP8max T cell output per ATO reached over the

7 week course of T cell differentiation. The mean ± s.e.m. (*P < 0.05, two-tailed Mann–Whitney *U*-test) is shown for each group (WT + TCR with hDLL4 stroma, *n* = 4 independent experiments; WT + TCR with hDLL4-A02BI stroma, *n* = 3 independent experiments; TKO + TCR with hDLL4 stroma, *n* = 4 independent experiments; TKO + TCR with hDLL4-A02BI stroma, *n* = 5 independent experiments). **c**, At 7 weeks of T cell differentiation, TKO + TCR PSC-derived SP8 T cells were analysed for maturation markers of conventional T cells. The gating strategy is indicated above the plots, and the numbers within the plots indicate the percentage of cells within each gate.

positive selection from a TKO line derived from the H1 parent PSC line transduced with the 1G4 TCR (Supplementary Fig. 7d), and generated SP8 T cells with the *HLA-A\*02:01*-restricted F5 TCR recognizing MART1~26–35~ peptide, expressed in ESI017 TKO PSCs (TKO + F5 TCR) (Supplementary Fig. 8a,b).

Previously published approaches generate SP8 T cells through agonist selection or antigen stimulation of isolated DP precursors[24–26]. Although direct comparison of these complete protocols with the model presented here is not possible, we attempted to compare these methods using the TKO + TCR DP precursors collected from week 3 ATOs. DP precursors were activated through CD3 engagement (OKT3 antibody or anti-CD3/CD28 microbeads) or K562 artificial antigen-presenting cells (aAPCs) engineered to express the cognate antigen and co-stimulatory factors CD80, CD83 and CD137L (4-1BBL) (Supplementary Fig. 9a). SP8s generated through activation of ATO⁻ DP precursors through CD3 engagement adopted a CD8αβ⁺CD45RO⁻CD45RA⁺CD62L⁻ phenotype (Supplementary Fig. 9b), but did not expand with further stimulation in the absence of PB mononuclear cell feeders (Supplementary Fig. 9c,d). SP8s generated through aAPC stimulation of DPs initially adopted a CD8αβ⁺CD45RO^intermed^CD45RA^intermed^CD62L⁻ phenotype and achieved ~15-fold further expansion either with additional aAPC or phytohaemagglutinin stimulation; in both methods of expansion, cells converted to a CD8αβ⁺CD45RO⁻CD45RA⁺CD62L⁻ phenotype similar to that seen with CD3-based

selection (Supplementary Fig. 9c,d). In contrast, ATO-generated SP8s displayed a typical naive mature (CD8αβ⁺CD45RA⁺CD62L⁺) phenotype (Supplementary Figs. 3f and 6e,f) after 6 weeks of ATO culture, and adopted a T effector memory phenotype (CD8αβ⁺CD45RO⁺CD45RA⁻CD62L⁻CD27⁻CCR7⁻) after isolation and expansion (Supplementary Fig. 9e).

**Transcriptomic analysis of TKO + TCR SP8 T cells**

To characterize the transcriptomic identity of TKO + TCR SP8 T cells, single-cell RNA sequencing (scRNA-seq) was performed on isolated SP8 cells from TKO + TCR ATOs along with SP8s from WT PSC-derived controls (WT with hDLL4 stroma, WT + TCR with hDLL4 stroma and WT + TCR with hDLL4-A02BI stroma). In addition, scRNA-seq datasets from thymic SP8 (ref. 27), whole PB[28,29] and unpublished PB SP8 datasets were incorporated for comparison. Dimensionality reduction through uniform manifold approximation and projection (UMAP) (Fig. 5a and Supplementary Fig. 10a,b) and quantification of SP8 T cell lineage genes (Fig. 5b and Supplementary Fig. 10c,d) indicated that TKO + TCR SP8 T cells closely resemble SP8s derived from unedited PSCs from thymus and PB. In addition, unsupervised hierarchical clustering and pairwise Pearson's correlation analysis of global gene expression for each sample showed that TKO + TCR SP8 T cells cluster and closely correlate with WT ATO-derived SP8 T cells, and with isolated thymic and PB SP8s (Fig. 5c).

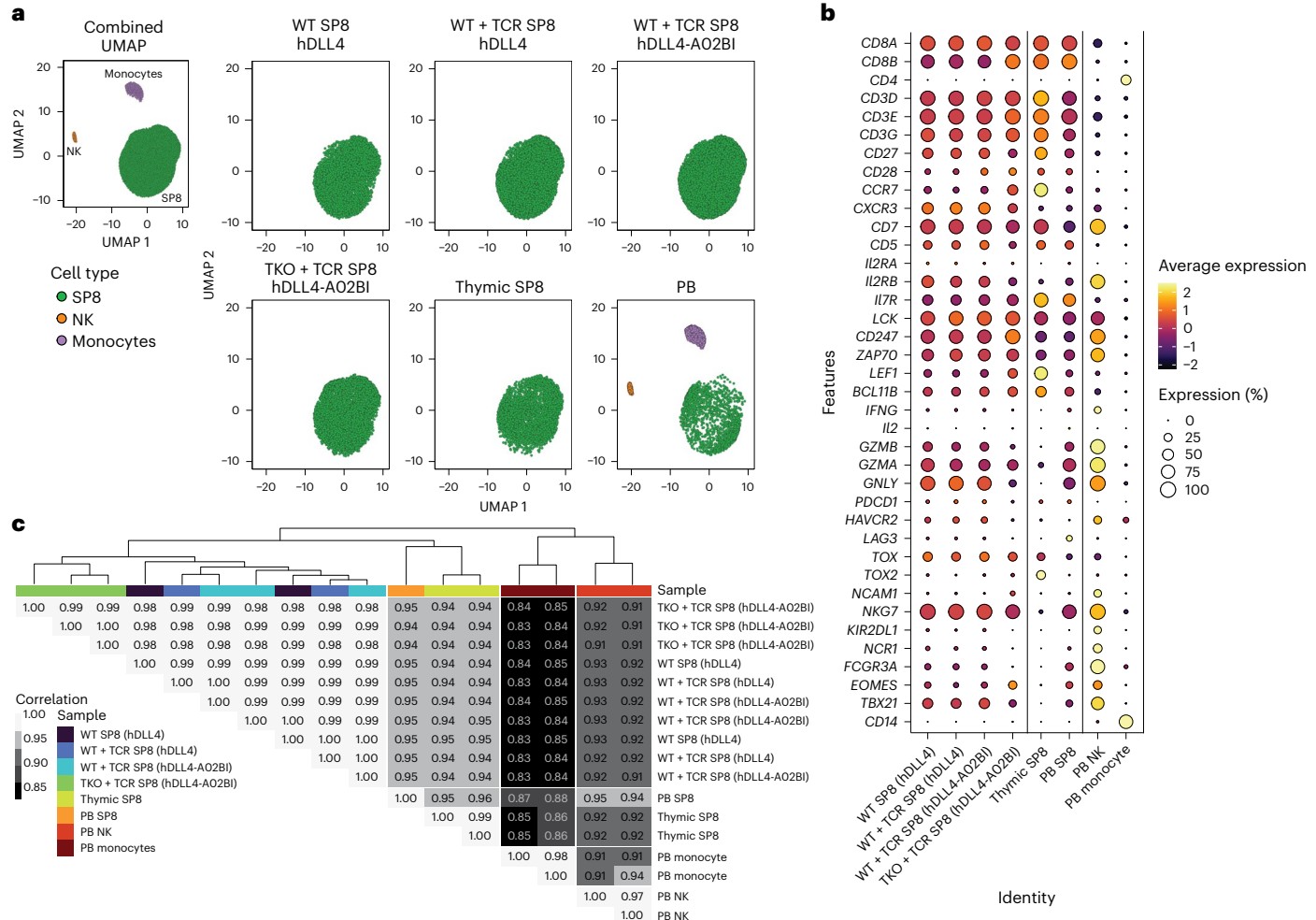

**Fig. 5 | scRNA-seq profiling of PSC-derived, peripheral and thymus SP8s.**
**a**, UMAP visualization of ATO-derived SP8 T cells from WT PSCs with hDLL4 stroma (*n* = 1 independent experiment, 4,710 cells total), WT + TCR PSCs with hDLL4 (*n* = 2 independent experiments, 8,618 cells total) and hDLL4-A02BI (*n* = 3 independent experiments, 12,797 cells total) stroma, and TKO + TCR PSCs with hDLL4-A02BI stroma (*n* = 3 independent experiments, 7,930 cells total) compared with thymic SP8 (*n* = 2 independent experiments, 4,793 cells

total), PB SP8 (*n* = 1 independent experiments, 2,083 cells total), PB NK (*n* = 2 independent experiments, 684 cells total) and PB monocytes (*n* = 2 independent experiments, 2,630 cells total). **b**, Average expression of specific genes from each scRNA-seq sample group. **c**, Dendrogram of hierarchical clustering analysis and heatmap showing Pearson's correlation of global gene expression for all pairwise combinations between each sample, listed individually to the right of the heatmap.

## TCR mispairing enables MHC-independent positive selection

Despite the presence of positive selection from TKO + TCR PSCs, overall SP8 T cell output was reduced compared with WT + TCR PSCs (Fig. 4a,b). As positive selection in the WT + TCR (*A\*0201*neg) ES017 line occurred in the absence of the 1G4 TCR's cognate MHC (Fig. 2b,d), we hypothesized that in the presence of competent RAG1 and RAG2, mispairing between endogenous and exogenous TCR chains would broaden the range of pMHCs that could induce positive selection and circumvent the cognate MHC restriction of the 1G4 TCR.

To investigate the mechanism by which WT T cells carrying a transgenic TCR escaped MHC restriction, SP8 T cells were isolated from ATOs generated from WT, WT + TCR and TKO + TCR PSCs, and sequenced using the 10X Genomics platform to detect exogenous and endogenous TCR V$_\alpha$ and V$_\beta$ chains at single-cell resolution.

Although all WT + TCR SP8 cells expressed the 1G4 transgene, TCR sequencing showed that only 21% ± 3.06% of WT + TCR SP8 exclusively expressed 1G4 (Fig. 6a); endogenously rearranged TCR V$_\alpha$ chains were detected in 76.78% ± 4.04% (mean ± s.d.) of WT + TCR SP8 T cells generated using hDLL4 stroma (*n* = 2, 8,618 cells total) and in 76.53% ± 1.56% of WT + TCR SP8 T cells generated with hDLL4-A02BI stroma

(*n* = 3, 12,797 cells total). A diverse TCR V$_\alpha$ repertoire was observed in WT + TCR SP8 cells (Fig. 6b), indicating (as expected) that the exogenous TCR transgene was ineffective in suppressing endogenous TCR V$_\alpha$ rearrangement during T cell development.

Interestingly, endogenously rearranged TCR V$_\beta$ chains were also detected in addition to the exogenous 1G4 TCR in 21.00% ± 9.18% of WT + TCR SP8 T cells generated with hDLL4 stroma and in 13.32% ± 3.57% of WT + TCR SP8 T cells generated with hDLL4-A02BI stroma (Fig. 6a,c). Typically, allelic exclusion of endogenous TCR V$_\beta$ loci is observed when an exogenous TCR is expressed before V(D)J recombination[30,31]; however, these results indicate that allelic exclusion of the endogenous TCR V$_\beta$ was not complete in the context of in vitro PSC differentiation of T cells (Fig. 6a). Whereas only a minority of T cells co-expressed endogenous TCR V$_\beta$ in addition to the exogenous TCR (Fig. 6a), the TCR V$_\beta$ repertoire was still broadly distributed (Fig. 6c).

As expected, the TKO + TCR SP8 T cells expressed only the transgenic 1G4 TCR, confirming that *RAG1* and *RAG2* deletion prevented all endogenous TCR V$_\alpha$ and TCR V$_\beta$ rearrangement (*n* = 3, 7,930 cells total; Fig. 6a–c). Combined with the previous observation that selection in DKO + TCR PSCs was MHC restricted (Fig. 2a,b and Supplementary

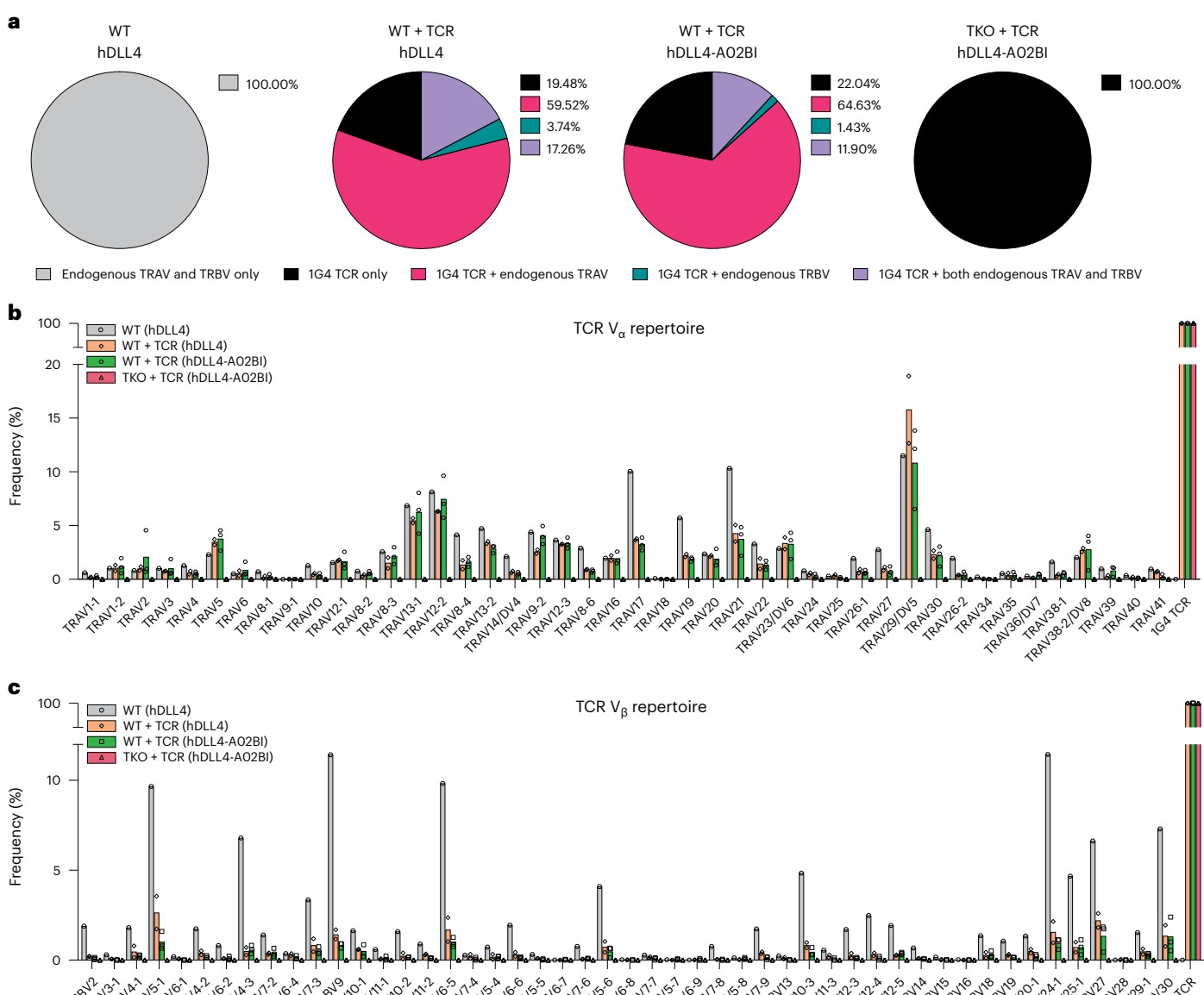

**Fig. 6 | Mispairing between endogenous TCR chains and transgenic TCR chains is absent in SP8s derived from TKO + TCR PSCs. a**, After 6 weeks of T cell differentiation, mature, conventional SP8 T cells were isolated for TCR repertoire analysis at a single-cell resolution from the following ATO conditions: WT PSCs with hDLL4 stroma (*n* = 1 independent experiment), WT + TCR PSCs with hDLL4 stroma (*n* = 2 independent experiments), WT + TCR PSCs with hDLL4-A02BI stroma (*n* = 3 independent experiments) and TKO + TCR PSCs with hDLL4-A02BI stroma (*n* = 3 independent experiments). Pie charts showing the frequency of barcoded cells that expressed full contigs of only endogenous

TCR chains (grey), only 1G4 TCR chains (black), 1G4 TCR and endogenous TRAV chains (pink), 1G4 TCR and endogenous TRBV chains (green), and 1G4 TCR and both endogenous TRAV and TRBV chains (purple). **b,c**, TCR $V_\alpha$ (**b**) and TCR $V_\beta$ (**c**) diversity from mature, conventional SP8 T cells at single-cell resolution. WT PSCs with hDLL4 stroma (grey; *n* = 1 independent experiment, 4,710 cells in total), WT + TCR PSCs with hDLL4 stroma (orange; *n* = 2 independent experiments, 9,619 cells in total), WT + TCR PSCs with hDLL4-A02BI stroma (green; *n* = 3 independent experiments, 12,797 cells in total) and TKO + TCR PSCs with hDLL4-A02BI stroma (red; *n* = 3 independent experiments, 7,930 cells in total).

Fig. 2d,e), these results show that WT + TCR PSCs escape *HLA-A*02:01* restriction of the exogenous 1G4 TCR through RAG-mediated endogenous TCR recombination.

**Functional characterization of TKO PSC-derived SP8 T cells**

To explore the functional consequences of TCR mispairing, in vitro and in vivo assays were performed on WT + TCR and TKO + TCR SP8 T cells isolated from week 6 ATO cultures. Before the functional assays, SP8 T cells were expanded with aAPCs expressing the cognate antigen (NYESO$_{157-165}$). Similar to WT + TCR SP8 T cells, TKO + TCR SP8 T cells showed polyfunctional cytokine production and CD107α mobilization in response to stimulation with aAPCs expressing the cognate antigen

but not an irrelevant (MART1$_{26-35}$) peptide (Fig. 7a). Both WT + TCR and TKO + TCR SP8 T cells upregulated surface CD25 and 4-1BB after overnight stimulation with cognate antigen (Fig. 7b) and underwent antigen-specific proliferation (Fig. 7c).

Despite uniform expression of the transgenic $V_\beta$13.1 TCR chain, ~60% of WT + TCR SP8 cells did not stain with tetramer for the 1G4 TCR, consistent with mispairing of TCRs at the cell surface (Fig. 7d). In contrast, TKO + TCR SP8 T cells retained a 1:1 ratio of 1G4 TCR tetramer to transgenic $V_\beta$13.1 staining (Fig. 7d). TKO + TCR SP8 T cells showed robust loss of surface TCR in response to cognate antigen (consistent with intracellular relocation), whereas 60% of WT + TCR SP8 T cells retained surface TCR (Fig. 7e).

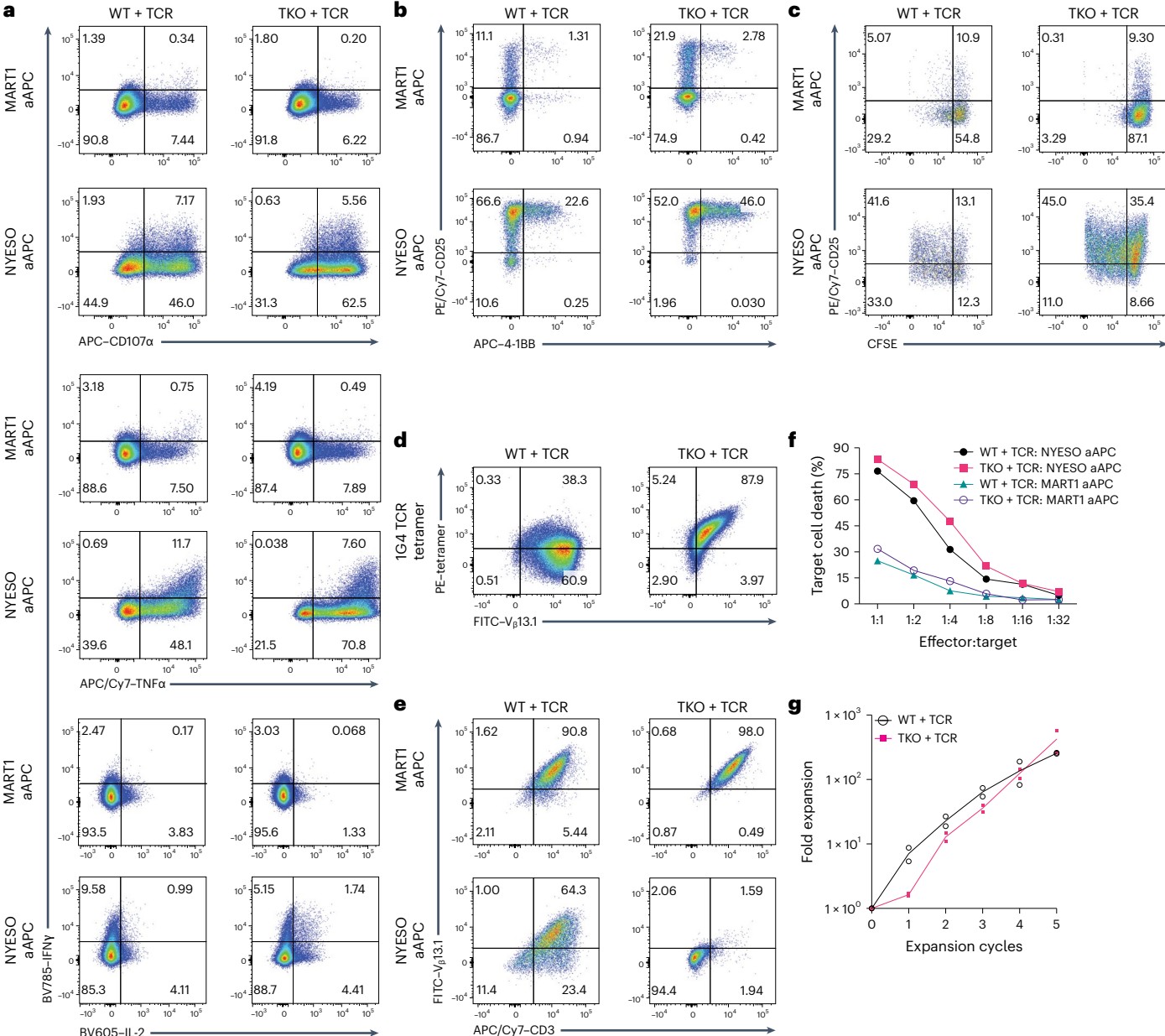

**Fig. 7 | Functional characterization of PSC-derived, antigen-specific T cells in vitro. a–f**, WT + TCR and TKO + TCR (1G4 TCR) SP8 T cells were isolated from week 6 ATOs, both generated with hDLL4-A02BI stroma and expanded with K562 aAPCs expressing the cognate antigen (NYESO), IL-2 and IL-7 before in vitro functional assays. The numbers indicate the percentage of cells within each gate. **a**, Expanded T cells were rested and then stimulated for 6 hours with K562 aAPCs presenting non-specific (MART1) or cognate (NYESO) antigen as an SCT to assay cytokine production and CD107α upregulation (gated on Zombie NIR⁻mStrawberry⁻TCRαβ⁺CD3⁺CD8α⁺ cells). Data are representative of three independent experiments. **b**, Upregulation of activation markers CD25 and 4-1BB on WT + TCR and TKO + TCR SP8 T cells in response to stimulation with MART1 or NYESO aAPCs for 24 hours (gated on DAPI⁻V$_β$13.1⁺CD3⁺CD8α⁺ cells). **c**, Proliferation of SP8 T cells, measured by carboxyfluorescein succinimidyl ester (CFSE), in response to stimulation with MART1 or NYESO aAPCs for 5 days (gated on DAPI⁻mStrawberry⁻CD8α⁺ cells). **d**, Staining with 1G4 tetramer and V$_β$13.1 from 1G4 TCR on SP8 T cells, after 6 days of expansion with NYESO aAPCs (gated on DAPI⁻CD8α⁺ cells). **e**, Cell-surface expression of transgenic 1G4 TCR (measured by V$_β$13.1) and CD3 on SP8 T cells in response to co-culture with MART1 or NYESO aAPCs for 24 hours (gated on DAPI⁻CD8α⁺ cells). **f**, In vitro cytotoxicity of WT + TCR and TKO + TCR SP8 T cells against MART1 or NYESO aAPCs based on Apotracker Green assay at 6 hours of co-culture (data are representative of *n* = 3 independent experiments). **g**, Fold expansion (from initial input) of WT + TCR and TKO + TCR SP8 T cells immediately after isolation from ATOs in response to NYESO aAPCs, IL-2 and IL-7 (data are shown as mean ± s.e.m.; *n* = 2 independent experiments).

Expanded WT + TCR and TKO + TCR T cells also showed potent antigen-specific cytotoxicity in vitro (Fig. 7f). Both WT + TCR and TKO + TCR SP8 T cells were able to undergo robust expansion over input SP8, reaching >250-fold expansion over 5 cycles of antigen-specific activation in the presence of interleukin-7 (IL-7) and interleukin-2 (IL-2) (Fig. 7g).

Expanded TKO + TCR and WT + TCR SP8 T cells were subjected to one freeze–thaw cycle and then further expanded with aAPCs expressing NYESO$_{157–165}$ peptide before testing in an in vivo tumour challenge. Immune-deficient (NOD scid gamma (NSG)) mice were intravenously (I.V.; tail vein) engrafted with luciferase-expressing NALM6 tumour cells engineered to express NYESO$_{157–165}$ peptide as a single-chain trimer

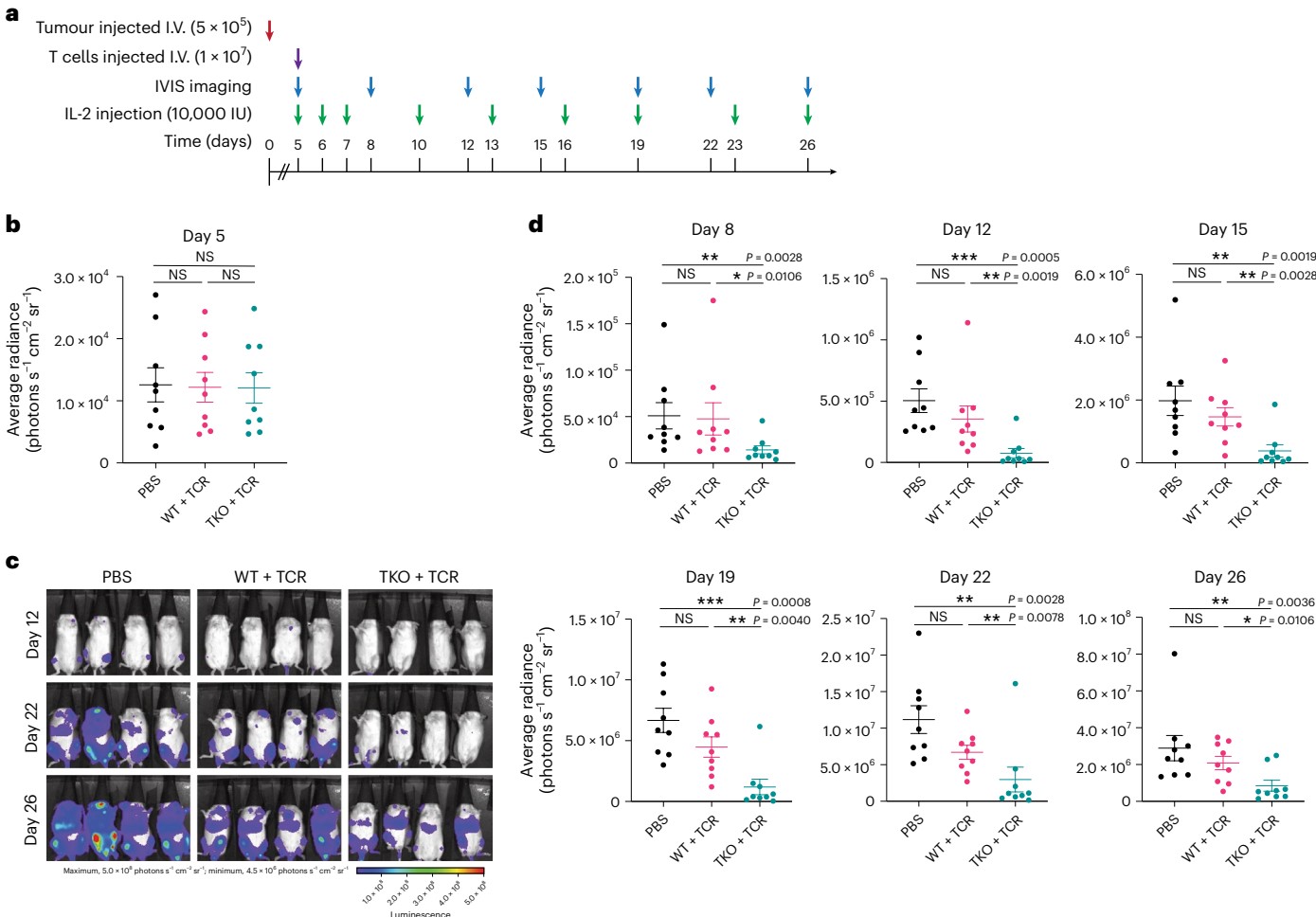

**Fig. 8 | In vivo function of 1G4-expressing, class I MHC-null, _RAG1_- and _RAG2_-null SP8 T cells. a**, Experimental design for in vivo tumour challenge in NSG mice I.V. engrafted with NALM6 tumour cells (5 × 10^5 per mouse) expressing the cognate NYESO SCT and firefly luciferase. Five days after tumour engraftment, mice were injected I.V. with PBS, WT + TCR SP8 T cells (1 × 10^7 per mouse) or TKO + TCR SP8 T cells (1 × 10^7 per mouse). Tumour bioluminescence was measured every 3–4 days; IL-2 (10,000 IU per dose) was administered on the indicated days. **b**, Stratification of mice with equivalent tumour bioluminescence signal at day 5 into treatment groups. The mean ± s.e.m. for each group is shown; two-tailed Mann–Whitney U-test for each group showed NS difference (n = 9 individual mice, all groups). **c**, Representative imaging of mice from each treatment group at the indicated time points. **d**, Summary of bioluminescence tumour signal for all animals in each treatment group at the indicated time points. The mean ± s.e.m. (*P < 0.05, **P < 0.01, ***P < 0.001, two-tailed Mann–Whitney U-test) for each group is shown (n = 9 individual mice, all groups).

(SCT; Fig. 8a). Engraftment of NALM6 cells was confirmed after 5 days by imaging, before mice were stratified into groups with equivalent tumour signal (n = 9 for each group; Fig. 8b). After stratification, SP8 T cells were injected I.V. (retro-orbitally) and monitoring of tumour progression was performed by imaging (Fig. 8a–c). IL-2 (10,000 IU per mouse) was provided intraperitoneally during the course of the experiment (Fig. 8a). At the conclusion of the experiment, TKO + TCR SP8 T cells showed significantly improved tumour control compared with mice that received either no T cells or WT + TCR SP8 T cells (Fig. 8b–d), and they showed improved survival compared with mice that received no T cells (Supplementary Fig. 10e).

## Discussion

The challenges inherent in using an allogeneic source for adoptive T cell therapy take on a new level of complexity in the setting of PSC-derived T cells, where the recapitulation of T cell differentiation requires the same TCR and MHC machinery that must be removed to avoid alloreactivity. Here we report that, by introducing the components required for positive selection into separate cellular sources, it is possible to generate antigen-restricted, mature, conventional T cells from PSCs

in which all endogenous TCR and class I MHC expression has been removed. Specifically, two transgenic class-I-restricted TCRs were expressed in PSCs to circumvent the loss of endogenous TCR rearrangement, and cognate MHC was provided by a supportive stromal line, also engineered to provide other critical external signals during T cell development. Whereas SP8 differentiation efficiency varied based on the TCR, the dependence of in vitro T cell differentiation on the MHC expressed by PSC-derived cells (self-selection) was removed, and the murine stromal cells were used as a surrogate for the thymic epithelial cells that normally provide these critical signals in the thymus.

Production of SP8 T cells from PSCs has been achieved by other groups after _TRAC_ or _RAG_ disruption to prevent expression of endogenous TCRs; all such approaches used agonist or antigen-specific stimulation of cells arrested at the DP stage in monolayer or non-stromal T cell differentiation cultures[24–26]. Although CD8αβ T cells could be generated using such approaches with ATO-derived DPs, the phenotypes were different from CD8αβ T cells spontaneously generated in ATO cultures, both after initial DP stimulation (based on lack of CD62L expression) and further expansion (based on persistent CD45RA expression in DP stimulated and expanded cells). In contrast, the ATO-derived

SP8s initially appeared to be phenotypically closer to mature, naive T cells, and converted to a T effector memory phenotype with further expansion; the functional relevance of these phenotypic differences will require rigorous analysis in future studies.

By deleting both *RAG1* and *RAG2* genes, we avoided the possibility of residual recombination activity that can occur with single gene disruption[25,26,32–34]. Furthermore, the absence of observable defects in function or cytotoxicity with full *RAG* ablation allowed us to manipulate T cell differentiation in the complete absence of endogenous TCRα and TCRβ rearrangement.

Consistent with data from patient samples[17,18], $RAG1^{-/-}RAG2^{-/-}$ PSCs were able to differentiate as far as the DP stage and cell cultures failed early. Although expression of a transgenic TCR produced some mature SP8s from $RAG1^{-/-}RAG2^{-/-}$ PSCs, cell output was low and most cells failed to advance beyond the double-negative stage, suggesting that self-selection is inefficient when it relies on a single TCR. The more robust output of mature T cells from unedited PSCs that were transduced with the same *A\*0201*-restricted 1G4 TCR was likely due to the wider array of MHC interactions provided by mispairing between the 1G4 TCR and the broad repertoire of endogenous TCRs produced in RAG-competent cells. This conclusion is further strengthened by the efficient T cell differentiation even from *A\*0201*[neg] RAG-competent PSCs that expressed the 1G4 TCR.

It was previously shown that TCRα rearrangement occurs from induced PSCs from reprogrammed T cells in which a pre-rearranged TCR exists[25]. The TCR $V_\beta$ rearrangements that we noted to also be present in the RAG1- and RAG2-competent cells showed that allelic exclusion induced by the exogenous 1G4 $V_\beta$ chain was not absolute; however, expression of these low-frequency $V_\beta$ chains was barely, if at all, detectable on the cell surface by flow cytometry, and the functional relevance of potential $V_\beta$ mispairing is unclear[11,12].

Although conventional, mature SP8 T cells generated from TKO + TCR PSCs were transcriptionally and functionally similar to WT + TCR PSC-derived SP8 T cells based on in vitro proliferation and killing, TKO + TCR T cells proved to be significantly superior in controlling tumour growth in vivo. In line with previous studies[25,35–38], our results are consistent with decreased potency of antigen-specific killing due to TCR mispairing, and we found that this effect was particularly important under in vivo conditions; however, we note the ability of such models to predict clinical responses remains to be determined.

In contrast to the poor efficiency of positive selection through self-selection of a single transgenic TCR and intact MHC in the PSCs, a significant improvement in positive selection and cell output was seen when fully human class I MHC (including hB2M) was instead provided by the stromal component of the cultures. It is intriguing to speculate how the MHC–TCR interaction may be quantitively or even qualitatively different when it involves stromal–T, rather than T–T, interaction, but it is conceivable that the mechanosensory stimulation of the TCR is different in each case[39,40].

The murine stromal cell line was further engineered to provide an additional co-factor that could enhance positive selection. ICAM1 has been reported to potentiate TCR activation and antigen sampling through *LFA1* interactions at low pMHC densities[21–23], which is an inherent nature of positive selection when T cells are scanning for low-affinity, 'weak' peptides. It has also been shown that ICAM1–LFA1 signalling leads to cytoskeletal remodelling, increased TCR signal transduction, and both NOTCH-responsive and survival gene expression[41]. Although the direct mechanism of ICAM1 during T cell development was not characterized, its inclusion in the murine MS5 stromal cells significantly increased output of SP8 T cells, while having no observable effects on the ratios of developing T cells. Future studies could focus on delineating the effects of including additional signals and co-factors at the immune synapse similar to those provided by thymic epithelial cells during physiological thymopoiesis. In addition, further studies to understand how these manipulations affect TCR signalling during

positive selection would be of interest, although the analyses will be challenging given the dynamic nature of complex in vitro development systems, such as the ATO, which comprise heterogeneous populations of differentiating cells.

In summary, we have shown that both stem cells and the in vitro microenvironment can be modified to deliver the necessary signals when alloreactive mechanisms are disrupted. To avoid all immune-mediated rejection of allogeneic T cells, additional manipulations of the class I MHC-null T cells to avoid natural killer (NK)-mediated rejection will be required[42–44], along with the deletion of class II MHC. Notably, we have shown that the block to positive selection caused by the removal of class I MHC expression in allogeneic PSCs can be rescued by pairing an exogenous TCR in developing T cells with an engineered murine stromal microenvironment. In addition to providing a cognate MHC, the accessory stromal cells of the ATO system can be modified to deliver other extrinsic signals or co-stimulation factors that improve T cell function and specification, providing an experimental method to dissect T cell differentiation and positive selection in vitro. Owing to their inherent self-renewal capacity, gene-edited PSCs and their stromal counterparts can be readily engineered with additional strategies that will augment T cell function and immune evasion for immunotherapy.

## Methods

### Cell lines

The MS5-hDLL4 cell line was generated in our laboratory as previously described[12]. For clarity, we reproduce the following description from ref. 12. Briefly, MS5 cells[45] were transduced with a lentiviral vector encoding full-length h*DLL4*. The highest 5% of DLL4-expressing cells were isolated by fluorescence-activated cell sorting (FACS) using an anti-hDLL4 antibody and passaged in DMEM with 10% fetal calf serum (FCS). Stable expression was confirmed by flow cytometry for h*DLL4* expression after several weeks of culture, along with quantitative RT–PCR and RNA sequencing.

For the cell lines including class I MHC and scaffolding proteins, the previously derived MS5-hDLL4 line was transduced with varying combinations of individual lentiviruses encoding human *HLA-A\*02:01*, h*B2M* and human *ICAM1* (h*ICAM1*). The highest 5% of transduced cells were isolated by FACS using antibodies detecting *HLA-A\*02:01*, hB2M and hICAM1, and then passaged in DMEM with 10% FCS for expansion and cryopreservation. Stable expression was also confirmed by flow cytometry for hDLL4, HLA-A\*02:01, hB2M and hICAM1.

aAPCs were generated in our laboratory as previously described[12]. For clarity, we reproduce the following description from ref. 12. K562 cells (catalogue number CCL-243; ATCC) were transduced with lentiviral vectors encoding full-length human CD80, CD83, CD137L and HLA-A\*02:01-B2M-NYESO[157–165] SCT (gifts from David Baltimore; Caltech). Target cells for the cytotoxicity assay were created by transduction of K562 with either NYESO1 or MART1 SCTs.

### Lentiviral vector packaging

The full-length coding sequences of h*DLL4*, human *HLA-A\*02:01*, h*B2M* and h*ICAM1* were synthesized (Integrated DNA Technologies) and cloned into third-generation lentiviral vector backbone pCCL-c-MNDU3 (gift from Donald Kohn, University of California, Los Angeles (UCLA)). The mStrawberry fluorescence protein coding sequence was added downstream of *HLA-A\*02:01*, separated by a furin-SGSG-2A linker for polycistronic expression.

The codon-optimized TCR $V_\alpha$ and $V_\beta$ (including $V_\beta$13.1) chains of a TCR specific for *HLA-A\*02:01*/NYESO[157–165] (derived from the 1G4 TCR[20]) were previously described[19] (gift from Antoni Ribas; UCLA). TCR coding sequences and the mTagBFP2 fluorescence protein[46], all separated by furin-SGSG-2A linkers, were subcloned into the third-generation pCCL lentiviral vector downstream of the ubiquitin C promoter with intron 1.

Packaging and concentration of lentivirus particles were performed as previously described[11]. For clarity, we reproduce the following description from ref. [11]. Briefly, 293T cells (catalogue number CRL-3216; ATCC) were co-transfected with the lentiviral vector plasmids pCMV-DR8.9 and pCAGGS-VSVG using TransIT 293T (Mirus Bio) for 17 hours followed by treatment with 20 mM sodium butyrate for 8 hours, followed by generation of cell supernatants in serum-free UltraCulture for 48 hours. Supernatants were concentrated by ultrafiltration using Amicon Ultra-15 100K filters (EMD Millipore) at 4,000$g$ for 40 min at 4 °C and stored as aliquots at −80 °C.

## Human pluripotent cell lines

The human Embryonic Stem Cell Research Oversight Committee and Institutional Review Board have approved all protocols for the use of PSCs for this study. The human embryonic stem cell lines[47] (catalogue number WA01; WiCell) and ESI017 (ref. [48]) (catalogue number ES-700; ESI BIO) were maintained and expanded on Matrigel-coated 6-well plates (Growth Factor Reduced Matrigel Matrix; catalogue number 356231; BD Biosciences) in mTeSR Plus complete medium (mTeSR Plus Basal Medium and mTeSR 5X Supplement; catalogue number 100-0276; Stem Cell Technologies). Culture medium was changed daily. After reaching ~70% confluency, PSC cultures were dissociated with TrypLE Express (catalogue number 12604-013; Thermo Fisher Scientific) and seeded in single-cell suspension at a density of $2 \times 10^5$ cells per well of a Matrigel-coated 6-well plate in mTeSR Plus complete medium and ROCK inhibitor Y-27632 dihydrochloride (10 µM; catalogue number 1254; Tocris Bioscience), which was removed from culture medium after 1 day.

## Design and validation of CRISPR–Cas9 guide RNAs

Using published algorithms found on the Benchling web tool (https://benchling.com), 5 guide RNAs (gRNAs) with optimal in silico-predicted on- and off-target scores (out of 100) were designed to target sequences near the start of *RAG1*, *RAG2* and *B2M*[49–54]. On-target efficiency was assayed in vitro at each target locus by nucleofection of gRNA expressing pX459 plasmid (catalogue number 62988; Zhang Lab, MIT; Addgene) into K562 erythroleukemia cells (catalogue number CCL-243; ATCC).

Genomic DNA was collected from nucleofected cells, and the regions flanking the cleavage sites were amplified by PCR for *RAG1* (693 bp; forward, 5′-TGTATACTGGGACCCTTGGGGAG-3′; reverse, 5′-AGAATCCCACAGATGCGGCAGAG-3′), *RAG2* (865 bp; forward, 5′-TCATCAGTGAGAAGCCTGGCTG-3′; reverse, 5′-GTCA-CGGCTTTTGTAACCTCGG-3′) and *B2M* (644 bp; forward, 5′-TGA-AGTCCTAGAATGAGCGCCC-3′; reverse, 5′-TAAACTTTGTCCC-GACCCTCCC-3′). Products were purified and on-target CRISPR–Cas9 cutting efficiency was determined by Sanger sequencing of the PCR products using the Tracking of Indels by Decomposition (TIDE) tool[55]. The percentage of edited cells was calculated based on the indels produced as a result of double-stranded breaks from CRISPR–Cas9.

For each gene target, the two gRNA candidates with the highest on-target in vitro cutting activity were chosen for off-target cleavage activity in vitro via the genome-wide, unbiased identification of double-stranded breaks enabled by sequencing (GUIDE-seq) method[56]. One gRNA with high on- and low off-target cutting activity was chosen for each target gene to proceed with editing in PSC lines.

## Gene editing of human PSC lines

CRISPR–Cas9 gene editing of PSCs was performed with ribonucleoprotein (RNP) complexes[57] of purified spCas9-NLS (QB3 MacroLab, University of California, Berkeley) and custom-synthesized sgRNAs (Synthego). spCas9-NLS and gRNAs were stored at −80 °C before use for gene editing of PSCs.

Briefly, PSCs were allowed to grow in wells of a Matrigel-coated 6-well plate until reaching ~70% confluency before being dissociated with TrypLE Express (catalogue number 12604-013; Thermo Fisher

Scientific) and resuspended in single-cell solution. Before nucleofection, 84 pmol of each gRNA was complexed with 60 pmol of spCas9, individually, at a ratio of 1 pmol spCas9 to 1.4 pmol gRNA for 15 min at room temperature. PSCs were resuspended at a concentration of $2 \times 10^5$ cells in 14 µl of P4 Primary Cell Nucleofector Solution (P4 Primary Cell 4D-Nucleofector X Kit S, catalogue number V4XP-4032; Lonza).

For sgRNA reactions, fully complexed RNPs (60 pmol spCas9:84 pmol gRNA) were added to resuspended cells and the volume was brought up to 20 µl using P4 Primary Cell Nucleofector Solution (Lonza). For dual gRNA reactions, RNPs were complexed individually and then added (2× 60 pmol spCas9 total) into the cell suspension with a custom-synthesized single-stranded oligonucleotide donor template (ssODN, 100 bp, resuspended at 100 µM; Ultramer DNA Oligo; Integrated DNA Technologies) to a final concentration of 3 µM in solution. Twenty microlitres of combined PSC, RNP and/or ssODN solutions were added into individual wells of the 16-well Nucleocuvette Strip (Lonza). Nucleofection was performed on the 4D-Nucleofector Core and X Unit (catalogue numbers AAF1003B and AAF-1003X; Lonza) using pulse and frequency code CB-150. Cells were allowed to rest in the cuvette for 10 min before transferring into 1.5 ml mTeSR Plus and ROCK inhibitor Y-27632 dihydrochloride (10 µM), and then plated in 1 well of a Matrigel-coated 12-well plate. Culture medium was changed daily, and ROCK inhibitor Y-27632 dihydrochloride (10 µM) was removed from the medium after 48 hours. Edited PSCs were allowed to reach ~70% confluency before being expanded for single-cell cloning and cryopreservation.

KO of *RAG1* and *RAG2* in both H1 and ESI017 parent lines was achieved using a gene-ablation strategy that excised the ~30 kb region on chromosome 11 in which their coding regions were located using both *RAG1* CRISPR number 4 and *RAG2* CRISPR number 14 gRNAs (Supplementary Fig. 1c,d), and a 100 bp ssODN homology-directed repair template (5′-TTTTCATTGTTCTCAGGTACCTCAGCCAGCATG GCAGCCTCTTTCCCAC CCCCTGGTATTGCTGGAGCCTCTC CTGGGGACTTTTGAACAGGTGACCCGA-3′) (Supplementary Fig. 1g). Complete ablation of *RAG1* and *RAG2* genes was confirmed by PCR amplification of a 693 bp sequence within the excised region in PSC clones (*RAG* PCR number 1, forward, 5′-TGTATACTGGGACCCTTGGGGAG-3′; reverse, 5′-AGAATCCCACAGATGCGGCAGAG-3′); proper re-joining of the chromosomal ends was confirmed with PCR amplification of a novel 609 bp product using primers that targeted sequences outside of the ~30 kb region excised by gene editing in PSC clones (*RAG* PCR number 2, forward, 5′-TGTATACTGGGACCCTTGGGGAG-3′; reverse, 5′-GTCACGGCTTTTGTAACCTCGG-3′) (Supplementary Fig. 1g,h). KO of *B2M* in DKO lines was achieved by introducing indels at the beginning of its coding sequence (exon 1 with *B2M* gRNA number 3), and polyclonal KO lines were purified via FACS (Supplementary Fig. 4c–f).

## Single-cell cloning of edited human PSC lines

Single-cell cloning was achieved with low-density plating of expanded, edited PSCs[54]. Briefly, expanded, edited PSCs were dissociated into single-cell solution with TrypLE Express (catalogue number 12604-013; Thermo Fisher Scientific) and then plated in Matrigel-coated 10 cm dishes at a density of $0.5–1 \times 10^4$ cells per plate in mTeSR Plus complete culture medium with ROCK inhibitor Y-27632 dihydrochloride (10 µM). Culture medium was changed daily, and ROCK inhibitor Y-27632 dihydrochloride (10 µM) was removed from the medium after 48 hours. After colony formation, 24–48 individual colonies were scraped with a 200 µl 'P200' pipette tip under a microscope, and then they were transferred into individual wells of a Matrigel-coated 12-well plate with mTeSR Plus culture medium. Once cells reached 60–80% confluency, cells were passaged via scraping for expansion and genotyping PCRs to determine bi-allelic KO of edited genes. Clones with bi-allelic KOs were expanded, cleaned for differentiation, and then genotyped once again before cryopreservation and karyotyping.

## Transduction of human PSC lines

NYESO TCR-transduced PSC lines were generated by transduction of unedited or KO H1 or ESI017 PSCs with either lentiviral vectors encoding the 1G4 TCR (*HLAA\*02:01* restricted recognizing NYESO$_{157-165}$ peptide) or the F5 TCR (*HLA-A\*02:01* restricted recognizing MART1$_{26-35}$ peptide) and the fluorescence marker mTagBFP2. Briefly, PSCs were dissociated into single-cell suspension and plated at a density of $2 \times 10^5$ cells per well of a Matrigel-coated 6-well plate in mTeSR Plus culture medium with ROCK inhibitor Y-27632 dihydrochloride (10 µM). The following day, the culture medium was changed to 1 ml of mTeSR, and concentrated lentiviral supernatant was added directly into the wells. The medium was changed each day until cells reached ~70% confluency, when cells were dissociated with TypLE Express (catalogue number 12604-013; Thermo Fisher Scientific) and purified via FACS sorting using the phenotype TRA1-81$^+$mTagBFP2$^+$. Isolated cells were returned to culture on Matrigel-coated 6-well plates and mTeSR Plus culture medium for expansion and cryopreservation.

## Generation and isolation of human embryonic mesodermal progenitors

Mesodermal commitment was induced as previously described with certain optimizations[12,58,59]. Briefly, PSCs were maintained as single-cell cultures on Matrigel-coated 6-well plates in mTeSR Plus complete medium. PSCs were collected as a single-cell suspension after treatment with TrypLE Express (catalogue number 12604-013; Thermo Fisher Scientific) for 6 min at 37 °C, washed and counted. Cells were resuspended directly in X-VIVO 15 medium (catalogue number 04-418Q; Lonza) supplemented with recombinant human (rh) activin A (10 ng ml$^{-1}$; catalogue number 338-AC-0101; R&D Systems), rhBMP4 (10 ng ml$^{-1}$; catalogue number 314-BP-010; R&D Systems), rhVEGF (10 ng ml$^{-1}$; catalogue number 298-VS-005; R&D Systems), rhFGF (10 ng ml$^{-1}$; catalogue number 233-FB-025; R&D Systems) and ROCK inhibitor Y-27632 dihydrochloride (10 µM; catalogue number 1254; Tocris Bioscience). Cells were plated on Matrigel-coated 6-well plates at $3.3 \times 10^6$ cells per well in 3 ml. The medium was then changed daily with X-VIVO 15 supplemented with rhBMP4 (10 ng ml$^{-1}$), rhVEGF (10 ng ml$^{-1}$) and rhFGF (10 ng ml$^{-1}$). At day 3.5, cells were washed 3 times with PBS and incubated with Accutase (catalogue number AT-104; Innovative Cell Technologies): 1 ml per well for 10 min at 37 °C. Cells were collected by dilution with magnetic-activated cell sorting (MACS) buffer (PBS, 0.5% bovine serum albumin and 2 mM EDTA) followed by depletion of CD326$^+$ (EpCAM) cells by MACS using CD326 (EpCAM) MicroBeads (catalogue number 130-061-101; Miltenyi). In addition, embryonic mesodermal progenitors (EMPs) derived from *RAG1$^{-/-}$ RAG2$^{-/-}$ B2M$^{-/-}$* TKO PSCs were stained with PE-conjugated hB2M antibody (BioLegend) and also depleted of B2M$^+$ cells using MACS Anti-PE MicroBeads (catalogue number 130-048-801; Miltenyi) in addition to CD326 MicroBeads.

## Human PSC-derived embryonic mesodermal organoid and ATO cultures

Embryonic mesodermal organoids (EMOs) and ATOs were generated as previously described[12]. For clarity, we reproduce the following description from ref. 12. Briefly, sequential generation of EMOs and then ATOs was accomplished in three-dimensional aggregates through changing media (Fig. 1a and Supplementary Fig. 2a). EMOs were established by aggregating purified EMPs with MS5-hDLL4 cells. MS5-hDLL4 cells were collected by trypsinization and resuspended in 'haematopoietic induction media' comprising EGM2 (catalogue number CC-4176; Lonza) supplemented with ROCK inhibitor Y-27632 dihydrochloride (10 µM; catalogue number 1254; Tocris Bioscience) and TGFβRI inhibitor SB-431542 (10 µM; SB Blocker, catalogue number 1614; Tocris Bioscience). At day −14, $5 \times 10^5$ MS5-hDLL4 cells were combined with $5 \times 10^4$ (H1) or $1 \times 10^5$ (ESI017) purified EMPs per ATO in

1.5 ml Eppendorf tubes and centrifuged at 300$g$ for 5 min at 4 °C in a swinging bucket centrifuge.

Multiple (up to 180) EMOs were prepared per tube. Supernatants were carefully removed and the cell pellet was briefly vortexed and resuspended in haematopoietic induction medium at a volume of 5 µl per EMO. Three EMOs were individually plated (5 µl per EMO) on a 0.4 mm Millicell transwell insert (catalogue number PIMC0R5G50; EMD Millipore) and then placed in a 6-well plate containing 1 ml of haematopoietic induction medium per well. The medium was changed every 2–3 days for 1 week with medium composed of EGM2 with SB Blocker (10 mM). At day −7, the medium was changed to EGM2 + SB Blocker (10 mM) with the haematopoietic cytokines, 5 ng ml$^{-1}$ rhTPO (catalogue number 288-TPN-025; R&D Systems), 5 ng ml$^{-1}$ rhFLT3L (catalogue number 308-FK-025; R&D Systems) and 50 ng ml$^{-1}$ rhSCF (catalogue number 255-SC-200; R&D Systems). This medium was changed every 2–3 days for an additional 7 days. At day 0, ATOs were initiated by simply changing to ATO culture medium (RB27) composed of RPMI 1640 (catalogue number 10-040-CV; Corning), 4% B27 supplement (catalogue number 17504044; Thermo Fisher Scientific), 30 mM L-ascorbic acid 2-phosphate sesquimagnesium salt hydrate (catalogue number A8960-5G; Sigma-Aldrich) reconstituted in PBS, 1% penicillin-streptomycin (catalogue number 15140122; Thermo Fisher Scientific), 1% Glutamax (catalogue number 35050061; Thermo Fisher Scientific), 10 ng ml$^{-1}$ rhSCF (R&D Systems), 5 ng ml$^{-1}$ rhFLT3L (R&D Systems) and 5 ng ml$^{-1}$ rhIL-7 (catalogue number 207-IL-200; R&D Systems). The medium was completely changed every 3–4 days.

For experiments involving the use of engineered stromal lines for ATO T cell differentiation (MS5-hDLL4-A\*0201, MS5-hDLL4-A\*0201-hB2M and MS5-hDLL4-A\*0201-hB2M-ICAM), EMOs were collected in bulk after 14 days (day 0) by adding 1 ml MACS buffer (PBS, 0.5% bovine serum albumin and 2 mM EDTA) to each filter, briefly disaggregating the ATO by scraping with a 1 ml 'P1000' pipette and then passed through a 50 µm nylon strainer. Collected cells were counted, and then $2 \times 10^4$ cells were combined with $2.5 \times 10^5$ engineered MS5-hDLL4 stromal cell lines per ATO in 1.5 ml Eppendorf tubes. Cell mixtures were centrifuged at 300$g$ for 5 min at 4 °C in a swinging bucket centrifuge before the formation of 3 ATO aggregates (5 µl per aggregate) per 0.4 mm Millicell transwell insert (catalogue number PIMC0R5G50; EMD Millipore), similarly to the EMO aggregate formation described above.

## Isolation of ATO-derived SP8 and DP precursor cells

ATOs were collected by adding MACS buffer (PBS, 0.5% bovine serum albumin and 2 mM EDTA) to each filter, briefly disaggregating the ATO by pipetting with a 1 ml P1000 pipette, and then they were passed through a 50 µm nylon strainer; SP8 T cells were collected after 6 weeks of T cell differentiation, and DP precursors were collected after 3 weeks. For bulk-scale collection of ATOs, aggregates were collected in a similar fashion; however, up to 150 aggregates were collected in a well filled with MACS buffer and then they were transferred onto a 50 µm nylon filter, where aggregates were physically dissociated on top of the filter using the back end of a sterile 1 ml syringe. After dissociation, the filter was washed with MACS buffer and cell mixtures were centrifuged at 300$g$ for 5 min at 4 °C in a swinging bucket centrifuge.

## Stimulation and expansion of DP precursors

ATO-derived DP precursors were isolated as described above and seeded at an initial density of $5 \times 10^5$ cells per well of a 48-well plate with 1 ml of specific culture medium described below. The activation and expansion of DP precursors using OKT3 antibody followed previously published protocols[25,26]. DP precursors were isolated as described above and stimulated with 500 ng ml$^{-1}$ Ultra-LEAF Purified anti-human CD3 antibody (clone OKT3; BioLegend) and activated through CD3 engagement with Ultra-LEAF Purified anti-human CD3 antibody (catalogue number 317326; BioLegend) in MEMα (catalogue

number 12561049; Thermo Fisher Scientific) supplemented with 15% FCS, 1% Glutamax (catalogue number 35050061; Thermo Fisher Scientific), 1% penicillin-streptomycin (catalogue number 15140122; Thermo Fisher Scientific), 1% Insulin-Transferrin-Selenium solution (catalogue number 41400045; Thermo Fisher Scientific) and 50 µg ml$^{-1}$ L-ascorbic acid 2-phosphate sesquimagnesium salt hydrate (Sigma-Aldrich) in the presence of 10 ng ml$^{-1}$ rhIL-7 (R&D Systems) and 10 nM dexamethasone (catalogue number D4902; Sigma-Aldrich) for 3 days before anti-human CD3 antibody and dexamethasone were removed. Expansion of T cells was carried out after 7 days for an additional 14 days (21 days total) in MEMα (catalogue number 12561049; Thermo Fisher Scientific) supplemented with 15% FCS, 1% Glutamax (catalogue number 35050061; Thermo Fisher Scientific), 1% penicillin-streptomycin (catalogue number 15140122; Thermo Fisher Scientific), 1% Insulin-Transferrin-Selenium solution (catalogue number 41400045; Thermo Fisher Scientific) and 50 µg ml$^{-1}$ L-ascorbic acid 2-phosphate sesquimagnesium salt hydrate (Sigma-Aldrich) in the presence of 10 ng ml$^{-1}$ rhIL-7 (R&D Systems), 5 ng ml$^{-1}$ rhIL-15 (catalogue number 247-ILB-025; R&D Systems) and 2 µg ml$^{-1}$ phytohaemagglutinin (catalogue number L1668-5MG; Sigma-Aldrich). For both activation and expansion stages, fresh medium was added every 2–3 days, and cells were replated into larger wells as necessary.

Activation and expansion of DP precursors was also performed using Dynabeads Human T-Activator CD3/CD28 (catalogue number 11131D; Thermo Fisher Scientific) according to the manufacturer's protocol. Briefly, beads were added in a 1:1 ratio of T cells to beads in AIM V (catalogue number 12055083; Thermo Fisher Scientific) supplemented with 5% human AB serum (catalogue number 100-512; Gemini Bio-Products), 20 ng ml$^{-1}$ rhIL-2 (catalogue number 200-02; Peprotech) and 5 ng ml$^{-1}$ rhIL-7 (R&D Systems). Beads were replenished every 7 days, fresh medium was added every 2–3 days and cells were replated into larger wells as necessary.

Activation and expansion of DP precursors was also performed with irradiated antigen-expressing K562 aAPCs (K562 CD80-CD83-CD137L expressing either HLA-A*02:01-B2M-NYESO$_{157–165}$ SCT or HLA-A*02:01-B2M-MART1$_{26–35}$ SCT) in a 1:3 aAPC:T cell ratio in AIM V (Thermo Fisher Scientific) supplemented with 5% human AB serum (Gemini Bio-Products), 20 ng ml$^{-1}$ rhIL-2 (Peprotech) and 5 ng ml$^{-1}$ rhIL-7 (R&D Systems). Irradiated aAPCs were added every 7 days in a 1:3 aAPC:T cell ratio, fresh medium was added every 2–3 days and cells were replated into larger wells as necessary.

### Single-cell RNA library preparation and sequencing

ATO-derived T cells were isolated as described above and then FACS sorted for ATO-derived SP8 T cells from WT (mCD29$^-$CD45$^+$CD8α$^+$CD4$^-$) and 1G4 TCR-transduced (mCD29$^-$mTagBFP2$^+$CD45$^+$CD8α$^+$CD4$^-$) PSC ATOs into PBS + 0.04% bovine serum albumin. Cells were counted and resuspended at a concentration of ~1,000 cells per µl and provided to the Technology Center for Genomics and Bioinformatics (TCGB) core for unique molecular identifier (UMI) tagging and generation of gene expression (GEX) and human TCR repertoire (V(D)J) libraries using the 10X Chromium Next GEM Single Cell V(D)J Reagent Kit v.1.1 (10X Genomics). Similarly, PB SP8s were FACS isolated (CD45$^+$CD8α$^+$CD8β$^+$CD4$^-$TCRαβ$^+$CD3$^+$) from PB mononuclear cells obtained from healthy donors, and provided to the TCGB core for UMI tagging and library generation. Fully constructed libraries for all samples were run in one S4 flow cell on the Illumina NovaSeq Platform.

### GEX and TCR reference genomes

The reference genomes used for GEX and TCR alignment, both of which correspond to the GRCh38 genome, were downloaded from the 10X Genomics website. To detect the exogenous TCR, which was expressed using a codon-optimized sequence, the GRCh38 reference genome was customized using the Cell Ranger v.7.0.0 (10X Genomics) 'mkref' pipeline.

### scRNA-seq data filtration and cleaning

Sequenced reads from each sample were aligned to the human reference genome GRCh38 and filtered for empty droplets using the Cell Ranger v.7.0.0 (10X Genomics) 'multi' pipeline that generated count matrices from the GEX libraries and assembled full TCR contigs from the V(D)J libraries. On average, we achieved >30,000 mean reads per cell for GEX expression libraries and >7,000 mean reads per cell for V(D)J libraries.

For samples that were acquired for this study, GEX (RNA) count matrices were loaded with Seurat v.4.3.0 (Satija Lab)[60]. As V(D)J libraries were also generated from the same complementary DNA, TCR genes aligned by whole transcriptomic sequencing were removed to prevent clustering bias from samples with intact germline TCR recombination. Count matrices were loaded separately and barcoded cells were filtered for cells with outlier UMI counts (low-quality cells and doublets), high mitochondrial gene expression (due to cellular stress or loss of cytoplasmic RNA) and low number of sequenced genes. After initial data filtration, individual datasets were bioinformatically cleaned for doublets using DoubletFinder v.2.0.3 (ref. 61), and cells were cleaned for stochastic dropouts of identity genes based on their sorted SP8 phenotype (CD8α, CD8β, CD3 > 0 and CD4 < 0). Cleaned cells were then batch corrected for technical and biological variations using the reciprocal principal component analysis (RPCA) integration method in Seurat. For integration of the combined ATO-derived SP8 T cell samples, molecular count data for each sample were individually normalized and variance stabilized using SCTransform in Seurat, and then cell cycle phase scores were calculated for each individual sample based on the expression of canonical cell cycle genes within a barcoded cell. Following cell cycle scoring, raw counts were normalized and variance was stabilized again using SCTransform in Seurat with the additional steps of regressing calculated cell cycle scores and mitochondrial genes to mitigate the effects of cell cycle heterogeneity and mitochondrial gene expression.

To perform RPCA integration of ATO-derived SP8 samples, highly variable genes (nfeatures = 5,000) were identified and then used to find integration anchors between datasets. Clustering of the fully integrated object of ATO-derived SP8 samples was performed using the IKAP (Identifying K mAjor cell Population group in scRNA-seq analysis) package[62]. IKAP analysis identified the optimal principal components and clusters for the dataset. Of the three clusters identified, one cluster was removed based on high mitochondrial gene expression, indicating stressed cells or loss of cytoplasmic RNA, and the other two were identified as immature (CD45RO$^+$) and mature, naive (CD45RA$^+$) SP8 T cells based on differentially expressed genes. Mature, naive CD45RA$^+$ SP8 T cells were used for downstream analyses.

For publicly available datasets from human thymic SP8 (GSE148981; ref. 27), PB NK and CD14$^+$ monocyte (PB monocytes) cells (10X Genomics[28,29]), raw sequencing reads were downloaded from their respective repositories and then aligned to the human reference genome GRCh38 using the Cell Ranger v.7.0.0 (10X Genomics) 'count' pipeline, similarly to method described above, which only generated count matrices from the GEX reads. Individual samples were loaded, filtered and cleaned using Seurat v.4.3.0 (Satija Lab), similarly to that described above. Total PB samples were SCTransform normalized in Seurat and variance stabilized before integration, and principal components were identified from the integrated object. Then, dimensional reduction (UMAP) and cell clustering was performed on the integrated object using the functions included in the Seurat package (Satija Lab). PB NK and CD14$^+$ monocytes (PB monocytes) and PB SP8 T cells were identified and subset out of the integrated object for downstream analyses.

### Analysis and visualization of scRNA-seq data

Cleaned samples were merged into a single Seurat (Satija Lab) object, preserving only the raw RNA count matrices for each object.

Each sample was then normalized and variance stabilized using SCTransform in Seurat before regressing cell cycle phase scores and mitochondrial genes, as described above, before remerging the list back into a single object. Given the individual SCTransform models calculated from each sample, the minimum median UMI was used to recorrect the counts and data slot for downstream analysis using PrepSCTFindMarkers in Seurat (Satija Lab). Scaled expression values of highly variable genes, which were identified as the union of the top 5,000 genes from each individual sample during SCTransform in Seurat, were used to calculate principal component analysis for dimensionality reduction using the UMAP function in Seurat (Satija Lab). Individual cell populations (SP8, NK and monocytes) clustered together, and minor outliers were manually cleaned for gene expression visualization using the DotPlot function in Seurat (Satija Lab).

To perform global gene expression analysis between individual samples and their respective sources, a pseudobulk approach was taken by extracting the raw counts and metadata from the object, described above, and creating an object using the SingleCellExperiment package[63]. Individual samples were assigned identities based on cell type identity and source to create a DEseq2 (ref. 64) object in R, and pseudobulked counts were normalized using the regularized-logarithm transformation (rlog) function. Pearson's correlations of global gene expression for all pairwise combinations between each sample were calculated from the rlog-normalized data using the cor() function in R v.4.2.1 (ref. 65). Dendrogram and hierarchical clustering analysis were visualized using the pheatmap package[66].

For the purposes of analysing SP8 T cells without the addition of PB monocytes and PB NK cells, SP8 T cells were subset out of the object and each individual sample was normalized using SCTransform in Seurat and integrated using the RPCA integration method in Seurat with nfeatures = 5,000 highly variable genes. Clustering of the fully integrated object of ATO-derived SP8 samples was performed with the IKAP package[62], which identified the optimal principal components and clusters for UMAP dimensionality reduction of the integrated object. Dendrogram, hierarchical clustering and global gene expression analyses were performed as described above.

### TCR repertoire analysis by scRNA-seq

Full, endogenous TCR $V_\alpha$ and $V_\beta$ contigs were aligned and assembled using the Cell Ranger v.7.0.0 (10X Genomics) multi pipeline, as described above. As the exogenous TCR was expressed using a codon-optimized sequence, reads from the 5′ GEX libraries were aligned using a custom human genome reference (GRCh38), described above, which included codon-optimized sequence for the 1G4 TCR. After full reconstruction of TCRs and gene alignment, Cell Ranger output matrices were filtered using Seurat v.4.3.0 (Satija Lab), as described above. For samples including the exogenous 1G4 TCR, barcoded cells were also extracted for cells that expressed the exogenous transgene. Finally, the remaining barcoded cells were exported as a list and used to filter V(D)J sequencing outputs in R for TCR diversity calculations, using the number of barcoded cells remaining after quality control as a denominator, and calculations of percentages of cells expressing endogenous and/or exogenous TCR.

### Expansion of ATO-derived T cells for functional assays

ATO-derived SP8 T cells were isolated as described above and expanded in vitro using irradiated antigen-expressing K562 aAPCs (K562 CD80-CD83-CD137L and HLA-A*02:01-B2M-NYESO$_{157-165}$ SCT) in a 1:3 aAPC:T cell ratio in AIM V (Thermo Fisher Scientific) supplemented with 5% human AB serum (Gemini Bio-Products), 20 ng ml$^{-1}$ rhIL-2 (Peprotech) and 5 ng ml$^{-1}$ rhIL-7 (R&D Systems). Fresh medium was added every 2–3 days, and cells were replated into larger wells as necessary. Restimulations for proliferation assays were performed every 5 days. Before functional and proliferation assays, expanded SP8 T cells were rested for an additional 2 days after the previous expansion cycle

(7 days from aAPC stimulation) with reduced cytokines, 5 ng ml$^{-1}$ rhIL-2 (Peprotech) only, on the night before the assays.

### CFSE proliferation assays

As described above, SP8 T cells were expanded and rested before performing proliferation assays. Briefly, $5 \times 10^5$ were labelled with CFSE (BioLegend) at a final concentration of 5 µM and co-cultured with irradiated K562 aAPCs expressing cognate (NYESO, HLA-A*02:01-B2M-NYESO$_{157-165}$) or irrelevant (MART1, HLA-A*02:01-B2M-MART1$_{26-35}$) SCTs at a 1:1 T cell:aAPC ratio in a 24-well plate and 3 ml AIM V (Thermo Fisher Scientific) with 5% human AB serum (Gemini Bio-products) and 20 ng ml$^{-1}$ rhIL-2 (Peprotech)[12]. On day 5, cells were washed and stained for CD25 and 4-1BB (BioLegend) and analysed by flow cytometry.

### T cell cytokine assays

Expanded SP8 T cells from ATOs were rested as described above. A total of $2 \times 10^5$ rested ATO-derived SP8 T cells were co-cultured with K562 aAPCs expressing cognate (NYESO) or irrelevant (MART1) SCTs at a 2:1 T cell:aAPC ratio in 96-well U-bottom plates and 200 µl AIM V (Thermo Fisher Scientific) with 5% human AB serum (Gemini Bio-products) and protein transport inhibitor cocktail (catalogue number 00-4980-03; eBioscience) for 6 hours. After 4 hours, CD107α-APC antibody (BioLegend) was added to wells at a 1:50 final dilution. Cells were washed and stained for CD3, CD4 and CD8α (BioLegend) and Zombie NIR Fixable Viability Dye (BioLegend) before fixation and permeabilization with an intracellular staining buffer kit (catalogue number 88-8824-00; eBioscience) and intracellular staining with antibodies against IFNγ, TNFα and IL-2 (BioLegend).

### In vitro cytotoxicity assays

Expanded SP8 T cells from ATOs were rested as described above. For cytotoxicity assays, 2-fold serial dilutions of rested SP8 T cells were performed in 96-well U-bottom plates starting at $1 \times 10^5$ cells in 200 µl AIM V (Thermo Fisher Scientific) with 5% human AB serum (Gemini Bio-products). K562 aAPCs expressing cognate (NYESO) or irrelevant (MART1) SCTs were plated at $1 \times 10^5$ target cells per well. Apoptotic cell death of target cells was quantified by Apotracker Green (BioLegend) and DAPI staining at 6 hours. Target cell death was calculated by subtracting the percentage of Apotracker Green-positive target cells in wells receiving no T cells from wells that received T cells.

### In vivo tumour assay

All animal experiments were conducted under an ethics protocol approved by the UCLA Chancellor's Animal Research Committee (protocol number ARC-2008-175). Female 6- to 8-week-old NOD. Cg-Prkdcscid Il2rgtm1Wjl/SzJ (NSG) mice (strain 005557, RRID IMSR_JAX:005557; Jackson Laboratory) were I.V. injected (tail vein) with $5 \times 10^5$ NALM6 target cells transduced with the HLA-A*02:01-B2M-NYESO$_{157-165}$ SCT and firefly luciferase. In vivo imaging was performed by intraperitoneal injection of luciferin (IVIS spectrum; Perkin Elmer) 5 days after tumour injection. Groups of $n = 9$ mice were randomized based on average luciferase signal activity. ATO-derived SP8 T cells expressing NYESO TCR ($1 \times 10^7$ cells per mouse) were I.V. injected (retro-orbitally) after mice were grouped on day 5; control mice received PBS injections. Mice were dosed intraperitoneally with 10,000 IU rhIL-2 (catalogue number 202-IL-500; R&D Systems) on days 5–7 post-tumour injection, and then every 3 days until euthanasia was humane and necessary, as determined by our ethics committee. Tumour bioluminescence was repeated twice a week (every 3–4 days) for at least 26 days, and mice were euthanized based on disease burden criteria (Fig. 7a). Luminescence images were analysed using Living Image software v.4.7.4 (Perkin Elmer).

### Flow cytometry and antibodies

Staining for flow cytometry was performed in PBS with 0.5% bovine serum albumin and 2 mM EDTA for 15 min at 4 °C in the dark. TruStain FcX

(catalogue number 422302; BioLegend) was added to all samples for 5 min before antibody staining. Tetramer staining was performed with the PE-conjugated HLA-A*02:01/NYESO[157-165] tetramer (catalogue number TB-M011-1; MBL International) or PE-conjugated HLA-A*02:01/MART1[26-35] tetramer (catalogue number TB-0009-1; MBL International) at a 1:50 final dilution at room temperature for 20 min before additional antibody staining for 15 min at 4 °C. DAPI was added to all samples before analysis.

Analysis was performed on a BD LSRII Fortessa, and FACS sorting was performed on BD FACSAria or BD FACSAria-H instruments (BD Biosciences) at the UCLA Broad Stem Cell Research Center Flow Cytometry Core. BD instruments used BDFACSDiva software v.8.0.2 (BD Biosciences). For all analyses (except intracellular staining), DAPI-positive cells were gated out, and doublets were removed through forward scatter height versus forward scatter width and side scatter height versus side scatter width gating. The following anti-human antibody clones for surface and intracellular staining were obtained from BioLegend: CD3 (UCHT1), CD4 (RPA-T4), CD5 (UCHT2), CD7 (CD7-6B7), CD8α (SK1), CD25 (BC96), CD27 (O323), CD28 (CD28.2), CD326 (EpCAM; clone 9C4), CD34 (581), CD45 (HI30), CD45RA (HI100), CD45RO (UCHL1), CD56 (HCD56), CD62L (DREG-56), CD107α (H4A3), B2M (2M2), CCR7 (G043H7), HLA-ABC (W6/32), IFNγ (4S.B3), IL-2 (MQ1-17H12), TCRαβ (IP26), TNFα (MAb11), V$_β$13.1 (H131) and 4-1BB (clone 4B4-1). The anti-human antibody clone CD8β (clone REA715) was obtained from Miltenyi. Anti-mouse CD29 (clone HMb1-1) was obtained from BioLegend. Flow cytometry data were analysed with FlowJo software (Tree Star).

## Statistics

In all figures, exact *n* values represent independent experiments and are specified in the figure legends, and mean ± s.e.m. values are shown. Statistics were analysed using GraphPad Prism software and *P* values were calculated using the two-tailed unpaired *t*-test, two-tailed Mann–Whitney *U*-test and the log-rank test. The *P* values are indicated directly on the figure within the corresponding graphs; *$P < 0.05$, **$P < 0.01$ and ***$P < 0.001$ were considered statistically significant.

## Reporting summary

Further information on research design is available in the Nature Portfolio Reporting Summary linked to this article.

## Data availability

The main data supporting the results in this study are available within the paper and its Supplementary Information. The raw and analysed datasets generated during this study for gene expression and TCR-encoding genes are available from the NCBI Gene Expression Omnibus (GSE237780). The public datasets used for this study can be found at the NCBI Gene Expression Omnibus for the human thymic SP8, GSE148981 (ref. 27), and at 10X Genomics[28,29] (https://www.10xgenomics.com/resources/datasets/8-k-pbm-cs-from-a-healthy-donor-2-standard-2-1-0 and https://www.10xgenomics.com/resources/datasets/human-pbmc-from-a-healthy-donor-10-k-cells-v-2-2-standard-5-0-0) for PB NK cells and CD14+ monocytes. The raw and analysed datasets generated during the study are available for research purposes from the corresponding author on reasonable request. Source data are provided with this paper.

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

## Acknowledgements

We thank J. Scholes, F. Codrea and J. Calimlim from the UCLA Broad Stem Cell Research Center (BSCRC) Flow Cytometry core for their assistance with FACS sorting, the UCLA TCGB core for high-throughput sequencing; D. B. Kohn and C. Kuo for their advice regarding gene-editing techniques; L. Yang and Y. Chen for their advice on T cell biology and immunotherapy; A. Pyle, H. Xi and C. Young for their collective experience with generating clonal PSC KO lines; and S. C. de Barros for assisting with illustrations. This research was made possible with support from NIH/NCI F31CA239555 (P.C.C.), NIH/NHLBI T32HL086345 (P.C.C), UCLA BSCRC fellowships (X.Y., S.P.Y. and C.S.S.), NIH K08CA235525 (C.S.S.), the V Foundation Scholar Award (C.S.S.) and the California Institute of Regenerative Medicine (CIRM) DISC2-10134 (G.M.C.). The contents of this publication are solely the responsibility of the authors and do not represent the official views of CIRM or any other agency of the state of California. The funders had no role in study design, data collection and analysis, decision to publish or preparation of the manuscript.

## Author contributions

P.C.C. conceptualized and performed experiments, curated and analysed data, prepared figures, and wrote and edited the article. X.Y. assisted with the in vivo experiments and edited the article. A.Z. assisted with the in vivo experiments. C.T. assisted with gene editing, with the cloning of stem cell lines and with the stem cell and ATO cultures. C.R.R. assisted with data processing and analysis. S.P.Y., C.E., S.T. and Y.Z. assisted with stem cell and ATO cultures. S.L. maintained the cell line cultures. A.M.-H. provided conceptual advice and guidance. C.S.S. provided critical materials, conceptual advice and guidance. G.M.C. directed the project and co-wrote and edited the article.

## Competing interests

Pluto Immunotherapeutics is supporting further preclinical research on the ATO system at UCLA through a sponsored research agreement with G.M.C. as principal investigator. A.M.-H., G.M.C. and C.S.S. are co-founders of Pluto Immunotherapeutics, which holds certain rights relating to the ATO system. G.M.C. and C.S.S. are consultants to, and A.M.-H. is a current employee of, Pluto Immunotherapeutics. The other authors declare no competing interests.

## Additional information

**Correspondence and requests for materials** should be addressed to Gay M. Crooks.

# Reporting Summary

## Statistics

For all statistical analyses, confirm that the following items are present in the figure legend, table legend, main text, or Methods section.

| n/a | Confirmed | |
|---|---|---|
| ☐ | ☒ | The exact sample size (*n*) for each experimental group/condition, given as a discrete number and unit of measurement |
| ☐ | ☒ | A statement on whether measurements were taken from distinct samples or whether the same sample was measured repeatedly |
| ☐ | ☒ | The statistical test(s) used AND whether they are one- or two-sided <br> *Only common tests should be described solely by name; describe more complex techniques in the Methods section.* |
| ☒ | ☐ | A description of all covariates tested |
| ☒ | ☐ | A description of any assumptions or corrections, such as tests of normality and adjustment for multiple comparisons |
| ☐ | ☒ | A full description of the statistical parameters including central tendency (e.g. means) or other basic estimates (e.g. regression coefficient) AND variation (e.g. standard deviation) or associated estimates of uncertainty (e.g. confidence intervals) |
| ☐ | ☒ | For null hypothesis testing, the test statistic (e.g. *F*, *t*, *r*) with confidence intervals, effect sizes, degrees of freedom and *P* value noted <br> *Give P values as exact values whenever suitable.* |
| ☒ | ☐ | For Bayesian analysis, information on the choice of priors and Markov chain Monte Carlo settings |
| ☐ | ☒ | For hierarchical and complex designs, identification of the appropriate level for tests and full reporting of outcomes |
| ☐ | ☒ | Estimates of effect sizes (e.g. Cohen's *d*, Pearson's *r*), indicating how they were calculated |

*Our web collection on statistics for biologists contains articles on many of the points above.*

## Software and code

Policy information about availability of computer code

| Data collection | BDFACSDiva software v8.0.2 was used for flow cytometry data collection. <br> Living Image version 4.7.4 was used for the measurement and analysis of in vivo luciferase activity. |
|---|---|
| Data analysis | FlowJo v10.8.1 was used for the analysis of flow-cytometry data. <br> Benchling was used for sgRNA and vector designs (www.benchling.com). <br> GUIDEseq v1.0.1 was used for off-target sgRNA analysis (Tsai Lab). <br> GraphPad Prism v9.4.1 was used for graphical representations and statistical analysis. <br> CellRanger v7.0.0 from 10X Genomics was used for aligning raw TCR-sequencing reads. <br> Seurat v4.3.0 from the Satija Lab was run on RStudio v2022.07.1+554 with R v4.2.1 were used for filtering, cleaning and identifying single cells from gene expression and TCR sequencing data from CellRanger outputs. The codes are available on request. <br> DoubletFinder v2.0.3 from McGinnis et al. 2019, for identifying doublets, bioinformatically. <br> IKAP from Chen et al. 2019, for unsupervised clustering of ATO-derived T cells. <br> SingleCellExperiment from Amezquita et al. 2020, for extracting counts for pseudobulk analysis. <br> DEseq2 from Love et al. 2014, for performing normalizing pseudobulk data. <br> pheatmap from Kolde. R, 2019, for visualizing normalized Pearson's correlation data. <br> Living Image version 4.7.4 was used for the analysis of in vivo luciferase activity. |

For manuscripts utilizing custom algorithms or software that are central to the research but not yet described in published literature, software must be made available to editors and reviewers. We strongly encourage code deposition in a community repository (e.g. GitHub). See the Nature Portfolio guidelines for submitting code & software for further information.

## Data

Policy information about availability of data

All manuscripts must include a data availability statement. This statement should provide the following information, where applicable:
- Accession codes, unique identifiers, or web links for publicly available datasets
- A description of any restrictions on data availability
- For clinical datasets or third party data, please ensure that the statement adheres to our policy

The main data supporting the results in this study are available within the paper and its Supplementary information. Source data for the figures are provided with this paper. The raw and analysed datasets generated during this study for gene expression and TCR-encoding genes are available from the NCBI Gene Expression Omnibus (GSE237780). The public datasets used for this study can be found at the NCBI Gene Expression Omnibus for the human thymic SP8, GSE14898127, and at 10X Genomics28,29 (https://www.10xgenomics.com/resources/datasets/8-k-pbm-cs-from-a-healthy-donor-2-standard-2-1-0, https://www.10xgenomics.com/resources/datasets/human-pbmc-from-a-healthy-donor-10-k-cells-v-2-2-standard-5-0-0) for peripheral blood natural killer cells and CD14+ monocytes. The raw and analysed datasets generated during the study are available for research purposes from the corresponding author on reasonable request.

## Human research participants

Policy information about studies involving human research participants and Sex and Gender in Research.

| | |
|---|---|
| Reporting on sex and gender | The study did not involve human research participants. |
| Population characteristics | — |
| Recruitment | — |
| Ethics oversight | — |

Note that full information on the approval of the study protocol must also be provided in the manuscript.

# Field-specific reporting

Please select the one below that is the best fit for your research. If you are not sure, read the appropriate sections before making your selection.

☒ Life sciences ☐ Behavioural & social sciences ☐ Ecological, evolutionary & environmental sciences

For a reference copy of the document with all sections, see nature.com/documents/nr-reporting-summary-flat.pdf

# Life sciences study design

All studies must disclose on these points even when the disclosure is negative.

| | |
|---|---|
| Sample size | Sample sizes were determined via preliminary experiments, and by relying on similar published studies within the field (such as the below studies included in the reference list of the paper).<br><br>24. Stegen, S. J. C. van der et al. Generation of T-cell-receptor-negative CD8αβ-positive CAR T cells from T-cell-derived induced pluripotent stem cells. Nat. Biomed. Eng. 1–14 (2022) doi:10.1038/s41551-022-00915-0.<br>26. Wang, B. et al. Generation of hypoimmunogenic T cells from genetically engineered allogeneic human induced pluripotent stem cells. Nat. Biomed. Eng. 5, 429–440 (2021). |
| Data exclusions | No data were excluded. |
| Replication | The xperimental findings were reproduced in biological and/or technical replications, as indicated in the figure legends. |
| Randomization | Animals were assigned into each group on the basis of average luciferase-signal activity 5 days after tumour engraftment. The assignment criteria were to achieve relative equal average tumour burden in each group; then all groups were randomly selected as either experiment or control. |
| Blinding | For the in vivo experiments, mice were injected by an operator who was blinded to the treatment groups. All groups were treated with the exact same procedures and experienced the same imaging sessions. The imaging signals were batch-generated by imaging software; therefore, there was no bias for either the experimental group or the control group. Blinding was not used for the in vitro experiments, as each cell line used had distinctive phenotypes. |

# Reporting for specific materials, systems and methods

We require information from authors about some types of materials, experimental systems and methods used in many studies. Here, indicate whether each material, system or method listed is relevant to your study. If you are not sure if a list item applies to your research, read the appropriate section before selecting a response.

## Materials & experimental systems

| n/a | Involved in the study |
|-----|-----------------------|
| ☐ | ☒ Antibodies |
| ☐ | ☒ Eukaryotic cell lines |
| ☒ | ☐ Palaeontology and archaeology |
| ☐ | ☒ Animals and other organisms |
| ☒ | ☐ Clinical data |
| ☒ | ☐ Dual use research of concern |

## Methods

| n/a | Involved in the study |
|-----|-----------------------|
| ☒ | ☐ ChIP-seq |
| ☐ | ☒ Flow cytometry |
| ☒ | ☐ MRI-based neuroimaging |

## Antibodies

| | |
|---|---|
| Antibodies used | Antibody List [company, (clone), RRID, catalogue number, dilution]:<br>Anti-human Vβ13.1 - FITC [Biolegend (clone H131) RRID: AB_2564032, 362404, 1:20]<br>Anti-human B2M - FITC [Biolegend (clone 2M2) RRID: AB_492837, 316304, 1:20]<br>Anti-human CD62L - PE [Biolegend (clone DREG-56) RRID: AB_314466, 304806, 1:20]<br>Anti-human CD62L - FITC [Biolegend (clone DREG-56) RRID: AB_314464, 304804, 1:20]<br>Anti-human CD28 - PE [Biolegend (clone CD28.2) RRID: AB_314310, 302908, 1:20]<br>Anti-human CD5 - APC [Biolegend (clone UCHT2) RRID: AB_314098, 300612, 1:20]<br>Anti-human CD8β - APC [Miltenyi (clone REA715) RRID: AB_2659522, 130-110-511, 1:20]<br>Anti-human DLL4 - APC [Biolegend (clone MHD4-46) RRID: AB_11204071, 346508, 1:20]<br>Anti-human CD107α - APC [Biolegend (clone H4A3) RRID: AB_1279055, 328620, 1:20]<br>Anti-human CD137 - APC [Biolegend (clone 4B4-1) RRID: AB_830672, 309810, 1:20]<br>Anti-human CD56 - PE/Cy7 [Biolegend (clone HCD56) RRID: AB_2563927, 362510, 1:20]<br>Anti-human CCR7 - PE/Cy7 [Biolegend (clone G043H7) RRID: AB_11126145, 353226, 1:20]<br>Anti-human CCR7 - PE/Dazzle [Biolegend (clone G043H7) RRID: AB_10913813, 353204, 1:20]<br>Anti-human CD25 - PE/Cy7 [Biolegend (clone BC96) RRID: AB_314282 302612 1:20]<br>Anti-human CD7 - PerCP/Cy5.5 [Biolegend (clone CD7-6B7) RRID: AB_2632912, 343116, 1:20]<br>Anti-human CD45RA - PerCP/Cy5.5 [Biolegend (clone HI100) RRID: AB_893357, 304122, 1:20]<br>Anti-human CD54 (ICAM1) - PerCP/Cy5.5 [Biolegend (clone HA58) RRID: AB_2715948, 353120, 1:20]<br>Anti-human TCRαβ - PE/Dazzle 594 [Biolegend (clone IP26) RRID: AB_2566599, 306726, 1:20]<br>Anti-human CD27 - BV510 [Biolegend (clone O323) RRID: AB_2562086, 302836, 1:20]<br>Anti-human CD27 - BV785 [Biolegend (clone O323) RRID: AB_11219185, 302831, 1:20]<br>Anti-human CD45 - BV510 Biolegend (clone HI30) RRID: AB_2561940, 304036, 1:20]<br>Anti-human CD56 - BV510 [Biolegend (clone HCD56) RRID: AB_2561944, 318340, 1:20]<br>Anti-human CD8α - BV510 [Biolegend (clone RPA-T4) RRID: AB_2564624, 344732, 1:20]<br>Anti-human CD8α - BV605 [Biolegend (clone RPA-T4) RRID: AB_2564391, 300556, 1:20]<br>Anti-human IL-2 - BV605 [Biolegend (clone MQ1-17H12) RRID: AB_2563877, 500332, 1:20]<br>Anti-human CD4 - BV711 [Biolegend (clone SK1) RRID: AB_2565243, 344734, 1:20]<br>Anti-human HLA-A2 - APC/Cy7 [Biolegend (clone BB7.2) RRID: AB_2561568, 343310, 1:20]<br>Anti-human CD3 - APC/Cy7 [Biolegend (clone UCHT1) RRID: AB_830754, 300425, 1:20]<br>Anti-human CD3 - PerCP/Cy5.5 [Biolegend (clone UCHT1) RRID: AB_893299, 300430, 1:20]<br>Anti-human CD3 - BV785 [Biolegend (clone UCHT1) RRID: AB_2687178, 300472, 1:20 ]<br>Anti-human CD326 (EPCAM) - PerCP/Cy5.5 [Biolegend (Clone 9C4) RRID: AB_2098808, 324214, 1:20]<br>Anti-human CD45RO - BV785 [Biolegend (clone UCHL1) RRID: AB_2563819, 304234, 1:20]<br>Anti-human HLA-A,B,C - FITC [Biolegend (clone W6/32) RRID: AB_314873, 311404, 1:20]<br>Anti-human IFNγ - BV785 [Biolegend (clone 4S.B3) RRID: AB_2563882, 502542, 1:20]<br>Anti-human TNFα - APC/Cy7 Biolegend (clone Mab11) RRID: AB_2562870, 502944, 1:20]<br>Anti-human CD137 - PE/Dazzle 594 [Biolegend (clone 4B4-1) RRID: AB_2566260, 309826, 1:20]<br>Anti-human CD3 Ultra-LEAF™ Purified Antibody [Biolegend (clone OKT3) RRID: AB_11150592, 317326, 500ng/mL]<br>Anti-mouse CD29 - PE/Cy7 [Biolegend (clone HMb1-1) RRID: AB_528790, 102222, 1:50]<br>iTAg Tetramer - HLA-A*02:01 NY-ESO-1 (SLLMWITQC) - PE [MBL International, TB-M011-1, 1:50]<br>iTAg Tetramer - HLA-A*0201 MART-1 (ELAGIGILTV) - PE [MBL International, TB-0009-1 1:50 ] |
| Validation | All antibodies used are commercially available and were validated for use in flow cytometry. Data are available on the manufacturer's website. |

## Eukaryotic cell lines

Policy information about cell lines and Sex and Gender in Research

| | |
|---|---|
| Cell line source(s) | MS5-hDLL4, K562-CD80-CD83-CD137L-HLA-A*0201/B2M/NYESO157-165 aAPC, K562-HLA-A*0201/B2M/NYESO(157-165) |

| Cell line source(s) | aAPC, and K562-HLA-A*0201/B2M/MART1(26-35) aAPC were previously generated in the Crooks Lab. For this study, the MS5-hDLL4-ICAM, MS5-hDLL4-A*0201 (A02), MS5-hDLL4-A*0201-hB2M (A02B), and MS5-hDLL4-A*0201-hB2M-ICAM (A02BI) lines were generated in the Crooks Lab via transduction of the MS5-hDLL4 line with lentivirus. |
|---|---|
| | The H1 and ESI017 PSC lines were purchased from WiCell and ESI BIO, respectively. ESI017 RAG1–/–RAG2–/– double knockout (DKO) and RAG1–/–RAG2–/–B2M–/– triple knockout (TKO) PSC lines; H1 DKO and TKO PSC lines; ESI017 DKO and TKO transduced with UBC-1G4TCR-mTagBFP2 (ESI017 DKO+TCR and ESI017 TKO+TCR) PSC lines; H1 DKO and TKO transduced with UBC-1G4TCR-mTagBFP2 (H1 DKO+TCR and H1 TKO+TCR) PSC lines; ESI017 and H1 transduced with UBC-1G4TCR-mTagBFP2 (ESI017 WT+TCR and H1 WT+TCR) PSC lines were generated for this study in the Crooks Lab. |
| | 293T cells were purchased from ATCC (Cat. CRL-3216) for the packaging of lentivirus particles. |
| Authentication | K562 aAPCs and MS5-hDLL4 lines were authenticated via flow cytometry for their respective markers. Certificates of authentication were provided for the PSC lines that were purchased from WiCell and ESI BIO. Edited PSC lines were authenticated by genotyping PCR, karyotyping, and flow cytometry. Changes in morphology and pluripotency were routeinly monitored via flow cytometry. |
| Mycoplasma contamination | The cell lines were routinely checked for mycoplasma contamination measured using bioluminesence; no infections were detected. |
| Commonly misidentified lines (See ICLAC register) | No commonly misidentified cell lines were used. |

# Animals and other research organisms

Policy information about studies involving animals; ARRIVE guidelines recommended for reporting animal research, and Sex and Gender in Research

| Laboratory animals | We used NOD.Cg-Prkdcscid Il2rgtm1Wjl/SzJ (NSG) mice, purchased from The Jackson Laboratory (Strain #005557; RRID:IMSR_JAX:005557. Details can be found at https://www.jax.org/strain/005557). Female mice between the ages of 6–8 weeks were used. No further modification of the animals occurred before tumour engraftment and T cell treatment. and then bred at UCLA. 8–14-week-old female mice were used for the in vivo experiments. |
|---|---|
| | All animals were housed in the UCLA-CNSI vivarium, which is maintained at approximately 20–26 °C, 30–70% humidity, with a 12-h-light/dark cycle (6am/6pm). Up to 5 NSG mice were kept in an individually ventilated cage. Details can be found at https://rsawa.research.ucla.edu/arc/temp-and-humidity. |
| Wild animals | The study did not use wild animals. |
| Reporting on sex | Sex was not considered in the study design, yet the findings apply to both sexes. |
| Field-collected samples | The study did not involve samples collected from the field. |
| Ethics oversight | All animal experiments were conducted under a protocol approved by the UCLA Chancellor's Animal Research Committee (ARC), also known as the UCLA institutional animal care and use committee (IACUC). |

Note that full information on the approval of the study protocol must also be provided in the manuscript.

# Flow Cytometry

## Plots

Confirm that:

☒ The axis labels state the marker and fluorochrome used (e.g. CD4-FITC).

☒ The axis scales are clearly visible. Include numbers along axes only for bottom left plot of group (a 'group' is an analysis of identical markers).

☒ All plots are contour plots with outliers or pseudocolor plots.

☒ A numerical value for number of cells or percentage (with statistics) is provided.

## Methodology

| Sample preparation | Cells were periodically harvested from filters over 7 weeks of in vitro T-cell differentiation. A total of 9 aggregates in 3 filters (3 aggregates/filter) were disassociated in MACS buffer (PBS, 0.5% BSA, 2mM EDTA), and filtered through a 50-μm strainer. Cells were blocked using FcX block (Biolegend) and stained with antibodies at a 1:20 dilution in a final volume of 100 μL with MACS buffer at 4 C. For intracellular cytokine assays, surface antibodies were stained prior to fixation, and fixation/permeabilization was performed according to the protocol provided by the BD fix/perm kit. |
|---|---|
| Instrument | Flow-cytometry samples were analysed using the BD LSRII Fortessa, and the samples were sorted using either the BD FACSARIA or FACSARIA-H instruments. |

| Software | BD systems used BDFACSDiva software v8.0.2 for flow-cytometry data collection. |
|---|---|
| | FlowJo v10.8.1 was used for the analysis of flow-cytometry data. |
| Cell population abundance | Sorted populations, using BD either BD FACSARIA or FACSARIA-H instruments and Miltenyi MACS columns, were confirmed by flow cytometry. |
| Gating strategy | Flow-cytometry samples were first gated on cell size and granularity using FSC-A/SSC-A plots, drawing the lymphocyte gate. Doublets were excluded twice using FSC-H/SSC-H and FSC-W/SSC-W gates before dead cells were excluded using a viability stain [DAPI or Zombie (Biolegend)]. Gating strategies are shown in the supplementary information. |

☒ Tick this box to confirm that a figure exemplifying the gating strategy is provided in the Supplementary Information.

