## [Peer Review File · Nature Biomedical Engineering]

Generation of antigen-specific mature T cells from RAG1^{-/-} RAG2^{-/-} B2M^{-/-} stem cells by engineering their microenvironment

Corresponding author: Gay Crooks

Editorial note

This document includes relevant written communications between the manuscript's corresponding author and the editor and reviewers of the manuscript during peer review. It includes decision letters relaying any editorial points and peer-review reports, and the authors' replies to these (under 'Rebuttal' headings). The editorial decisions are signed by the manuscript's handling editor, yet the editorial team and ultimately the journal's Chief Editor share responsibility for all decisions.

Any relevant documents attached to the decision letters are referred to as **Appendix #**, and can be found appended to this document. Any information deemed confidential has been redacted or removed. Earlier versions of the manuscript are not published, yet the originally submitted version may be available as a preprint. Because of editorial edits and changes during peer review, the published title of the paper and the title mentioned in below correspondence may differ.

Correspondence

Fri 21 Oct 2022

Decision on Article NBME-22-2014A

Dear Dr Crooks,

Thank you again for submitting to *Nature Biomedical Engineering* your manuscript, "Engineering the in vitro microenvironment for positive selection of stem cell-derived, Class I MHC-null, antigen-specific T cells". The manuscript has been seen by 4 experts, whose reports you will find at the end of this message. You will see that the reviewers appreciate the work, and that they raise a number of technical criticisms that we hope you will be able to address. In particular, because we have placed editorial emphasis on the methodological advances of the work, we would expect that a revised version of the manuscript addresses the technical and methodological shortcomings highlighted by the reviewers, and in particular:

- * Thorough characterization of the T cells, as per the many relevant points of all reviewers.
- * Functional comparisons with alternative T-cell-induction-and-selection methods previously reported (in particular methods leveraging CD3 stimulation), as per the arguments of Reviewers #1 and #4.
- * Additional TCR-HLA combinations in multiple PSCs, as argued by Reviewer #1.
- * Thorough methodological reporting, with emphasis on facilitating reproducibility.

Editorially, evidence of the method working with iPSCs, although welcome, won't be a necessary requirement for the revision to be considered. And because of our editorial emphasis on the method rather than its clinical applicability, we will not require evidence of a more clinically relevant ATO system to be included in the revised manuscript. We have communicated these points to the reviewers.

When you are ready to resubmit your manuscript, please upload the revised files, a point-by-point rebuttal tothe comments from all reviewers, the reporting summary, and a cover letter that explains the main improvements included in the revision and responds to any points highlighted in this decision.

Please follow the following recommendations:

- * Clearly highlight any amendments to the text and figures to help the reviewers and editors find and understand the changes (yet keep in mind that excessive marking can hinder readability).
- * If you and your co-authors disagree with a criticism, provide the arguments to the reviewer (optionally, indicate the relevant points in the cover letter).
- * If a criticism or suggestion is not addressed, please indicate so in the rebuttal to the reviewer comments and explain the reason(s).
- * Consider including responses to any criticisms raised by more than one reviewer at the beginning of the rebuttal, in a section addressed to all reviewers.
- * The rebuttal should include the reviewer comments in point-by-point format (please note that we provide all reviewers will the reports as they appear at the end of this message).
- * Provide the rebuttal to the reviewer comments and the cover letter as separate files.

We hope that you will be able to resubmit the manuscript within 20 weeks from the receipt of this message. If this is the case, you will be protected against potential scooping. Otherwise, we will be happy to consider a revised manuscript as long as the significance of the work is not compromised by work published elsewhere or accepted for publication at *Nature Biomedical Engineering*.

We hope that you will find the referee reports helpful when revising the work, which we look forward to receive. Please do not hesitate to contact me should you have any questions.

Best wishes,

Valeria

Dr Valeria Caprettini
Associate Editor, Nature Biomedical Engineering

Reviewer #1 (Report for the authors (Required)):

Chang et al. found that endogenous TCR gene rearrangements occurred during differentiation of CD8 T cells from human embryonic stem cells (ESCs) by ATO culture, and that both expression of HLA class I molecule and the HLA-restricted TCR on differentiating cells were required to induce positive selection into CD8 single positive T cells. In addition, with a view to clinical applications, the authors found that HLA class I gene transfer into MS5hDLL4 could compensate HLA class I lack on differentiating ESC-derived CD8 T cells in HLA-restricted TCR mediated positive selection, allowing the generation of HLA class I KO “universal” ESC-CD8 T cells. Those findings could be informative for development of pluripotent stem cell-based CD8 T cell immunotherapy.

On the other hand, there are some concerns that only limited combinations of TCR and HLA have been investigated in the study, the observed phenomenon itself is within the range expected from previous murine experiments, and the missing comparison between the generated T cells and existing T cells induced by different methods in vitro/vivo to evaluate therapeutic efficacy.

Major comments.

1, For a better understanding of the impacts of genetic modification on ESC and MS5hDLL4, additional

analysis such as RNA expression and TCR-mediated signaling should be applied to differentiating CD4/8 T cells, differentiated CD8 T cells, and expanded CD8 T cells those induced by conditions of WT/MS5hDLL4, WT+TCR/MS5hDLL4, DKO+TCR/MS5hDLL4, TKO+TCR/MS5hDLL4, TKO+TCR/MS5hDLL4-A02BI. Especially, expression level and kinetics of T-cell differentiation-related transcriptional factor and strength and kinetics of TCR-downstream signaling among the T cells from different conditions will be informative to make authors claims further be confirmed.

2, The observation of CD8 T cell differentiation failure in ATO under HLA-TCR mismatch setting and its rescue by MS5hDLL4-A02BI is quite interesting and should be further evaluated. In general, TCR stimulation and expansion of matured T cell induces further differentiation to terminal effector T cell via effector memory T cells. From that point, it is worthy to clarify contribution of ICAM-1 transduction for expansion of T cells keeping naïve-like phenotype. Especially, how TCR-signal intensity and duration changed after ICAM-1 transduction; monitoring them during ATO differentiation would be great, but difficult, so a simple co-culture system with MS5hDLL4-A02B or MS5hDLL4-A02BI and TKO + TCR DP- T cells would be useful to evaluate TCR-signal intensity and duration. It would also be interesting to see if there is a dependence or threshold to ICAM-1 expressed on MS5hDLL4. besides, evaluation of memory/effector-related gene expression in T cells differentiated with/without ICAM-1 and functional experiments such as those performed in Fig 5 will provide a better understanding of contribution of ICAM-1.

Also, the experiments presented in the paper are based only on the expression of HLA-A*02:01 in combination with the same HLA-restricted NY-ESO-1 TCR, and it would be desirable to confirm reproducibility with other HLA/TCRs. For example, using the HLA-A2-restricted WT1-TCR or using the HLA-A*24:02 transduced MS5hDLL4 and the HLA-A24-bound WT1-TCR.

3, Although the authors wrote in discussion part that “In future studies, it would be interesting to compare the function of T cells produced with antibody stimulation of the TCR with those induced through cellular presentation of cognate MHC in the ATO”, experiments to evaluate effector function and therapeutic efficacy of ATO-derived T cells should be performed in this study in comparison with 2D culture induced T cells (with OP9 feeder and/or feeder free cultures) and primary T cells. Although animal experiments indicated a trend of tumour growth inhibition in the initial stages, overall survival rate is essential for discussing the therapeutic effect. In addition, PK data after transplantation is desirable to be obtained.

Minor comments.

1, The differentiation process in ATO seems depends on survival of MS5hDLL4 cells to give appropriate magnitude of Notch- and TCR-signaling at appropriate timing. Authors are requested to show data how long MS5hDLL4-A02BI survive in ATO to give Notch- or TCR-signaling.

2, The authors state that “The resulting SP8 T cells exhibited potent antigen-specific cytotoxicity in vitro as well as tumour control in vivo that was superior to PSC-derived T cells with intact endogenous TCR rearrangement.” However, it simply seems depends on tetramer-positive cell number difference between WT+TCR and TKO+TCR due to lower incidence of TCR mispairing in TKO+TCR cells. In case the authors think there should be any difference in T cell quality, the same number of tetramer positive cells should be used for the animal experiment to confirm it.

3, About Fig. 2B, SP8 differentiation from CD4/CD8 DP-T appears to be delayed in H1 DKO+TCR condition. Authors are requested to explain the reason for it.

4, Please indicate the expression levels of hDLL4, HLA-A*02:01 and ICAM in MS5.

5, About Fig. 5A, "TNFg" on Y-axis should be "IFNg"

6, About Fig. 5G, T-cell reactivity to NY ESO aAPC IL2 IL7 stimulation seems very weak. Authors are requested to explain the reason for it and to evaluate proliferative reactivity of T cells by standard stimuli such as CD3/28 beads.

Reviewer #2 (Report for the authors (Required)):

Manuscript: Engineering the in vitro microenvironment for positive selection of stem cell-derived, Class I MHC-null, antigen-specific T cells

This manuscript demonstrates the feasibility to generate allogeneic T cells from PSC by manipulating not only the original cell source (stem cells) but also the in vitro microenvironment when alloreactive mechanisms are disrupted. The study provides an improved version of the already existing “Artificial Thymic Organoid (ATO)” 3D culture system, allowing for the generation of CD8SP T cells, which lack endogenous TCRs and MHC Class I expression (see also Montel-Hagen et al. 2019, same group). These characteristics will help to develop an “off-the-self” T cell product, which will be able to be used as a cell therapy without the limitations of the “donor T cell mediated alloreactivity; GvHD” and “host mediated rejection of HLA-mismatched allogeneic donor cells”.

In the present study, the authors used the CRISPR Cas9 system able to generate pluripotent stem cells (PSCs) knockout for all the endogenous TCRs (RAG1 and RAG2 knockout) and the beta-2-microglobulin gene (MHC Class I antigen). To succeed positive selection in the ATO they genetically modified the murine MS5-hDLL4 stromal line to express the cognate human MHC. However, as the authors mentioned, further modifications are needed to fully avoid the allogeneic rejection. This work is focused on the translational/clinical use of PSC-derived T cells; however, it is known that the use of PSC is associated with ethical issues, unlike the use of human induced pluripotent stem cells. The authors should have focused on the best offer available at this time and offering a big step forward into clinics, which was not achieved here. Additionally, by not targeting other mechanisms (key players) involved in alloreactivity, like class II MHC or even the most likely activation of NK cells by the absence of class I MHC, an incredibly critical aspect is missing, which, in the case of combination with the present data, would substantially increase the importance of the work.

Overall, although the topic is interesting and important for the use of T cells as cellular therapies, the ATO system and the genetic manipulation of PSCs so that the T cells express a certain TCR was already published from the same research group. In the present study, the authors have managed to improve the already existing “system” however a more detailed analysis of the T cells generated using this technique will bring an additional novelty in this study supporting its publication in the journal of “Nature Biomedical Engineering”.

Major concerns:

1. In the present manuscript, the authors used PSC (e.g. embryonic stem cells; ESCs) as the starting cell source. It is known that the use of stem cells has the disadvantage of raising ethical concerns. On the other hand, hiPSC offers the advantage of substantially decreasing ethical concerns, hence, bringing this work one step closer to the clinics. This point is of particular interest, as the authors state in the introduction the increased need of “off-the-shelf” T cell products for clinical translation. Whether ESCs or iPSCs are used later as a starting material, the authors should provide PoC using iPSC. Given the strong clinical introduction, the authors could also provide evidence, that the ESC lines used are applied in clinical trials.
2. To further improve/increase the novelty of this work, as the authors mentioned, additional strategies to overcome the alloreactive rejection must be applied. As the author already published a similar work before (or at least parts of it), adding additional mechanisms to avoid NK activation or deleting class II MHC would be extremely beneficial.
3. The underline goal of the study would be the development of suitable T cell products for clinical use (as mentioned also in the introduction). Having this in mind, the use of a murine stromal cell line does not follow that path. Given the strong introduction on the clinical use, the authors should elaborate on strategies to circumvent the use of murine stromal cells. As a consequence, maybe other “ATO” systems (coating, human cells (e.g. OP9?) etc.) should be developed to bring the approach really one step closer towards the clinics. Of note, I appreciate the fact that the MS5 cells resemble the current state-of-the-art for the ATO system, however at the current stage, it may be very much challenging to produce an off-the-shelf T cell product using the current ATO version. Thus, the scope of the current manuscript is rather mechanistic than clinically. Along this line, more mechanistic on the e.g. T cell development should be provided to give some new insights into the developmental trajectories of human PSC-derived T cells.
4. In this study, the authors in order to characterize the iPSC-derived TKO CD8SP T cells generated with the ATO system, performed TCR repertoire analysis and FACS analysis for specific markers, such as CD45RA, CD45RO, and CD62L. However, the authors should provide a more detailed characterization of the derived-cells, performing for instance transcriptomic profiling of the TKO CD8SP T cells. This might be helpful to evaluate possible differential regulation of genes due to the multiple genetic modifications.

5. Similar to point No#4, it would be beneficial to identify whether these cells express known factors important for their functionality and which subset of CD8SP T cells they resemble more. The authors should perform single-cell RNA seq analysis and subsequent trajectory analysis to further dissect and reveal the different developmental stages of the cells in the ATO system and possibly to compare it with normal thymic development. These are especially important when these cells are generated with the goal to be used as an “off-the-self” engineered T cell product for cellular therapies.

6. It is known that that the thymic microenvironment plays a crucial role in the epigenetic regulation of T cells during their development. The epigenetic modifications contribute to all stages of T cell differentiation and affect the cells’ functionality and stability. The authors should provide some data regarding the epigenetic status of the derived CD8SP T cells (e.g. methylation status of specific loci) also in comparison with their in vivo counterparts.

Minor concerns:

1. In the second sentence in the second paragraph of the introduction, the respective references are missing. Please add the appropriate literature.
2. Full name of EMO is missing from the main text and appears only in the supplementary material. The author should add it in the main text as well.
3. Figure 2D: The authors should describe the pre-gate they have done for the analysed populations.
4. The axis labels do not state the fluorochrome that was used. This is necessary according to the journal’s guidelines.
5. It would be interesting to see photomicrograph images from the ATOs as well as the ATO-derived T cells. In the case of ATO-derived T cells, cytopsin images would further increase the focus on the final product as well as enlightening the reader image about the cell product.
6. It would be interesting to have the FSC x SSC plot to observe ROI of the expected PSC-derived T cells.
7. Are there any contaminant cells? For example, an additional panel for several cell types (e.g. Macrophages) must be added. Cytopsin should also add additional information in this regard.
8. Some FACS plots seem to be in need for additional compensation (e.g. Supplementary fig.2, 3,5).

Reviewer #3 (Report for the authors (Required)):

A highly interesting MS reporting the generation of monoclonal TCR transgenic human CD8 T cells from PSCs using an in vitro system (ATO) for positive selection. It is shown that PSCs can be manipulated to lack both endogenous Rag (thus preventing endogenous TCR rearrangements) as well as b2m (with the perspective of protecting them from allorecognition in a therapeutic transfer setting). Furthermore, the MS describes crucial advances in the manipulation of the selecting mouse stromal component (expression of the selecting human MHC haplotype as well as ICAM). Finally, impressive data on the in vitro and in vivo functionality of these cells are presented. Together, the MS should be of outstanding interest to the readership of NBME, as it may pave the way towards overcoming major hurdles in the generation of T cell immunotherapies from allogeneic stem cells.

Minor technical criticisms or questions:

- 1) Fig 1: The TCR-less DP cells that transiently emerge from DKO cells are referred to as ‘innate-like’. The authors may want to consider whether doing so solely on the basis of CD8alpha/alpha expression is justified. Probably fair to say that their cellular identity remains entirely obscure; not an issue.
- 2) Suppl Fig 2: Throughout Fig S2, it seems that mCD29 and CD56 are used in the same channel. The gating strategy then indicates further gating on mCD29-CD56+ cells. Shouldn’t this be mCD29-CD56-?
- 3) Fig 5A: TNFg on the vertical axis!!?

Suggestions for improvement:

- 4) One thing that may be interesting to the reader is whether the cells persist/expand in the in vivo tumor model (and how this compares to the WT+TCR setting). Purely optional, but the authors may have some data on this anyhow.

Ludger Klein

Reviewer #4 (Report for the authors (Required)):

In this Study, Chang et al. described a new strategy to achieve terminal differentiation of TCR-specific T cells that have been rendered MHC Class I deficient (“cloaked”), using pluripotent stem cells and the previously established ATO system. Knockout of RAG1/RAG2 produces TCR-null cells by preventing V(D)J recombination, and consequently blocks the maturation of T cells beyond the DP stage. Expression of exogenous TCR can rescue this block by restoring positive selection, but this process typically depends on TCR engagement with endogenous class-I MHC, which needs to be removed to produce hypoinmunogenic cells for allogeneic immunotherapies. To overcome this, Chang et al. engineered the MS5 stroma cells used in the ATO to express HLA and adhesion molecule ICAM1 to enable positive selection. This strategy allows generation of MHC-null TCR-specific PSC-T cells that outperform the TCR-specific PSC-T cells without RAG1/RAG2/B2M KO. This paper provides another means of deriving T cells from PSCs that might be a source of allogeneic adoptive immunotherapies.

Major points:

1. Prevention of TCR rearrangement during T cell differentiation from PSCs by knocking out Rag 2 has been reported (Minagawa et al, 2018, Wang et al, 2021). RAG1/2 knockout is predicted to elicit the same effect. One key question is why TCR-null SP cells can be generated using the traditional OP9 system (Wang et al. 2021) but not via the ATO system? Will stimulation with anti-CD3 at DP stage also work in the ATO system? These are important questions to resolve to justify the engineering of stroma cells.
2. The ATO system has been shown to produce more mature T cells than the OP9 system. Do the TCR-null cells rescued by engineered stroma cells still display a mature phenotype? This is important because the differentiation of DKO/TKO PSCs into T cells via HLA rescue seems to be less efficient than WT cells (Fig 2B, 3B). A refined molecular characterization (eg. RNA-seq, ideally single cell) of TKO TCR PSC-T vs WT TCR PSC-T cells would address this question and add considerably to the characterization of the T cells produced.
3. Since hypoinmunogenic antigen-specific PSC-T cells have been reported (Wang et al. 2021), it would be extremely helpful to compare this new strategy with the previously reported system. For functional assays, it is conceivable that TKO PSC-T cells outperform WT PSC-T cells due to the lower immunogenicity. The key comparison should be between TKO TCR PSC-T cells generated using previously established methods and the cells generated using the ATO system with the engineered microenvironment.

Minor point:

Mechanistically, RAG is also required for pre-TCR, leaving the question for why cells end up blocked at the DP stage, not DN stage? A citation to the situation in murine RAG KOs or human Omenn Syndrome patients would be informative.

Fri 12 May 2023

Decision on Article NBME-22-2014B

Dear Dr Crooks,

Thank you again for submitting to *Nature Biomedical Engineering* your manuscript, "Engineering the in vitro microenvironment for positive selection of stem cell-derived, Class I MHC-null, antigen-specific T cells". The manuscript has been seen by the original 4 experts, whose reports you will find at the end of this message.

You will see that the reviewers appreciate the work, and that they raise a number of technical criticisms that we hope you will be able to address.

When you are ready to resubmit your manuscript, please upload the revised files, a point-by-point rebuttal to the comments from all reviewers, the reporting summary, and a cover letter that explains the main improvements included in the revision and responds to any points highlighted in this decision.

Please follow the following recommendations:

- * Clearly highlight any amendments to the text and figures to help the reviewers and editors find and understand the changes (yet keep in mind that excessive marking can hinder readability).
- * If you and your co-authors disagree with a criticism, provide the arguments to the reviewer (optionally, indicate the relevant points in the cover letter).
- * If a criticism or suggestion is not addressed, please indicate so in the rebuttal to the reviewer comments and explain the reason(s).
- * Consider including responses to any criticisms raised by more than one reviewer at the beginning of the rebuttal, in a section addressed to all reviewers.
- * The rebuttal should include the reviewer comments in point-by-point format (please note that we provide all reviewers will the reports as they appear at the end of this message).
- * Provide the rebuttal to the reviewer comments and the cover letter as separate files.

We hope that you will be able to resubmit the manuscript within 20 weeks from the receipt of this message. If this is the case, you will be protected against potential scooping. Otherwise, we will be happy to consider a revised manuscript as long as the significance of the work is not compromised by work published elsewhere or accepted for publication at *Nature Biomedical Engineering*.

We hope that you will find the referee reports helpful when revising the work, which we look forward to receive. Please do not hesitate to contact me should you have any questions.

Best wishes,

Valeria

Dr Valeria Caprettini
Associate Editor, Nature Biomedical Engineering

Reviewer #1 (Report for the authors (Required)):

The revised manuscript by Chong et al. clarifies and discusses the proximity among iPS-T cells from various condition by scRNA analysis and the difference in quality of DP cells by comparison with the previously reported 2D culture, although to a limited extent. Although we feel that the manuscript has improved in some

respects, there are still some points that remain a concern through peer review and need to be addressed.

Major.

To my first question, my proposed analysis of differentiating T cell samples was not performed. Since the paper focuses on the importance of TCR and HLA in differentiation, and authors stated as "Transgenic signals required to rescue positive selection were supplied by separate cellular sources; T cell precursors from edited PSC expressed a single TCR, and a murine stromal line provided cognate human MHC and other critical T cell development signals." in the abstract, I would like to see the changes in gene expression during differentiation, especially the differences and fluctuations in TCR signaling-related molecules.

On the other hand, the authors present scRNA data for T cells amplified after induction of differentiation under different conditions, leading to the conclusion that there are no differences in the final T cells. If these T cells are to be considered for therapeutic use, it is important that they have the same characteristics regardless of the method of production.

However, we have the following concerns about the analysis.

In Figure.4A, the authors compare peripheral blood NK, monocyte, SP8, thymus SP8, and their own SP8.

Why include NK, monocyte in the comparison? By including cell types that are clearly distant from each other, wouldn't it obscure the differences within the SP8 population? In fact, since several differentially expressed genes can be identified in Fig. 4B, would different clusters emerge if the analysis is limited to SP8 only?

I am concerned about the low CCR7 expression in Fig4B. Has the expression been lost due to expanded culture? And what about the expression of the SELL encoding CD62L, a characteristic of ATO-derived SP8?

Regarding my second question, the authors have stated that the mechanism of ICAM1 involvement is outside the scope of this manuscript. Since the manuscript demonstrates the importance of TCR-HLA signaling in differentiation, the mechanism of ICAM1 involvement should be clarified as authors expected. We feel that one methodology might be to evaluate downstream gene variation from scRNA-seq data during differentiation, and another methodology might be to evaluate signal intensity enhancement downstream of the TCR under co-culture, respectively.

The authors have taken reproducible data of the importance of TCR and MHC for differentiation using different TCR-MHC combinations. However, as shown in Fig. S8A, only 21.7% of SP8 is present in F5 TCR at 7 weeks; if the proliferative capacity of F5 TCR SP8 is also 10-fold per cycle as shown in Fig. 6G, it may be difficult to obtain cells at the level of clinical application. This point should be discussed.

In response to my third question, a comparison was made between DP cells obtained from 2D culture and ATO-derived DP cells, although limited. However, according to a previous report, the PHA-stimulated amplification method requires the use of PBMCs, but this is not mentioned in the manuscript, and if true, it is not a correct comparison with the previous report. In fact, this may be the reason why the OKT-PHA-PHA group does not increase after 7 days.

The authors wrote in the abstract that "Edited T cells exhibited superior tumor control in vivo likely due to the absence of TCR mispairing." However, the transplantation of 1×10^7 TKO SP8 into 5×10^5 NALM6 failed to suppress tumor growth. Survival data should be provided to compare the therapeutic effect with the previously reported Primary T. Hopefully, mice treated with Primary T under the same conditions will be set up, and both IVIS data and survival data are needed. If the authors cannot show a sufficient treatment effect from these data, it seems to me that the treatment effect can be de-emphasized and Fig. 6 can be included as a supplement.

Minor.

The bottom two groups in Fig6F may be a misdescription of MART aAPC. Also, "Expanded WT+TCR and TKO+TCR T cells also displayed potent antigen-specific cytotoxicity in vitro at an effector-to-target ratio as low as 1:32" in the manuscript.

However, in Fig6F, 1:32 appears to have little effect, and a ratio of 1:8 to 1:4 would be more appropriate.

Reviewer #2 (Report for the authors (Required)):

After evaluating the author's reply to my comments raised, I very much appreciate the changes made. However, I would like to focus on my previous question #2.

I would like to highlight that, even though I can understand the position of the authors, I do not share the opinion that this comment made is outside the scope of the manuscript. Of note, the authors highlight several

times in their manuscript the beneficial use of "allogenic" cells, while also applying ESC, which later on are indeed applied in a allogenic fashion.

Furthermore, the authors claim in the introduction section that "Cell-based immunotherapies in which autologous T cells have been engineered to express chimeric antigen receptors (CARs) and antigen-specific T cell receptors (TCRs) have produced impressive clinical responses in patients with otherwise treatment-refractory diseases.", therefore, bringing solutions to overcome this critical challenge is of most importance. Hence, I would like to reiterate the importance of this question and ask for additional data that would indicate the possibility to use this model in an allogeneic setting, highlighting that the system introduced is working in allogenic setting.

Reviewer #3 (Report for the authors (Required)):

I was very positive in my initial assessment of the MS and didn't raise major concerns. The authors have done a great job addressing most of the concerns raised by the other reviewers. The inclusion of additional data has even further improved this very interesting MS.

Reviewer #4 (Report for the authors (Required)):

The authors have responded to each of our prior numbered points, as indicated below:

1. Prevention of TCR rearrangement during T cell differentiation from PSCs by knocking out Rag 2 has been reported (Minagawa et al, 2018, Wang et al, 2021). RAG1/2 knockout is predicted to elicit the same effect. One key question is why TCR-null SP cells can be generated using the traditional OP9 system (Wang et al. 2021) but not via the ATO system? Will stimulation with anti-CD3 at DP stage also work in the ATO system? These are important questions to resolve to justify the engineering of stroma cells.

Authors performed new experiments to compare the DP stimulation required in other non-ATO systems to produce SP8 cells (Supplementary Fig. 9). In these experiments, authors included the use of CD3/28 beads, anti-CD3 antibody (OKT3), and artificial antigen presenting cells (aAPCs) for the SP induction and cell activation/expansion. The results show that though previously reported strategies can induce SP production from week 3 ATO DP cells, only CD8 SP cells collected directly from ATOs display a naive/stem cell memory phenotype. These data suggest the superiority of using engineered ATO system to produce class I MHC-null TKO PSC-T cells

2. The ATO system has been shown to produce more mature T cells than the OP9 system. Do the TCR-null cells rescued by engineered stroma cells still display a mature phenotype? This is important because the differentiation of DKO/TKO PSCs into T cells via HLA rescue seems to be less efficient than WT cells (Fig 2B, 3B). A refined molecular characterization (eg. RNA-seq, ideally single cell) of TKO TCR PSC-T vs WT TCR PSC-T cells would address this question and add considerably to the characterization of the T cells produced.

Authors performed scRNA-seq to compare the molecular features of TKO PSC-T cells generated using engineered ATO vs. traditional ATO PSC-Ts and thymic/PBMC SP8 T cells. The results show that TCR-null cells rescued by engineered ATO stroma cells display a similar phenotype than WT ATO PSC-T cells and are similar to conventional primary SP8 T cells, not NK or monocytes.

3. Since hypoinmunogenic antigen-specific PSC-T cells have been reported (Wang et al. 2021), it would be extremely helpful to compare this new strategy with the previously reported system. For functional assays, it is conceivable that TKO PSC-T cells outperform WT PSC-T cells due to the lower immunogenicity. The key comparison should be between TKO TCR PSC-T cells generated using previously established methods and the cells generated using the ATO system with the engineered microenvironment.

Authors provided a comparison of engineered ATO vs. existing strategies via immunophenotypic characterizations, but did not perform additional functional assays or in vivo assays to address the concern. Reviewer #3 also suggested similar experiment. It is understandable that in vivo experiments are time consuming, and it is challenging to obtain results within the revision time window. However, such

experiments are important to justify the new method of using engineered stroma cells in the ATO system, as alternative strategies for SP induction, such as using CD3/28 beads, might be more suitable for large scale manufacture.

Minor point:

Mechanistically, RAG is also required for pre-TCR, leaving the question for why cells end up blocked at the DP stage, not DN stage? A citation to the situation in murine RAG KOs or human Omenn Syndrome patients would be informative.

Authors have addressed the question.

Mon 03 Jul 2023

Decision on Article NBME-22-2014C

Dear Dr Crooks,

Thank you for your revised manuscript, "Engineering the in vitro microenvironment for positive selection of stem cell-derived, Class I MHC-null, antigen-specific T cells", which has been seen by Reviewers #1 and #2. In their reports, which you will find at the end of this message, you will see that the reviewers acknowledge the improvements to the work and raise a few additional technical criticisms that we hope you will be able to address.

As before, when you are ready to resubmit your manuscript, please upload the revised files, a point-by-point rebuttal to the comments from all reviewers, the reporting summary, and a cover letter that explains the main improvements included in the revision and responds to any points highlighted in this decision.

As a reminder, please follow the following recommendations:

- * Clearly highlight any amendments to the text and figures to help the reviewers and editors find and understand the changes (yet keep in mind that excessive marking can hinder readability).
- * If you and your co-authors disagree with a criticism, provide the arguments to the reviewer (optionally, indicate the relevant points in the cover letter).
- * If a criticism or suggestion is not addressed, please indicate so in the rebuttal to the reviewer comments and explain the reason(s).
- * Consider including responses to any criticisms raised by more than one reviewer at the beginning of the rebuttal, in a section addressed to all reviewers.
- * The rebuttal should include the reviewer comments in point-by-point format (please note that we provide all reviewers will the reports as they appear at the end of this message).
- * Provide the rebuttal to the reviewer comments and the cover letter as separate files.

We hope that you will be able to resubmit the manuscript within 12 weeks from the receipt of this message. If this is the case, you will be protected against potential scooping. Otherwise, we will be happy to consider a revised manuscript as long as the significance of the work is not compromised by work published elsewhere or accepted for publication at *Nature Biomedical Engineering*.

We look forward to receive a further revised version of the work. Please do not hesitate to contact me should you have any questions.

Best wishes,

Valeria

Dr Valeria Caprettini
Associate Editor, Nature Biomedical Engineering

Reviewer #1 (Report for the authors (Required)):

Dear authors,

I would like to thank authors for their thoughtful responses to additional questions. I also appreciate the additional data provided. I feel that the manuscript has improved, but we do have additional questions.

#1, I am aware of the authors' comments (1.1-1.3) regarding the analysis of gene expression over time in ATO and the analysis of the effects of ICAM1, and suggest that point in 1.3 be discussed in the Discussion as a limitation of the ATO analysis.

#3, I suggest the authors reconsider their clustering method, as they have shown a single cluster in green on all individual UMAP panel as a result of simple extracting of SP8s. It is odd that there is only a single cluster in the sample, and indeed there is a certain heterogeneity in each sample when looking at SELL and CCR7 expression. The correct approach would be to integrate the results from the six SP8 samples and then perform a UMAP clustering analysis to determine which samples contribute to which clusters and to what extent. That analysis can more reasonably and visually discuss the proximity and differences of the 6 cell types.

#7, I understood the situation. I agree that it is not always easy to fully reproduce the protocols of other institutions. However, since this is an OKT3-PHA stimulation protocol that does not include PBMCs, the author's description of "SP8s generated through activation of ATO-DP precursors through CD3 engagement adopted a CD8ab+ CD45ROCD45RA+CD62L- phenotype (Supplementary Fig. 9B), but did not expand with further stimulation (Supplementary Fig. 9C-D)." needs to be supplemented with the fact that "in the absence of PBMCs".

#8, I would like to thank to authors for the additional data. I agreed that TKO compared to the no treatment group statistically significantly prolongs survival by log-rank test over WT. However, I question whether the 2.4 day increase in mean survival has clinical significance. In addition, if the authors are truly interested in the utility of antigen-specific iPS-T cells generated from ATO, it is important to take a comparative approach with allogeneic T cells, rather than much emphasizing their in vivo efficacy to non-treatment group, I believe that at least these issues should be discussed seriously rather than emphasizing the in vivo efficacy.

Reviewer #2 (Report for the authors (Required)):

I would like to thank the authors in replying to my concern and given additional information in the discussion section.

However, given the focus of the manuscript and the rational how the manuscript is written (focus on translation rather than biology/technique) I still believe that additional insights into the mode of action and how the cells behave in vivo would be an additional asset to the reader of the manuscript.

Given the changes made in the discussion would be acceptable for me.

Thu 24 Aug 2023

Decision on Article NBME-22-2014D

Dear Dr Crooks,

Thank you for your revised manuscript, "Engineering the in vitro microenvironment for positive selection of stem cell-derived, Class I MHC-null, antigen-specific T cells". Having consulted with Reviewers #1, I am pleased to write that we shall be happy to publish the manuscript in *Nature Biomedical Engineering*.

We will be performing detailed checks on your manuscript, and in due course will send you a checklist detailing our editorial and formatting requirements. You will need to follow these instructions before you upload the final manuscript files.

Best wishes,

Valeria

Dr Valeria Caprettini
Associate Editor, Nature Biomedical Engineering

Reviewer #1 (Report for the authors (Required)):

I appreciate the authors' responses to my questions and suggestions. The manuscript has been improved and now contains substantial data and insightful discussions.

Rebuttal 1

April 10, 2023

Responses to Reviewers' Comments

We appreciate the thoughtful and constructive comments by the reviewers. We have thoroughly revised the manuscript and added detailed information/discussions to address the comments and concerns. As suggested by the reviewers, we have provided extensive additional data within the manuscript, and in some cases, we have supplied data within the responses to minor comments. We would of course be happy to include these data in the manuscript if felt to be necessary.

We have begun by highlighting the general responses that are relevant to questions raised by more than one reviewer:

General Responses:

#1. Further Characterization of the T cells produced (Reviewers 1, 2 and 4)

In a new **Figure 4**, we provide thorough molecular characterization using scRNA-seq of sorted CD8 $\alpha\beta$ ⁺ (SP8) mature T cells from the following sources:

- ATOs derived from the following type of PSCs:
 - RAG1/RAG2/B2M knockout (i.e., the experimental Triple KO aka “TKO” line) expressing the 1G4 TCR transgene, and generated by the modified hDLL4-A02BI stroma (i.e., stroma engineered to express Class I MHC (HLA-A*0201), plus human hB2M and ICAM1)
 - Unmanipulated (aka WT) PSCs cultured with standard hDLL4 stroma
 - WT PSCs expressing 1G4 cultured with standard hDLL4 stroma
 - WT PSCs expressing 1G4 cultured with hDLL4-A02BI stroma
- Peripheral Blood (PB)

Additionally, we compare our scRNA-seq data to publicly available data from thymic SP8 T cells as well as PB datasets, which include PB monocytes and PB NK cells. In summary, this new transcriptional characterization shows that SP8 T cells produced in ATOs in all PSC types are highly similar to each other ($r=0.98-0.99$) and to conventional SP8 T cells from thymus ($r=0.94-0.95$) and PB ($r=0.94-0.95$).

#2. Functional comparisons with alternative T-cell-induction-and-selection methods previously reported (in particular methods leveraging CD3 stimulation) (Reviewers #1 and #4).

Although direct comparison of our model to other approaches is not possible, we have adapted protocols developed by two prominent groups in the field to generate CD8 $\alpha\beta$ ⁺ T cells from DP precursors (the Kaneko and Sadelain groups). We have now included comparisons between the CD8 $\alpha\beta$ ⁺ cells generated directly by positive selection in ATOs and those generated using the adapted methods. After rigorous analysis of their published work, we find that direct comparisons using the various methods used by these groups over the years raise some challenges as described below.

For the first stages of differentiation, all PSC-T cell protocols including our own involve some form of mesoderm induction (\pm embryoid bodies), followed by various ways to induce hematopoietic differentiation. All **non**-ATO systems then isolate CD34⁺ cells from hematopoietic cultures, and induce T cell commitment and differentiation to the DP stage using notch ligands (hDLL1 or hDLL4)-either plate or bead bound or presented by stromal monolayers-and cytokines. None of these methods are consistent between groups and they also vary within the same group from one paper to the next. Methodological details of cultures are also often incomplete but almost all are serum dependent and many use OP9 stromal monolayer cultures creating further issues for reproducibility across labs.

Given the variability of these T cell induction systems, we have focused our comparisons on the generation of CD8 $\alpha\beta$ ⁺ T cells, comparing those that are generated spontaneously by as the dominant population in standard ATOs (>75% of all 45+ cells are SP8 in ATOs by week 6), to CD8 $\alpha\beta$ ⁺ cells generated in ATOs using other protocols that require stimulation of DPs. The method of stimulation of DPs also varies across publications. Our chosen comparison conditions are described in the papers listed below, some of which were also suggested by the reviewers.

Citation	First stage: DP stimulation	Second stage: T cell expansion
Minagawa et al., Cell Stem Cell , 2018	Anti-hCD3 AB (Clone OKT3)	PHA/Irradiated PBMCs
Wang et al., Nat Biomed Eng , 2021	Anti-hCD3 AB (Clone OKT3)	PHA/Irradiated PBMCs
Stegen et al., Nat Biomed Eng , 2022	3T3-CD19-41BBL(CD137L)	3T3-CD19-41BBL(CD137L)

As suggested by the reviewers, we included the use of CD3/28 beads as well as anti-CD3 antibody (OKT3) to activate DPs in our comparisons. Additionally, we also included the use of our artificial antigen presenting cells (aAPCs), which is similar

(but not identical) to the antigen-specific stimulation of CAR+ DPs by adherent target cells used by the Sadelain group and the company Fate Therapeutics (although, again, details of published protocols are incomplete).

All of these protocols require isolation of DPs from the culture prior to stimulation. This is because the other T cell differentiation methods generate a large component of DN and NK cells, which if present during stimulation, also respond to the growth factors present and kill the DPs. In contrast, DPs dominate ATO cultures by Week 3, and together with SP8 comprise >80% of output (see response to Reviewer 2, minor comments #6).

Following DP stimulation, a CD8 $\alpha\beta$ ⁺ T cell phenotype was assessed after 7 days of “activation” and again after 2 additional weeks of expansion (21 days total) (for schema see **Supplementary Figure 9A**). The 2-week expansion protocols ranged from continuation of CD3 stimulation with mitogen stimulation via PHA versus antigen-specific stimulation through aAPCs. Interestingly, in our hands, CD3 agonist-based selection methods failed to produce cell expansion, and did not generate sufficient CD8 $\alpha\beta$ ⁺ T cells for further characterization (**Supplementary Fig. 9D**). We found that the use of initial activation of DPs by aAPCs generated the highest frequency of CD8 $\alpha\beta$ ⁺ T cells.

We noted an important difference in phenotype of the SP8 cells generated using the above protocols for DP stimulation, compared to our method that generates CD8 $\alpha\beta$ ⁺ T cells directly in the ATOs. Both methods produced CD8 $\alpha\beta$ ⁺CD45RA⁺ cells; however, CD62L was absent in the DP stimulated cells (**Supplementary Fig. 9B**), whereas CD62L was consistently expressed in a majority of the ATO-derived CD45RA⁺CD8 $\alpha\beta$ ⁺ T cells (**Fig. 2C, Fig. 3F, Supplementary Figs. 3, 6, 7**). We note that in normal *in vivo* T cell differentiation, the CD45RA⁺CD62L^{neg} phenotype is seen in terminally differentiated “effector memory cells re-expressing CD45RA” (T_{EMRA}), while the CD45RA⁺CD62L⁺CD27⁺CD28⁺ phenotype as generated in ATOs is consistent with naïve, mature T cells and stem cell memory T cells (T_{SCM}).

After a further 14 days expansion with either aAPCs or PHA, the “T_{EMRA}-like” phenotype persisted in the cells generated through DP stimulation (**Supplementary Fig 9C**). In contrast, CD8 $\alpha\beta$ ⁺ T cells harvested directly from ATOs (both WT and TKO) acquired the CD45RO⁺CD45RA^{neg}CD62L^{neg} phenotype of effector memory T cells (T_{EM}) (**Supplementary Fig. 9E**). While detailed functional studies will be required in future studies to determine if the phenotypes of DP stimulated vs ATO-derived cells define true terminally differentiated “T_{EMRA}” cells vs differentiation of true T_N to T_{EM} respectively, we feel the clear phenotypic differences are worthy of note. We have added these points to the Discussion.

In an attempt to aid the review, we have listed changes to previous figure numbers and their new numbering as follows:

Previous Fig. #	New Fig. #
1	Same
2	Same
3	Same
N/A	4
4	5
5	6
6	7
Previous Supplemental Fig. #	New Supplemental Fig, #
N/A	5
5	6
6	7
N/A	8
N/A	9

Our revisions to the manuscript text are underlined and changes to Figures and supplementary information **are noted with a vertical line to the left of the page**.

In the responses below, the reviewers’ comments are reproduced in blue italic font, and our responses are in regular black font.

Reviewer #1

... there are some concerns that only limited combinations of TCR and HLA have been investigated in the study, the observed phenomenon itself is within the range expected from previous murine experiments, and the missing comparison between the generated T cells and existing T cells induced by different methods in vitro/vivo to evaluate therapeutic efficacy.

Major comments.

1. For a better understanding of the impacts of genetic modification on ESC and MS5hDLL4, additional analysis such as RNA expression and TCR-mediated signaling should be applied to differentiating CD4/8 T cells, differentiated CD8 T cells, and expanded CD8 T cells those induced by conditions of WT/MS5hDLL4, WT+TCR/MS5hDLL4, DKO+TCR/MS5hDLL4, TKO+TCR/MS5hDLL4, TKO+TCR/MS5hDLL4-A02BI. Especially, expression level and kinetics of T-cell differentiation-related transcriptional factor and strength and kinetics of TCR-downstream signaling among the T cells from different conditions will be informative to make authors claims further be confirmed.

Response:

As mentioned above in **General Responses #1**, we now provide scRNA-seq characterization of the SP8 T cells produced in ATOs from four different PSC-derived lines. No significant differences were seen between the global gene expression or T cell specific genes.

2a. The observation of CD8 T cell differentiation failure in ATO under HLA-TCR mismatch setting and its rescue by MS5hDLL4-A02BI is quite interesting and should be further evaluated. In general, TCR stimulation and expansion of matured T cell induces further differentiation to terminal effector T cell via effector memory T cells. From that point, it is worthy to clarify contribution of ICAM-1 transduction for expansion of T cells keeping naïve-like phenotype. Especially, how TCR-signal intensity and duration changed after ICAM-1 transduction; monitoring them during ATO differentiation would be great, but difficult, so a simple co-culture system with MS5hDLL4-A02B or MS5hDLL4-A02BI and TKO + TCR DP- T cells would be useful to evaluate TCR-signal intensity and duration. It would also be interesting to see if there is a dependence or threshold to ICAM-1 expressed on MS5hDLL4. besides, evaluation of memory/effector-related gene expression in T cells differentiated with/without ICAM and functional experiments such as those performed in Fig 5 will provide a better understanding of contribution of ICAM-1.

Response:

In addition to the originally submitted phenotypic and cell growth data comparing TKO-PSC derived ATOs using each of the engineered stromal lines (hDLL4-HLA-A*0201 ± hB2M ± ICAM1) (**Supplementary Fig. 6, 7A-C**), we have expanded these data to include flow cytometry data of effector/memory-related genes as well as data with just the inclusion of ICAM1 (hDLL4-ICAM1). We see significant improvement in cell output with the addition of ICAM1 (**Supplementary Fig. 7C**), and in WT lines the expression of CD62L appears to be higher with the addition of ICAM1 in the murine stromal lines. (**Supplementary Fig. 6A,E, Supplementary Fig. 7A-B**). However, we do not feel that the current data set definitively proves that the naïve phenotype is maintained in the presence of ICAM1 and the rigorous experiments required to explore this are beyond the scope of the current work.

We also provide new data demonstrating cell surface expression of hDLL4, Class I MHC, hB2M and ICAM1 in the engineered stromal lines (**Supplementary Fig. 5**). While the role of ICAM during expansion and impact of ICAM1 on TCR signal intensity are very interesting questions, our goal in this report was to enhance positive selection.

*2b. Also, the experiments presented in the paper are based only on the expression of HLA-A*02:01 in combination with the same HLA-restricted NY-ESO-1 TCR, and it would be desirable to confirm reproducibility with other HLA/TCRs. For example, using the HLA-A2-restricted WT1-TCR or using the HLA-A*24:02 transduced MS5hDLL4 and the HLA-A24-bound WT1-TCR.*

Response:

We have provided new data from experiments using the ESI017 TKO line and the HLA-A*0201-restricted F5 TCR recognizing MART1₂₇₋₃₅ peptide (**Supplementary Fig. 8**). These results are consistent with our 1G4 TCR experiments, i.e., CD8 $\alpha\beta$ ⁺ T cells were only produced from TKO cells cultured with the hDLL4-A02BI stroma (**Supplementary Fig. 8A**) and the CD8⁺ cells produced expressed CD45RA, CD62L, CD28, CD27, and CCR7 (**Supplementary Fig. 8B**).

We have also included data using the H1 (A0201+) line, edited to create TKO PSCs and transduced with the 1G4 TCR (H1 TKO ESO), again showing that positive selection of CD8⁺ T cells with the A02BI stroma (**Supplementary Fig. 7D**).

3. Although the authors wrote in discussion part that “In future studies, it would be interesting to compare the function of T cells produced with antibody stimulation of the TCR with those induced through cellular presentation of cognate MHC in the ATO”, experiments to evaluate effector function and therapeutic efficacy of ATO-derived T cells should be performed in this study in comparison with 2D culture induced T cells (with OP9 feeder and/or feeder free cultures) and primary T cells. Although animal experiments indicated a trend of tumour growth inhibition in the initial stages, overall survival rate is essential for discussing the therapeutic effect. In addition, PK data after transplantation is desirable to be obtained.

Response:

Please see our description of new data in **General Responses #2** above. In brief we have compared cell phenotypes generated by various published protocols for DP stimulation and expansion.

Minor comments (Reviewer 1)

1. The differentiation process in ATO seems depends on survival of MS5hDLL4 cells to give appropriate magnitude of Notch- and TCR-signaling at appropriate timing. Authors are requested to show data how long MS5hDLL4-A02BI survive in ATO to give Notch- or TCR-signaling.

Response

In our previous publications using the ATO system we showed persistence of the MS5-hDLL4 stroma in ATOs and expression of the DLL4-2A-GFP transgene for at least 6 weeks (Seet et al., *Nat Methods*, 2017).

We have submitted new data that demonstrates the expression of *hDLL4*, Class I MHC, *hB2M*, and *ICAM1* in the engineered stromal lines (**Supplementary Fig. 5**). Additionally, we have attached a figure here that shows the continued survival of engineered MS5 stroma through live cell imaging as well as flow cytometry.

2. The authors state that “The resulting SP8 T cells exhibited potent antigen-specific cytotoxicity in vitro as well as tumour control in vivo that was superior to PSC-derived T cells with intact endogenous TCR rearrangement.” However, it simply seems depends on tetramer-positive cell number difference between WT+TCR and TKO+TCR due to lower incidence of TCR mispairing in TKO+TCR cells. In case the authors think there should be any difference in T cell quality, the same number of tetramer positive cells should be used for the animal experiment to confirm it.

Response:

Equivalent levels of TCR transgene expression in WT+TCR and TKO+TCR cells was accomplished by isolating vector+PSCs after transduction based on expression of the BFP marker (co-expressed with 1G4 from a 2A sequence in the lentiviral vector, described in methods). Persistent and high cell surface expression of the transgenic 1G4 TCR in WT and TKO derived T cells was shown by FACS of the β 13.1 transgene (**Fig 6D, E**). Lower tetramer surface expression on WT+TCR cells relative to TKO+TCR cells (**Fig. 6D**) was most likely due to the mispairing in the WT+TCR of endogenous TCRs with the transgenic chains (mostly TCR α mispairing with β 13.1) as shown in the 10x sequencing in **Figure 5**. However, as the reviewer points out, although the *in vivo* experiments support this conclusion they do not prove it. Thus, we have changed our manuscript (Abstract, results and discussion) to reflect that our observed improvement in tumour control was “likely” due or consistent with to the absence of TCR mispairing.

The reviewer makes an interesting point about possible differences in T cell quality. However, as mentioned in **General Responses #1**, the very tight transcriptional correlation of the TKO and WT SP8 T cells (**Fig. 4C**) does not indicate any obvious functional differences. Unfortunately, we believe that the *in vivo* experiment suggested would be technically impossible as the MFI for tetramer⁺ cells is so different between the arms that matching not only cell number but tetramer staining levels on each cell would not be feasible.

3. About Fig. 2B, SP8 differentiation from CD4/CD8 DP-T appears to be delayed in H1 DKO+TCR condition. Authors are requested to explain the reason for it.

Response:

We agree that, relative to WT and WT+TCR lines, there is an apparent delay in the DKO+TCR lines; we note that this is also present in the TKO+TCR conditions. “Delay” could be considered more as relative inefficiency and is likely due to antigen scarcity as there is only one TCR available for selection in H1 DKO+TCR. The rapid differentiation of SP8 in RAG-

competent (WT) lines that express the transgenic TCR is likely due to the presence of endogenous TCRs broadening the range of selectable peptide-antigen combinations.

4. *Please indicate the expression levels of hDLL4, HLA-A*02:01 and ICAM in MS5.*

Response:

As mentioned in response 1, we provide new data demonstrating cell surface expression of *hDLL4*, Class I MHC, *hB2M* and *ICAM1* in the engineered stromal lines (**Supplementary Fig. 5**).

5. *About Fig. 5A, "TNFg" on Y-axis should be "IFNg"*

Response: Our thanks for catching this error. We have now corrected this figure, which is now **Figure 6A**.

6. *About Fig. 5G, T-cell reactivity to NY ESO aAPC IL2 IL7 stimulation seems very weak. Authors are requested to explain the reason for it and to evaluate proliferative reactivity of T cells by standard stimuli such as CD3/28 beads.*

Response:

In our hands, expansion of PSC-derived T cells is very poor with CD3 engagement methods, including anti-CD3 AB and anti-CD3/28 beads (**Supplementary Fig. 9D**). This is why we, like some others in the field use antigen-specific T cell expansion; *Stegen et al., Nat Biomed Eng, 2022* reported ~30-fold expansion from DP stimulation using adherent target cells; however, there is no data regarding SP8 T cell expansion after DP stimulation.

Reviewer #2

Major concerns:

1. *In the present manuscript, the authors used PSC (e.g. embryonic stem cells; ESCs) as the starting cell source. It is known that the use of stem cells has the disadvantage of raising ethical concerns. On the other hand, hiPSC offers the advantage of substantially decreasing ethical concerns, hence, bringing this work one step closer to the clinics. This point is of particular interest, as the authors state in the introduction the increased need of "off-the-shelf" T cell products for clinical translation. Whether ESCs or iPSCs are used later as a starting material, the authors should provide PoC using iPSC. Given the strong clinical introduction, the authors could also provide evidence, that the ESC lines used are applied in clinical trials.*

Response:

We understand and appreciate the issues raised by the reviewer regarding the use of ESCs vs iPSCs. However, we note that while a diversity of ethical and political concerns regarding hESC are likely to push the field toward iPSC-based products, clinical trials have proceeded with hESC-derived tissue derivatives. In our original paper describing T cell differentiation from PSCs in ATOs (Montel-Hagen et al., *Cell Stem Cell*, 2019), we found phenotypically similar T cells were generated with four hESC lines and a fibroblast derived iPSC line. Given the substantial differences between individual lines from the same type of source, in general we find little/no convincing biological differences between hESCs and iPSCs. Our goal in the current report was to develop a feasible engineering approach to induce positive selection in the presence of a single transgenic TCR and absent MHC Class I to produce mature stem cell derived PSC-derived T cells.

2. *To further improve/increase the novelty of this work, as the authors mentioned, additional strategies to overcome the alloreactive rejection must be applied. As the author already published a similar work before (or at least parts of it), adding additional mechanisms to avoid NK activation or deleting class II MHC would be extremely beneficial.*

Response:

We thank the reviewer for this important point, and we fully acknowledge that further work will be needed to produce a fully universal clinical T cell product. We believe that pursuing the many modifications required is beyond the scope of this current publication, and this is acknowledged in the last paragraph of our discussion.

3. *The underline goal of the study would be the development of suitable T cell products for clinical use (as mentioned also in the introduction). Having this in mind, the use of a murine stromal cell line does not follow that path. Given the strong introduction on the clinical use, the authors should elaborate on strategies to circumvent the use of murine stromal cells. As a consequence, maybe other "ATO" systems (coating, human cells (e.g. OP9?) etc.) should be developed to bring the approach really once step closer towards the clinics. Of note, I appreciate the fact that the MS5 cells resemble the current state-of-the-art for the ATO system, however at the current stage, it may be very much challenging to produce an off-the-shelf T cell product using the current ATO version. Thus, the scope of the current manuscript is rather be mechanistically*

than clinically. Along this line, more mechanistic on the e.g. T cell development should be provided to give some new insights into the developmental trajectories of human PSC-derived T cells.

Response:

As mentioned in the previous responses, the goal of our studies is not to provide a final clinical product but to overcome the biological hurdles of providing a mature T cell product. While our introduction begins with clinical relevance of the PSC-derived T cells as a whole, we shift our focus to the development of a method to circumvent the issues regarding positive selection, and our final sentence only claims that this a proof of concept. To strengthen the methodology and mechanism of the paper we have therefore added a large amount of new data including scRNA-seq data on the SP8 cells.

We accept the use of the MS5 line in the ATO will require additional regulatory hurdles but also note that the FDA standards (US FDA. Source Animal, Product, Preclinical, and Clinical Issues Concerning the Use of Xenotransplantation Products in Humans, Dec. 2016. <https://www.fda.gov/regulatory-information/search-fda-guidance-documents/source-animal-product-preclinical-and-clinical-issues-concerning-use-xenotransplantation-products>) do not preclude use of a murine (such as OP9 stroma) or human cell line (such as K562 cells) cultured with the clinical product, as long as the final clinical product is shown to be devoid of contaminating cells. Several clinical trials using such products have been permitted.

4. In this study, the authors in order to characterize the iPSC-derived TKO CD8SP T cells generated with the ATO system, performed TCR repertoire analysis and FACS analysis for specific markers, such as CD45RA, CD45RO, and CD62L. However, the authors should provide a more detailed characterization of the derived-cells, performing for instance transcriptomic profiling of the TKO CD8SP T cells. This might be helpful to evaluate possible differential regulation of genes due to the multiple genetic modifications.

Response:

We thank the reviewer for this suggestion. As mentioned above in **General Responses #1**, we now provide scRNA-seq characterization of the SP8 T cells produced from four different PSC-derived ATOs and from human thymus and PB. No significant differences were seen between the global gene expression.

5. Similar to point No#4, it would be beneficial to identify whether these cells express known factors important for their functionality and which subset of CD8SP T cells they resemble more. The authors should perform single-cell RNA seq analysis and subsequent trajectory analysis to further dissect and reveal the different developmental stages of the cells in the ATO system and possibly to compare it with normal thymic development. These are especially important when these cells are generated with the goal to be used as an “off-the-self” engineered T cell product for cellular therapies.

Response:

In the scRNA-seq data mentioned above we include global gene expression and specific gene expression of SP8 cells generated from WT and TKO iPSC with those from thymus and PB.

6. It is known that that the thymic microenvironment plays a crucial role in the epigenetic regulation of T cells during their development. The epigenetic modifications contribute to all stages of T cell differentiation and affect the cells’ functionality and stability. The authors should provide some data regarding the epigenetic status of the derived CD8SP T cells (e.g. methylation status of specific loci) also in comparison with their in vivo counterparts.

Response:

We thank the reviewer for this very interesting question regarding epigenetic modifications, but we believe it to be beyond the scope of this current study.

Minor concerns (Reviewer 2)

1. In the second sentence in the second paragraph of the introduction, the respective references are missing. Please add the appropriate literature.

Response:

Thank you, we have added the appropriate citations for the mentioned sentence in the introduction.

2. Full name of EMO is missing from the main text and appears only in the supplementary material. The author should add it in the main text as well.

Response:

Thank you, we have now added it. It now appears in the methods as well as the legends for **Supplementary Figure 2A** and **Figure 3A**

3. Figure 2D: The authors should describe the pre-gate they have done for the analysed populations.

Response:

The pre-gate for **Figure 2D** is shown in **Figure 2B**, the relevant information is described in the legend.

4. The axis labels do not state the fluorochrome that was used. This is necessary according to the journal's guidelines.

Response:

Thank you for this point. We have now stated the fluorochrome on all axis labels of flow cytometry figures.

5. It would be interesting to see photomicrograph images from the ATOs as well as the ATO-derived T cells. In the case of ATO-derived T cells, cytospin images would further increase the focus on the final product as well as enlightening the reader image about the cell product.

Response:

Thank you. Immunofluorescence images of sections of the ATOs can be found in our original PSC-ATO paper (Montel-Hagen et al., *Cell Stem Cell*, 2019), Figure 2D. Briefly, Montel-Hagen et al. 2019 shows representative whole-mount immunofluorescence analysis of GFP (marking MS5-hDLL4 cells), CD3, and DAPI. We have also included some live cell imaging images in response to Reviewer #1, minor comment #1.

6. It would be interesting to have the FSC x SSC plot to observe ROI of the expected PSC-derived T cells.

Response:

We have provided FSC x SSC plots to observe the ROI of the expected PSC-derived T cells taken from a hDLL4-A02BI ATO, which expresses mStrawberry in the stroma. The first plot is gated on single cells based on height and width of both FSC and SSC, and the DAPI gate is used to remove debris in the left part of the plot. Each subsequent level of gating is indicated with arrows as well as their corresponding markers above the plots. Non-T lineage CD45+ cells are suspected to be monocytes based on FSC-A x SSC-A.

7. Are there any contaminant cells? For example, an additional panel for several cell types (e.g. Macrophages) must be added. Cytospins should also add additional information in this regard.

Response:

As seen in the FSC/SSC plots of the DAPI- population in response #6, contaminant cells (i.e. murine stroma and NK cells) are <1% of all live cells and large dense cells falling outside the lymphoid gate on FSC vs SSC are likely monocyte/macrophages (<5% of all live cells).

8. Some FACS plots seem to be in need for additional compensation (e.g. Supplementary fig.2, 3, 5).

Response:

We reviewed the figures, and we are not clear on the reviewer's concerns. We would be happy to return to this point if we could have further clarification or more specific direction.

Reviewer #3

A highly interesting MS reporting the generation of monoclonal TCR transgenic human CD8 T cells from PSCs using an in vitro system (ATO) for positive selection. It is shown that PSCs can be manipulated to lack both endogenous Rag (thus preventing endogenous TCR re-arrangements) as well as b2m (with the perspective of protecting them from allorecognition in a therapeutic transfer setting). Furthermore, the MS describes crucial advances in the manipulation of the selecting mouse stromal component (expression of the selecting human MHC haplotype as well as ICAM). Finally, impressive data on the in vitro and in vivo functionality of these cells are presented. Together, the MS should be of outstanding interest to the readership of NBME, as it may pave the way towards overcoming major hurdles in the generation of T cell immunotherapies from allogeneic stem cells.

Minor technical criticisms or questions:

1. Fig 1: The TCR-less DP cells that transiently emerge from DKO cells are referred to as 'innate-like'. The authors may want to consider whether doing so solely on the basis of CD8alpha/alpha expression is justified. Probably fair to say that their cellular identity remains entirely obscure; not an issue.

Response:

Thank you for this point of clarification. In our use of the term innate-like we were trying to make that clear that the CD8 α ⁺CD4⁻ cells that arise early in differentiation are not generated through positive selection of DPs. We agree that the term innate-like is difficult to define. We have now rephrased to say that we did not consider this population of cells to appear to be through the process of positive selection due to the lack of detectable surface TCR and CD3 (Paragraph 1, Results)

2. Suppl Fig 2: Throughout Fig S2, it seems that mCD29 and CD56 are used in the same channel. The gating strategy then indicates further gating on mCD29-CD56⁺ cells. Shouldn't this be mCD29-CD56⁻?

Response:

We thank the reviewer for identifying this error and have corrected the figure.

3. Fig 5A: TNFg on the vertical axis!!?

Response:

We thank the reviewer for identifying this error and have corrected the figure (now shown as **Fig. 6A**).

4. One thing that may be interesting to the reader is whether the cells persist/expand in the in vivo tumor model (and how this compares to the WT+TCR setting). Purely optional, but the authors may have some data on this anyhow.

Response:

We thank the reviewer for this comment, but unfortunately, we did not perform the assays.

Reviewer #4

Major points:

1. Prevention of TCR rearrangement during T cell differentiation from PSCs by knocking out Rag 2 has been reported (Minagawa et al, 2018, Wang et al, 2021). RAG1/2 knockout is predicted to elicit the same effect. One key question is why TCR-null SP cells can be generated using the traditional OP9 system (Wang et al. 2021) but not via the ATO system? Will stimulation with anti-CD3 at DP stage also work in the ATO system? These are important questions to resolve to justify the engineering of stroma cells.

Response:

We thank the reviewer for these suggestions. We have performed new experiments to compare the DP stimulation required in other non-ATO systems to produce SP8 cells (**Supplementary Fig. 9**). Our complete response to this issue is found in **General Responses #2** at the beginning of this document

With regards to the reviewer's comments concerning TCR-null SP cells (Minagawa et al., *Cell Stem Cell*, 2018; Wang et al., *Nat Biomed Eng*, 2021), both publications had a functioning TCR prior to generation of SP T cells using the traditional 2D culture methods. In Minagawa et al., 2018 and Wang et al., 2019, the authors knocked out RAG2 in order to prevent further endogenous TCR rearrangements in their T-iPSCs, which already had a fully rearranged TCR pair. This TCR was re-expressed on the surface after re-differentiation into DP precursors, which were harvested from culture before DP stimulation was used to generate SP8 T cells. When using a non-T-derived iPSC, the authors transduced an exogenous TCR, similar to our approach, in order to restore TCR expression after T cell differentiation. Thus, in both cases, their SP T

cells were not TCR-null. The only instance where they had TCR-null SP T cells was in Figure S2D of Wang et al., 2021, where they knocked out both TRAC in primary T cells from peripheral blood; thus, these mature cells did not need to go through positive selection.

2. The ATO system has been shown to produce more mature T cells than the OP9 system. Do the TCR-null cells rescued by engineered stroma cells still display a mature phenotype? This is important because the differentiation of DKO/TKO PSCs into T cells via HLA rescue seems to be less efficient than WT cells (Fig 2B, 3B). A refined molecular characterization (eg. RNA-seq, ideally single cell) of TKO TCR PSC-T vs WT TCR PSC-T cells would address this question and add considerably to the characterization of the T cells produced.

Response:

Thank you for the suggestion of a refined molecular characterization. We have now provided this as scRNA-seq in new **Figure 4** and our full response to this question is in the **General Responses #1** at the beginning of this document.

3. Since hypoinmunogenic antigen-specific PSC-T cells have been reported (Wang et al. 2021), it would be extremely helpful to compare this new strategy with the previously reported system. For functional assays, it is conceivable that TKO PSC-T cells outperform WT PSC-T cells due to the lower immunogenicity. The key comparison should be between TKO TCR PSC-T cells generated using previously established methods and the cells generated using the ATO system with the engineered microenvironment.

Response:

Thank you for the suggestion. Our attempts to compare the ATO system with the positive selection method used by other groups are described in detail under **General Responses #2**.

Minor point:

Mechanistically, RAG is also required for pre-TCR, leaving the question for why cells end up blocked at the DP stage, not DN stage? A citation to the situation in murine RAG KO or human Omenn Syndrome patients would be informative.

Response:

We thank the reviewer for their insight into this question. The block at the DP stage has been characterized by Luigi Notarangelo's group in the following publications: Bosticardo et al., *Blood Adv*, 2020; and Gardner et al., *J Clin Immunol*, 2021. Both studies found that T cell differentiation was blocked at the DP stage in the absence of functional RAG-complex proteins. In humans, it seems that pre-TCR is not a prerequisite for differentiation into the double positive stage, as defined by surface expression of CD8⁺CD4⁺. Our manuscript has been updated to reflect these citations, which appear as citation #17 and 18 (Paragraph 1, Results).

Rebuttal 2

June 16, 2023

Response to Reviewers' Comments

We appreciate the thoughtful and constructive comments by the reviewers. As suggested by the reviewers, we have included additional analyses in our new Supplemental Figure 10. These changes include scRNA-seq data showing UMAP analysis with PB NK and monocytes removed (Supp figure 10A), UMAP showing *CCR7* and *SELL* expression (Supp figure 10B) as well as survival data from our *in vivo* experiment (Supp Fig 10C). Below, you will find our responses to the outstanding issues with the manuscript. Our revisions to the manuscript text are underlined and changes to the Supplemental Figure 10 are noted with a vertical line to the left of the page.

Reviewer #1

The revised manuscript by Chong et al. clarifies and discusses the proximity among iPS-T cells from various condition by scRNA analysis and the difference in quality of DP cells by comparison with the previously reported 2D culture, although to a limited extent. Although we feel that the manuscript has improved in some respects, there are still some points that remain a concern through peer review and need to be addressed.

Major points

Point #1.

To my first question, my proposed analysis of differentiating T cell samples was not performed (see below). Since the paper focuses on the importance of TCR and HLA in differentiation, and authors stated as "Transgenic signals required to rescue positive selection were supplied by separate cellular sources; T cell precursors from edited PSC expressed a single TCR, and a murine stromal line provided cognate human MHC and other critical T cell development signals." in the abstract, I would like to see the changes in gene expression during differentiation, especially the differences and fluctuations in TCR signaling-related molecules.

Responses to Point #1

We respectfully argue that the reviewer's request falls out of scope of our studies as it would not provide data of central relevance to the message, and it is also technically unfeasible for the following reasons:

- 1.1 Scope:** The goal of our work was to enable production of mature SP8 T cells from PSC lines that lack all endogenous TCR expression and Class I MHC expression (i.e. TKO lines). We believe the data we have provided convincingly show that engineering of PSC and accessory cells achieve this goal through exogenous provision of TCR and HLA components known to be critical for positive selection. Analysis of "*Expression level and kinetics of T-cell differentiation-related transcriptional factors*" performed on cells from the many experimental arms throughout ATO differentiation would provide little if any additional information to the fundamental conclusions of our studies.
- 1.2 Relevance to primary message:** Positive selection of TKO derived cells can only be studied in two experimental arms (*TKO+TCR/MS5hDLL4-A02B* and *TKO+TCR/MS5hDLL4-A02BI*) which differ only in the presence of *ICAM1* expression in the ATO accessory cells. *ICAM1* enhances yield of SP8 T cells (not differentiation) and is an additional (but not primary) point of the work; *ICAM1* is not mentioned in the abstract. Mechanistic studies on the role of *ICAM1* in TCR signalling have been reported by others as cited in our manuscript (Ma et al. *Science Advances*, 2022). Even if these analyses were feasible in the ATO system, they would add little to the primary point of the paper.
- 1.3 Feasibility:** Analyses to show "*Strength and kinetics of TCR-downstream signalling*" throughout differentiation are not feasible in the ATO system. Transcriptional analysis is not sensitive enough to determine strength of signalling. Classic signalling studies are typically performed using pulsed activation signals applied to homogeneous cell lines and require time course analysis of phosphorylation over seconds to minutes. Differentiating cells in the ATOs are dynamically changing and heterogeneous, and the TCR signalling events are constantly provided by the stroma. In addition, signalling will rapidly change during harvesting and disaggregation of cells from each ATO.

Point #2

On the other hand, the authors present scRNA data for T cells amplified after induction of differentiation under different conditions, leading to the conclusion that there are no differences in the final T cells. If these T cells are to be considered for therapeutic use, it is important that they have the same characteristics regardless of the method of production.

Responses to Point #2

- 2.1.** We have shown by flow cytometry and scRNA-seq analysis that the PSC-derived T cells produced using genetic modifications of the PSC and accessory cells of the ATO system, are phenotypically and transcriptionally highly similar to unedited (WT) PSC and to primary CD8 T cells from thymus and peripheral blood (PB). We do not, however, state that there are "no differences", and in fact would not expect a synthetic, culture-derived cell to be identical to a primary cell population.
- 2.2.** The ideal characteristics for a T cell product harvested from PB remain controversial in the field; however, there is growing pre-clinical and clinical data, particularly from the Stanley Riddell group that less mature T subsets in PB are more potent on a per cell basis than those with a T_{EM} phenotype. Our study is a proof of concept that we can produce

cells with many characteristics of T_N after extensive editing and re-engineering to provide signals of positive selection. Future clinical studies will be required to make a direct comparison of cell types derived from PB and PSC.

Point #3

In Figure 4A, the authors compare peripheral blood NK, monocyte, SP8, thymus SP8, and their own SP8. Why include NK, monocyte in the comparison? By including cell types that are clearly distant from each other, wouldn't it obscure the differences within the SP8 population? In fact, since several differentially expressed genes can be identified in Fig. 4B, would different clusters emerge if the analysis is limited to SP8 only?

Responses to Point #3

The reasoning behind including other cell types in our analysis is to show that we are generating T cells and not other cell types (such as NK or Monocytes) that appear commonly in non-ATO culture systems. We now provide UMAP data (**Supplementary Figure 10A**) showing that removal of both NK and Monocytes yields the same clustering of SP8 T cells from ATO and primary sources.

As there are minor differences in specific genes across each experimental arm shown in Figure 4B, we also performed pairwise comparisons in global gene expression in Figure 4C: these show that there is a high correlation between our ATO-derived SP8 T cells (>0.98) and those isolated from primary tissues sources (PB and thymus) (>0.94) irrespective of whether monocytes and NK cells are included in the analysis.

Point #4

I am concerned about the low CCR7 expression in Fig4B. Has the expression been lost due to expanded culture? And what about the expression of the SELL encoding CD62L, a characteristic of ATO-derived SP8?

Response to Point #4

FACS analysis of *SELL* expression shows a clear 30-50% of SP8 cells from ATOs are CD62L+, consistent with our previous publications (Supplementary Figure 6). We now provide data (**Supplementary Figure 10B**) showing *SELL* and also *CCR7* expression via UMAP analysis in SP8 cells from primary sources (PB and thymus) vs PSC-ATO cells.

Point #5

Regarding my second question, the authors have stated that the mechanism of ICAM1 involvement is outside the scope of this manuscript. Since the manuscript demonstrates the importance of TCR-HLA signaling in differentiation, the mechanism of ICAM1 involvement should be clarified as authors expected. We feel that one methodology might be to evaluate downstream gene variation from scRNA-seq data during differentiation, and another methodology might be to evaluate signal intensity enhancement downstream of the TCR under co-culture, respectively.

Response to Point #5

We refer to Response #1.2 above regarding *ICAM1* studies and Response #1.3 regarding feasibility of signalling studies. scRNA-seq of the final product from our differentiation of SP8 T cells from edited PSCs demonstrate that we generate SP8 T cells that are very transcriptionally similar whether in the presence or absence of *ICAM1*, and even with *RAG1/2* deletion.

Point #6

The authors have taken reproducible data of the importance of TCR and MHC for differentiation using different TCR-MHC combinations. However, as shown in Fig. S8A, only 21.7% of SP8 is present in F5 TCR at 7 weeks; if the proliferative capacity of F5 TCR SP8 is also 10-fold per cycle as shown in Fig. 6G, it may be difficult to obtain cells at the level of clinical application. This point should be discussed.

Response to Point #6

We agree that the F5 data suggest that differentiation efficiency will likely be specific to certain TCRs and have added this important point to the discussion.

Point #7

In response to my third question, a comparison was made between DP cells obtained from 2D culture and ATO-derived DP cells, although limited. However, according to a previous report, the PHA-stimulated amplification method requires the use of PBMCs, but this is not mentioned in the manuscript, and if true, it is not a correct comparison with the previous report. In fact, this may be the reason why the OKT-PHA-PHA group does not increase after 7 days.

Response to Point #7

We attempted to replicate the alternative PSC differentiation methods that use DP stimulation but agree that the cause of poor expansion from PHA stimulation is likely due to the absence of PBMCs (which are typically used as APCs in a polyclonal TCR setting). We note that we had extremely poor yield after activation simply using the OKT3 antibody (before

the PHA stim). However, as we expressed a single specific TCR in the TKO PSCs, we were able to also compare antigen-specific expansion via aAPC in ATO-derived vs DP induced SP8s (as per the Sadelain protocol), replacing the need for the use of PBMCs.

Point #8

The authors wrote in the abstract that "Edited T cells exhibited superior tumor control in vivo likely due to the absence of TCR mispairing." However, the transplantation of 1×10^7 TKO SP8 into 5×10^5 NALM6 failed to suppress tumor growth. Survival data should be provided to compare the therapeutic effect with the previously reported Primary T. Hopefully, mice treated with Primary T under the same conditions will be set up, and both IVIS data and survival data are needed. If the authors cannot show a sufficient treatment effect from these data, it seems to me that the treatment effect can be de-emphasized and Fig. 6 can be included as a supplement.

Response to Point #8

We now provide survival data (**Supplementary Figure 10C**) which shows only TKO cells have significantly better survival than untreated (PBS) control. As shown in main **Figure 7**, we found that a single injection of frozen and thawed TKO SP8 T cells significantly improved suppression of tumour growth over the course of the experiment compared to WT SP8 T cells. Although the TKO cells did not completely eradicate tumour, we note that most groups use multiple T cell injections in their *in vivo* studies. We believe these data are important to include in the main figure as they provide support for the hypothesis that mispairing in the WT has functional consequences.

Minor points

The bottom two groups in Fig6F may be a misdescription of MART aAPC. Also, "Expanded WT+TCR and TKO+TCR T cells also displayed potent antigen-specific cytotoxicity in vitro at an effector-to-target ratio as low as 1:32" in the manuscript.

However, in Fig6F, 1:32 appears to have little effect, and a ratio of 1:8 to 1:4 would be more appropriate.

Response to Minor Point

Thank you for pointing out the confusing way this is written; we have clarified our statement of potency.

Reviewer #2

After evaluating the author's reply to my comments raised, I very much appreciate the changes made.

However, I would like to focus on my previous question #2.

Point #1

I would like to highlight that, even though I can understand the position of the authors, I do not share the opinion that this comment made is outside the scope of the manuscript. Of note, the authors highlight several times in their manuscript the beneficial use of "allogenic" cells, while also applying ESC, which later on are indeed applied in an allogenic fashion.

Furthermore, the authors claim in the introduction section that "Cell-based immunotherapies in which autologous T cells have been engineered to express chimeric antigen receptors (CARs) and antigen-specific T cell receptors (TCRs) have produced impressive clinical responses in patients with otherwise treatment-refractory diseases.", therefore, bringing solutions to overcome this critical challenge is of most importance. Hence, I would like to reiterate the importance of this question and ask for additional data that would indicate the possibility to use this model in an allogeneic setting, highlighting that the system introduced is working in allogenic setting.

Response to Point #1

We thank the reviewer for this point and agree that while this is an important additional question for a clinical product; however, we do believe the inclusion of immune-evasion mechanisms fall out of scope for our current report. Other groups have dedicated their pursuits towards this goal and it is not yet clearly known what the ideal immune evasion characteristics for a clinical product will be at this stage, nor what are the correct surrogate assays to use to predict the clinical response. We have included this point in the discussion of our manuscript.

Reviewer #3

I was very positive in my initial assessment of the MS and didn't raise major concerns. The authors have done a great job addressing most of the concerns raised by the other reviewers. The inclusion of additional data has even further improved this very interesting MS.

Response: We thank the reviewer for their comment.

Reviewer #4

The authors have responded to each of our prior numbered points, as indicated below:

Point #1.

Prevention of TCR rearrangement during T cell differentiation from PSCs by knocking out Rag 2 has been reported (Minagawa et al, 2018, Wang et al, 2021). RAG1/2 knockout is predicted to elicit the same effect. One key question is why TCR-null SP cells can be generated using the traditional OP9 system (Wang et al. 2021) but not via the ATO system? Will stimulation with anti-CD3 at DP stage also work in the ATO system? These are important questions to resolve to justify the engineering of stroma cells.

- Authors performed new experiments to compare the DP stimulation required in other non-ATO systems to produce SP8 cells (Supplementary Fig. 9). In these experiments, authors included the use of CD3/28 beads, anti-CD3 antibody (OKT3), and artificial antigen presenting cells (aAPCs) for the SP induction and cell activation/expansion. The results show that though previously reported strategies can induce SP production from week 3 ATO DP cells, only CD8 SP cells collected directly from ATOs display a naive/stem cell memory phenotype. These data suggest the superiority of using engineered ATO system to produce class I MHC-null TKO PSC-T cells

Response: We thank the reviewer for their comment.

Point #2.

The ATO system has been shown to produce more mature T cells than the OP9 system. Do the TCR-null cells rescued by engineered stroma cells still display a mature phenotype? This is important because the differentiation of DKO/TKO PSCs into T cells via HLA rescue seems to be less efficient than WT cells (Fig 2B, 3B). A refined molecular characterization (eg. RNA-seq, ideally single cell) of TKO TCR PSC-T vs WT TCR PSC-T cells would address this question and add considerably to the characterization of the T cells produced.

- Authors performed scRNA-seq to compare the molecular features of TKO PSC-T cells generated using engineered ATO vs. traditional ATO PSC-Ts and thymic/PBMC SP8 T cells. The results show that TCR-null cells rescued by engineered ATO stroma cells display a similar phenotype than WT ATO PSC-T cells and are similar to conventional primary SP8 T cells, not NK or monocytes.

Response: We thank the reviewer for their comment.

Point # 3.

Since hypoinmunogenic antigen-specific PSC-T cells have been reported (Wang et al. 2021), it would be extremely helpful to compare this new strategy with the previously reported system. For functional assays, it is conceivable that TKO PSC-T cells outperform WT PSC-T cells due to the lower immunogenicity. The key comparison should be between TKO TCR PSC-T cells generated using previously established methods and the cells generated using the ATO system with the engineered microenvironment.

- Authors provided a comparison of engineered ATO vs. existing strategies via immunophenotypic characterizations, but did not perform additional functional assays or in vivo assays to address the concern. Reviewer #3 also suggested similar experiment. It is understandable that in vivo experiments are time consuming, and it is challenging to obtain results within the revision time window. However, such experiments are important to justify the new method of using engineered stroma cells in the ATO system, as alternative strategies for SP induction, such as using CD3/28 beads, might be more suitable for large scale manufacture.

Response Point #3

The experiments proposed by the reviewer are a follow-on from the new data that we added generating SP8 T cells from alternative culture systems that require direct stimulation of isolated DPs. While we agree that these questions are of great interest to the field, we respectfully argue that the time and costs involved in further *in vivo* studies are not warranted as the methods from other groups are impossible to exactly reproduce.

Minor point:

Mechanistically, RAG is also required for pre-TCR, leaving the question for why cells end up blocked at the DP stage, not DN stage? A citation to the situation in murine RAG KOs or human Omenn Syndrome patients would be informative.

- Authors have addressed the question.

Response: We thank the reviewer for their comment.

Rebuttal 3

July 30, 2023

Responses to Reviewers' Comments

We appreciate the additional thoughtful and constructive arguments provided by the reviewers. As suggested by Reviewer #1, we have now included additional analysis of the SP8s produced *in vitro* in **Supplementary Figure 10** of this latest revision of the manuscript. These changes include UMAP projection of the integrated SP8 populations from different sources, which was calculated using the optimal number of PCs and clustered using IKAP analysis of the integrated dataset, and modified our feature plots (**Supplementary Figure 10D**) to use the updated UMAP projection. We have included all of the suggested changes in the manuscript as suggested. Our latest revisions to the manuscript text are underlined and noted with a vertical line to the left of the page.

Below, you will find our responses in blue to the two reviewers' comments, which appear in black. We have used the same numerical points as they appeared in the responses in the second revision of our manuscript.

Reviewer #1

I would like to thank authors for their thoughtful responses to additional questions. I also appreciate the additional data provided. I feel that the manuscript has improved, but we do have additional questions.

#1, I am aware of the authors' comments (1.1-1.3) regarding the analysis of gene expression over time in ATO and the analysis of the effects of ICAM1, and suggest that point in 1.3 be discussed in the Discussion as a limitation of the ATO analysis.

We thank the reviewer for their comments. Point 1.3 (Analyses to show “Strength and kinetics of TCR-downstream signalling” throughout differentiation are not feasible in the ATO system) has now been addressed in the Discussion (**Page 6**).

#3, I suggest the authors reconsider their clustering method, as they have shown a single cluster in green on all individual UMAP panel as a result of simple extracting of SP8s. It is odd that there is only a single cluster in the sample, and indeed there is a certain heterogeneity in each sample when looking at SELL and CCR7 expression. The correct approach would be to integrate the results from the six SP8 samples and then perform a UMAP clustering analysis to determine which samples contribute to which clusters and to what extent. That analysis can more reasonably and visually discuss the proximity and differences of the 6 cell types.

In our second revision, we chose to use single colours in the main figure in order to depict the global gene expression of different cell types. This combined dataset was not suitable for integrated analysis as some of the curated datasets for this analysis did not include shared cell populations (i.e. Monocytes and NK only).

In response to the reviewer's suggestions, we have now modified **Supplementary Figure 10** to include UMAP projections calculated based only on the integrated SP8 datasets, with optimal PCs and clusters identified through IKAP analysis (**Supplementary Figure 10A and Methods Page 11**). Interestingly, we found that integration produced a tighter cluster of cells in the dimensionally reduced UMAP space, and only 3 subclusters were identified from within the SP8 population using the IKAP algorithm, which expands or collapses clusters using differential gene expression.

Additionally, “to determine which samples contribute to which clusters and to what extent” we have provided the frequency of each cluster within each cell source of SP8 in **Supplementary Figure 10B**, and Pearson's correlation of global gene expression for all pairwise combinations between each cluster with dendrogram and hierarchical clustering analysis in **Supplementary Figure 10C**. These data reveal that the 3 clusters are present in each SP8 source and that they are highly similar ($R > 0.998$). For consistency we have also updated our feature plot for CCR7 and SELL in **Supplementary Figure 10D** with the new UMAP projection. The Results section (**Page 4**) has been updated to incorporate these changes.

#7, I understood the situation. I agree that it is not always easy to fully reproduce the protocols of other institutions. However, since this is an OKT3-PHA stimulation protocol that does not include PBMCs, the author's description of “SP8s generated through activation of ATO-DP precursors through CD3 engagement adopted a CD8ab+ CD45ROCD45RA+CD62L- phenotype (Supplementary Fig. 9B), but did not expand with further stimulation (Supplementary Fig. 9C-D).” Needs to be supplemented with the fact that “in the absence of PBMCs”.

We thank the reviewer for their comment and have modified the Results to reflect the suggested change (**Page 3**).

#8, I would like to thank to authors for the additional data. I agreed that TKO compared to the no treatment group statistically significantly prolongs survival by log-rank test over WT. However, I question whether the 2.4 day increase in mean survival has clinical significance. In addition, if the authors are truly interested in the utility of antigen-specific iPS-T

cells generated from ATO, it is important to take a comparative approach with allogeneic T cells, rather than much emphasizing their *in vivo* efficacy to non-treatment group, I believe that at least these issues should be discussed seriously rather than emphasizing the *in vivo* efficacy.

We are in agreement that the increase in mean survival does not necessarily reflect clinical response, and that clinical trials would be required to determine whether these responses and survival data are of clinical significance. As the focus of this manuscript is to define a method to induce positive selection in the absence of TCR rearrangement and Class I MHC expression, we would like to underscore that the importance of the *in vivo* data was to demonstrate the effect of TCR mispairing *in vivo*. We have now included an additional comment in Discussion to clarify that the clinical relevance has yet to be determined (**Page 6**).

Reviewer #2

I would like to thank the authors in replying to my concern and given additional information in the discussion section.

However, given the focus of the manuscript and the rational how the manuscript is written (focus on translation rather than biology/technique) I still believe that additional insights into the mode of action and how the cells behave *in vivo* would be an additional asset to the reader of the manuscript.

Given the changes made in the discussion would be acceptable for me.

We thank the reviewer for their comments.